# Imposing Boundary Conditions on Neural Operators
# via Learned Function Extensions

**Sepehr Mousavi** [1]   **Siddhartha Mishra** [2][3]   **Laura De Lorenzis** [1][3]

## Abstract

Neural operators have emerged as powerful surrogates for the solution of partial differential equations (PDEs), yet their ability to handle general, highly variable boundary conditions (BCs) remains limited. Existing approaches often fail when the solution operator exhibits strong sensitivity to boundary forcings. We propose a general framework for conditioning neural operators on complex non-homogeneous BCs through function extensions. Our key idea is to map boundary data to latent pseudo-extensions defined over the entire spatial domain, enabling any standard operator learning architecture to consume boundary information. The resulting operator, coupled with an arbitrary domain-to-domain neural operator, can learn rich dependencies on complex BCs and input domain functions at the same time. To benchmark this setting, we construct 18 challenging datasets spanning Poisson, linear elasticity, and hyperelasticity problems, with highly variable, mixed-type, component-wise, and multi-segment BCs on diverse geometries. Our approach achieves state-of-the-art accuracy, outperforming baselines by large margins, while requiring no hyperparameter tuning across datasets. Overall, our results demonstrate that learning boundary-to-domain extensions is an effective and practical strategy for imposing complex BCs in existing neural operator frameworks, enabling accurate and robust scientific machine learning models for a broader range of PDE-governed problems.

## 1. Introduction

A wide range of phenomena in physics and engineering are governed by systems of partial differential equations (PDEs). Due to the lack of analytical solutions in most practical settings, such problems are typically solved numerically using methods such as finite difference, finite element, or spectral methods. In many applications, including uncertainty quantification, design, control, and inverse problems, these solvers must be executed repeatedly under varying inputs (Quarteroni et al., 2015). As a result, the associated computational cost can become prohibitive, making such workflows impractical or infeasible. In this setting, machine learning (ML) models have emerged as effective data-driven surrogates for traditional numerical solvers; see (Mishra & Townsend, 2024) and the references therein. In recent years, a dominant paradigm has been to learn the underlying solution operator of PDE-governed systems directly from data generated by high-fidelity numerical solvers, using so-called *neural operators* (Kovachki et al., 2023; Bartolucci et al., 2023). The applicability, accuracy, computational efficiency, and scalability of such frameworks depend primarily on how the neural operator is parameterized. Representative examples include DeepONet (Lu et al., 2021), the Fourier Neural Operator (FNO) (Li et al., 2021; 2023), the Convolutional Neural Operator (CNO) (Raonic et al., 2024), graph-based architectures (Pfaff et al., 2021; Brandstetter et al., 2022; Li et al., 2023; Mousavi et al., 2025), and attention-based approaches (Cao, 2021; Herde et al., 2024; Wu et al., 2024; Alkin et al., 2024; Wen et al., 2025).

Conceptually, neural operators aim to learn mappings from one or more problem conditions to the corresponding solution. These conditions typically include the domain geometry, initial conditions, scalar parameters, spatially varying PDE coefficients, and boundary conditions (BCs). However, most existing *operator learning* (OL) frameworks are limited to problems with constant or weakly varying BCs across training and inference samples. In contrast, in many applications BCs constitute the primary inputs driving the behavior of the system, and the associated solution operator is highly sensitive to them. Consequently, it is crucial to equip neural operators with an explicit mechanism for conditioning on boundary data. Despite its importance, this

[1]Department of Mechanical and Process Engineering, ETH Zurich, Switzerland [2]Seminar for Applied Mathematics, ETH Zurich, Switzerland [3]ETH AI Center, Zurich, Switzerland. Correspondence to: Laura De Lorenzis <ldelorenzis@ethz.ch>.

*Proceedings of the 43$^{rd}$ International Conference on Machine Learning*, Seoul, South Korea. PMLR 306, 2026. Copyright 2026 by the author(s).

aspect has received very little attention in the literature, with only a small number of works addressing BCs through specialized treatments applied at or near the domain boundaries. Moreover, existing methods are often limited in scope: they are restricted to Cartesian geometries, fail to support complex BCs, or exhibit poor performance when complex BCs are varied across samples.

**Contributions.** In this work, we focus on OL for PDE boundary value problems (BVP) in which the solution exhibits extreme sensitivity to BCs. For parabolic and hyperbolic PDEs, the solution typically shows limited sensitivity to BCs over short time horizons. In parabolic PDEs, diffusion-like behavior causes the influence of BCs to propagate gradually over time rather than inducing an instantaneous global effect. Similarly, hyperbolic PDEs have finite propagation speed, resulting in local and time-delayed influence of the BCs. Elliptic PDEs, in contrast, are time-independent, and their solution at any point in the domain depends on the BCs imposed along the entire boundary. As a result, even small changes in BCs can lead to global variations in the solution, since there is no intrinsic propagation direction. For this reason, the current work considers elliptic PDEs as a worst-case scenario in terms of BC sensitivity. Nevertheless, all methods and techniques proposed in this work are directly applicable to initial-boundary value problems, with no expected limitations. The main contributions of this work are summarized as follows:

- A general and flexible OL framework that extends existing architectures to *explicitly incorporate* complex BCs.

- A procedure that unifies Dirichlet, Neumann, and Robin BCs into a physically consistent representation with well-balanced normalized input functions.

- A collection of 18 PDE datasets [1] (Mousavi et al., 2026) featuring diverse BCs across samples. In the most complex configuration, we consider eight boundary segments with random sizes, random locations, and randomly assigned component-wise BC types. The domains are discretized with unstructured and locally refined meshes and include highly non-convex geometries containing holes.

- Extensive numerical experiments [2] demonstrating the effectiveness of the proposed framework using graph-based and transformer-based neural operators, achieving state-of-the-art accuracy with substantial improvements over prior work across a wide range of geometries and BC configurations.

**Related work.** PENN (Horie & Mitsume, 2022) proposes correction layers that locally enforce Dirichlet and Neumann BCs in graph-based PDE solvers. This approach is

---

[1] available at doi.org/10.5281/zenodo.18377370
[2] codes available at github.com/sprmsv/olbc

evaluated on an incompressible fluid flow problem with piecewise constant BCs. However, the BCs are *topologically fixed* across samples. That is, different samples share the same BC assignments on specific boundary segments (e.g., inlet, outlet, walls), allowing ML models to implicitly infer the BCs from data even without explicit conditioning mechanisms. BOON (Saad et al., 2023) adopts a more general strategy by manipulating the kernel in the formulation of an integral operator. This approach is evaluated on hyperbolic and parabolic PDEs and compared against baselines without explicit boundary conditioning. Although the BCs are enforced exactly, the resulting improvement in global accuracy over the entire domain is marginal, suggesting a low sensitivity of the underlying solutions to the imposed BCs. BENO (Wang et al., 2024) employs a graph-based framework and decomposes the problem into a superposition of two sub-problems: one with homogeneous BCs and one with a homogeneous PDE acting in the domain. The boundary forcings are embedded using a transformer block, and are incorporated in the following message passing blocks of both branches. Other approaches include the use of ghost nodes in graph-based neural operators (Zeng et al., 2025), semi-periodic basis functions in spectral methods for homogeneous BCs (Liu et al., 2023), low-dimensional Fourier transforms applied on structured grids at the boundary (Kashi et al., 2024), and homogenization of the BCs for linear problems (Lötzsch et al., 2022).

## 2. Problem Formulation

We consider the generic time-independent PDE problem

$$\begin{cases} \mathcal{D}(a, u) = 0 & \forall x \in \Omega \\ \mathcal{B}(q, u) = 0 & \forall x \in \partial\Omega \end{cases}, \quad (1)$$

where $\Omega \subset \mathbb{R}^{d_x}$ is a $d_x$-dimensional spatial domain, $\partial\Omega \subset \mathbb{R}^{d_x}$ is its boundary (typically a low-dimensional manifold embedded in $\mathbb{R}^{d_x}$), $\mathcal{D}$ is a partial differential operator acting in the domain, $\mathcal{B}$ is a partial differential operator acting at the boundary (BCs), $u \in \mathcal{X}$ is the solution of the problem with $\mathcal{X} \subset W^{m,p}(\Omega; \mathbb{R}^{d_u})$ for suitable $m$, depending on $\mathcal{D}$, $a \in \mathcal{A}$ is a spatially varying coefficient defined in the interior with $\mathcal{A} \subset L^p(\Omega; \mathbb{R}^{d_a})$, and $q \in \mathcal{Q}$ is a spatially varying coefficient defined at the boundary with $\mathcal{Q} \subset L^p(\partial\Omega; \mathbb{R}^{d_q})$, for some $1 \le p < \infty$.

The solutions of the problem (1) can be characterized by the *solution operator* $\mathcal{G}^\dagger : \mathcal{A} \times \mathcal{Q} \to \mathcal{X}$, which maps the space-dependent domain and boundary coefficients $a$ and $q$ to the corresponding solution $u$. Our objective is to *learn a neural operator* $\mathcal{G}$ parameterized by $\theta$ that approximates this solution operator, that is, $\mathcal{G} \approx \mathcal{G}^\dagger$. Formally, we seek a parameterization such that

$$\mathcal{G}(a, q; \theta) = \mathcal{G}_\theta(a, q) \approx u = \mathcal{G}^\dagger(a, q). \quad (2)$$

The boundary operator $\mathcal{B}$ typically describes one or more

among Dirichlet, Neumann, and Robin-type BCs. To retain generality, we allow all three types to coexist, acting on disjoint boundary segments $\Omega_{D,N,R}$ that together cover the entire boundary, that is, $\partial\Omega = \partial\Omega_D \cup \partial\Omega_N \cup \partial\Omega_R$. In practice, BCs are prescribed separately on each of these boundary segments as follows:

$$\begin{cases} \mathcal{B}_D(u) = \gamma_D & x \in \partial\Omega_D, \\ \mathcal{B}_N(u) = \gamma_N & x \in \partial\Omega_N, \\ \alpha_R \odot \mathcal{B}_D(u) + \mathcal{B}_N(u) = \gamma_R & x \in \partial\Omega_R, \end{cases} \quad (3)$$

where we introduce Dirichlet and Neumann-type boundary operators, denoted by $\mathcal{B}_D$ and $\mathcal{B}_N$, acting on $u$, together with boundary functions $\gamma_D, \gamma_N, \gamma_R$, and $\alpha_R$. The symbol $\odot$ denotes the Hadamard product. For vector-valued solutions ($d_u > 1$), the BC type on a given boundary segment may differ across components. We refer to this setting as *component-wise* BCs. To accommodate this case, the above decomposition must be applied independently to each component; see Appendix C for an illustrative example.

## 3. Methods

This section outlines the main methodological approaches proposed and employed in this work. We begin by presenting a general strategy for merging and normalizing mixed-type BCs that avoids unnecessary branching while ensuring balanced distributions appropriate for training. We then explain how a standard neural operator can be extended with an additional module to incorporate BCs. Finally, we describe the three methods investigated in this work for implementing this extension.

**Merging and normalization.** In order to avoid unnecessary branching for each BC type, we propose the following abstract formulation that expresses all three BC types, without losing generality, in a unified fashion

$$\mathcal{B}(q, u) = \alpha \odot \mathcal{B}_D(u) + \beta \odot \mathcal{B}_N(u) - \gamma, \quad (4)$$

where $\alpha, \beta, \gamma : \partial\Omega \to \mathbb{R}^{d_u}$ are defined over the entire boundary. Together, they constitute the boundary coefficient function $q$. Given that Dirichlet and Neumann BCs are limit cases of a Robin BC, these boundary functions can easily be constructed from (3). Owing to their fundamentally different physical meanings, the associated forcing terms $\gamma_D, \gamma_N$, and $\gamma_R$ may differ significantly in magnitude and units. A naive unification of these BCs can therefore lead to physically inconsistent BC representations or scale imbalances, which may cause severe training instabilities. In Appendix B, we propose a principled procedure for expressing all BC types in the unified form of (4). The proposed approach preserves consistency with the original formulation, respects physical units, and yields well-balanced input functions suitable for stable training.

**Extender module.** Motivated by explicit constructions for linear problems (see Appendix A.1), we propose using extensions of the boundary functions to the whole domain as inputs to the neural operator. An *extension* of a boundary function $q$ into the domain is any function defined in the whole domain $\overline{\Omega}$ that is equal to $q$ at the boundary $\partial\Omega$. Relaxing this condition somewhat, we define a *pseudo-extension* of $q$ into the domain as any domain function that *encodes* $q$; i.e., it is possible to recover $q$ given only the pseudo-extension. For a given $q$, there exists infinitely many (pseudo-)extensions into the domain. Let us consider an *extender* $\Psi^\dagger$ as one of operators that map $q$ to a family of (pseudo-)extensions such that $\Psi^\dagger(q) = \psi$, where $\psi : \overline{\Omega} \to \mathbb{R}^{d_\psi}$ is the (pseudo-)extension. We can define a domain-to-domain operator $\Phi^\dagger$ that maps the domain coefficient and the (pseudo-)extension of the boundary coefficient to the solution of (1) such that

$$\Phi^\dagger(a, \psi) = \mathcal{G}^\dagger(a, q). \quad (5)$$

In Appendix A.3, we show that for linear problems and with certain conditions on the pseudo-extension, this operator exists and is affine in its second variable. Since both its inputs and its output are functions that are defined in the entire domain, $\Phi^\dagger$ can be approximated using conventional neural operators that are designed for such mappings. We rely on the existing methods for resolving the dependence on the domain function $a$, and focus on the dependence of the operator on the space-dependent boundary function $q$. The problem hence boils down to finding a suitable extender that can provide $\psi$ given $q$. Considering a potentially parameterized *extender* $\Psi_\theta$ and a parameterized *core* neural operator $\Phi_\theta$ that processes domain functions, we construct a learnable neural operator as

$$\mathcal{G}_\theta(a, q) = \Phi_\theta\left(\begin{bmatrix} a \\ \Psi_\theta(q) \end{bmatrix}\right) = \Phi_\theta\left(\begin{bmatrix} a \\ \psi \end{bmatrix}\right). \quad (6)$$

**Zero Extensions (0X).** The most straightforward choice is to consider zero padding, where $\psi$ matches $q$ at the boundary and is zero otherwise: $\psi(x) = \{0 \text{ in } \Omega, \ q \text{ on } \partial\Omega\}$. Using a binary mask that indicates the boundary, real zeros in $q$ are distinguished from the padded zeros.

**Harmonic Extensions (HX).** A natural choice for $\psi$ is the harmonic extension, which is defined as the unique smoothest extension and can be obtained by solving the Laplace equation $\{\Delta\psi = 0 \text{ in } \Omega, \ \psi = q \text{ on } \partial\Omega\}$. The harmonic extension has minimal Dirichlet energy (Evans, 2022) among all possible extensions that match $q$ at the boundary. One can obtain other extensions in a similar manner either by solving a non-homogeneous Poisson equation, e.g., to obtain different degrees of locality, or by considering a diffusion coefficient. In this case, stacking multiple extensions together is possible and potentially provides a richer encoding of $q$.

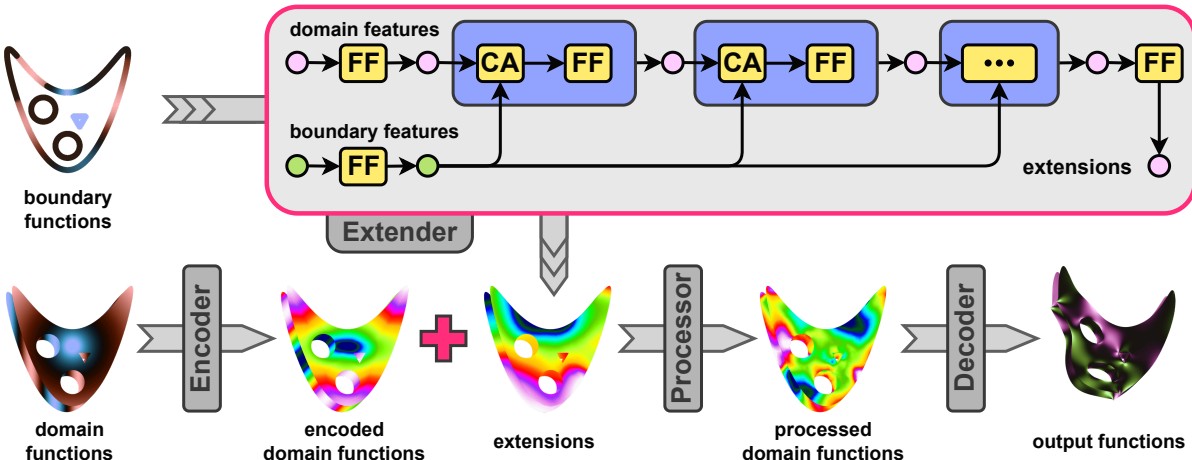

*Figure 1.* Overview of an *extended* encode-process-decode OL framework. The bottom row shows the placement of the proposed extender module into the framework; the top row shows our proposed attention-based architecture as a realization of the extender. Domain features and boundary features are depicted by pink and green dots, respectively. Multiple channels of domain functions are indicated by multiple layers. Initial geometrical domain features are progressively informed by the boundary features through successive cross-attention (CA) layers and independent feed-forward (FF) blocks. For clarity, residual connections between domain features are omitted in the figure. The output of the extender is directly concatenated with the -downsampled- output of the encoder block.

**Learned Pseudo-extensions (LX).** Computing harmonic extensions requires solving a Laplace equation, usually by the finite element method (FEM), which can be computationally expensive in certain applications and may become a bottleneck at inference time in an ML setting. Consequently, we seek an extender that can be evaluated efficiently. To this end, we propose a learnable extender that can be trained jointly with the core neural operator in an end-to-end fashion. To increase the expressive flexibility of this module, we relax the requirement of exact boundary matching and instead aim to construct pseudo-extensions that effectively embed all the information contained in $q$. In what follows, we propose an attention-based architecture that can be used as a learnable extender. Note that the inputs and the outputs of such architectures are discretized functions, which we denoted by bold letters.

We consider a set of domain coordinates $\mathbf{x} := \{\mathbf{x}_i \in \Omega\}_i$, a set of boundary coordinates $\partial\mathbf{x} := \{\mathbf{x}_j \in \partial\Omega\}_j$, discretized values of geometry-related domain functions (e.g., distance to the closest boundary) at all domain coordinates $\mathbf{a} := \{\mathbf{a}_i = a(\mathbf{x}_i) \in \mathbb{R}^{d_a}\}_{\mathbf{x}_i \in \mathbf{x}}$, and discretized values of boundary functions at all boundary coordinates $\mathbf{q} := \{\mathbf{q}_j = q(\mathbf{x}_j) \in \mathbb{R}^{d_q}\}_{\mathbf{x}_j \in \partial\mathbf{x}}$. In the initial steps of neural operator architectures, the raw discretized values are typically downsampled onto a coarser (in space) but higher-dimensional (in components) representation (Li et al., 2023; Mousavi et al., 2025; Wen et al., 2025). We consider $\tilde{\mathbf{a}}$ and $\tilde{\mathbf{q}}$ as the downsampled versions of domain and boundary features. In this work, we directly process raw boundary functions and have $\tilde{\mathbf{q}} = \mathbf{q}$. Our aim is to find discretized values $\boldsymbol{\psi}_i = \psi(\mathbf{x}_i)$ at every downsampled domain coordinate. First, all features are independently passed through

feed-forward (FF) layers,

$$\tilde{\mathbf{a}}_i^{(0)} = FF_{\text{domain}}^{(0)}(\tilde{\mathbf{a}}_i), \quad \tilde{\mathbf{q}}_j^{(0)} = FF_{\text{boundary}}^{(0)}(\tilde{\mathbf{q}}_j).$$

After that, the boundary features get *fed* into the domain features multiple times via cross-attention (CA) layers followed by FF layers, with residual connections between these blocks. At each block $k \in \{1, \dots, K\}$, we update the latent domain features as

$$\tilde{\mathbf{a}}_i^{(k)} = \tilde{\mathbf{a}}_i^{(k-1)} + FF^{(k)}\left(CA^{(k)}(\tilde{\mathbf{a}}_i^{(k-1)}, \tilde{\mathbf{q}}^{(0)})\right).$$

Finally, the *boundary-fed* domain features get passed through a final FF layer to get the pseudo-extension in the desired dimension,

$$\boldsymbol{\psi}_i = FF^{(K+1)}(\tilde{\mathbf{a}}_i^{(K)}).$$

Figure 1 provides an overview of the proposed architecture and illustrates how it can be integrated into a standard OL framework based on the encode-process-decode paradigm (Sanchez-Gonzalez et al., 2020). In particular, the encoder typically incorporates a downsampling step applied to the input coordinates. In this setting, the boundary extensions can be constructed directly in the downsampled space, which significantly reduces the computational cost of the attention mechanism and, consequently, of the extender itself. Similar downsampling strategies can also be applied to the raw boundary features.

## 4. Results

**Datasets.** We consider the Poisson, linear elasticity, and hyperelasticity problems, resulting in a total of 18 datasets (Mousavi et al., 2026). In order to isolate the impact

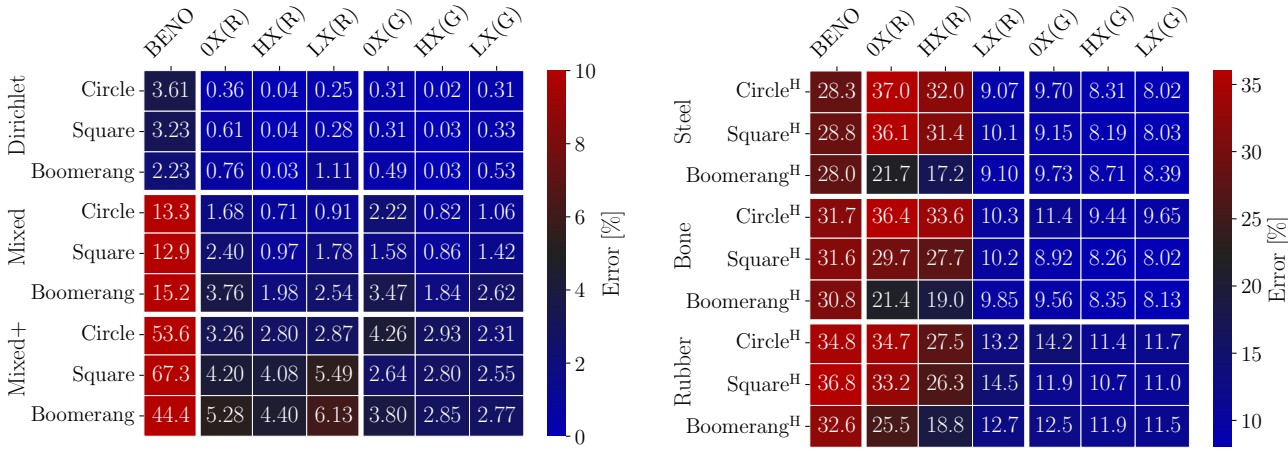

*(a)* Poisson problems.

*(b)* Elasticity and hyperelasticity problems.

*Figure 2.* Median relative $L^2$ error [%] of 256 test samples with different methods (columns). Each row corresponds to one dataset, see Appendix C for details. All trainings are done using 8,192 training and 256 validation samples. Geometries containing holes are indicated with H. Extension methods are combined with RIGNO (R) and GAOT (G). The reported errors for the elasticity and hyperelasticity problems are the average of all independent components of displacement, strain, and stress fields. These errors are reported separately in Appendix F.1 along with other metrics (described in Appendix D) on each field.

of the BCs, we fix the geometry and all PDE parameters in the samples of our datasets. However, we consider a total of six geometries in different datasets to test the capabilities of the proposed methods on arbitrary domains. With the BCs as the only source of variation, the solutions are very sensitive to them, as the target solution operator is a mapping from only the boundary functions to the solution. Our configurations include mixed BC types (e.g., Dirichlet, Neumann, and Robin on different segments), component-wise BCs (e.g., Neumann in one direction, Dirichlet in the other), up to eight boundary segments with random sizes and random locations, randomly assigned BC types, and random sinusoidal boundary forcings. Details are available in Appendix C.

We evaluate the proposed framework by extending two state-of-the-art neural operator architectures designed for spatially unstructured data. To demonstrate the generality and flexibility of our approach, we consider both the graph-based RIGNO architecture (Mousavi et al., 2025) and the transformer-based GAOT architecture (Wen et al., 2025). Figure 2 reports the test errors obtained with different extension methods, along with the BENO approach of (Wang et al., 2024), which is the only available method that can be applied to all the datasets that we consider here. Unless otherwise stated, all experiments are conducted using fixed hyperparameters for both the extender and the core architectures, along with fixed hardware and optimization settings, following the configurations detailed in Appendix E. In most experiments, RIGNO is used as the core architecture unless otherwise specified.

**BCs for simple problems can be learned via basic extensions.** We observe from Figure 2a that harmonic extensions consistently achieve the lowest errors. This is not surprising as harmonic extensions are themselves solutions of a Laplace problem and are therefore closely related to the solution of the target Poisson problem; a broader discussion of this connection is provided in Appendix A.2. Importantly, solving the original Poisson problem is computationally cheaper than computing the corresponding harmonic extensions. In contrast, zero extensions incur no additional computational cost and can be readily integrated into any OL framework. Although they are generally less accurate than learned extensions, zero extensions lead to errors that are reasonably low and, in some cases, comparable to those achieved with learned methods. Notably, the BENO baseline produces significantly higher errors compared to domain-to-domain neural operators using the simple zero extension. We argue that this performance gap stems primarily from BENO's reliance on a global BC encoding within its core, which erodes the preservation of important local spatial structure. Given BENO's strong performance on the Poisson problem from its original work (Wang et al., 2024), where errors typically remain below one percent, the higher errors obtained here with BENO further highlight the challenging nature of the datasets introduced in the present work.

**BCs and domain sources can be learned at the same time.** In many applications, the solution depends not only on the BCs but is also sensitive to input domain functions, such as initial conditions in time-dependent problems. OL frameworks must therefore be capable of capturing both types of dependencies simultaneously. To evaluate the pro-

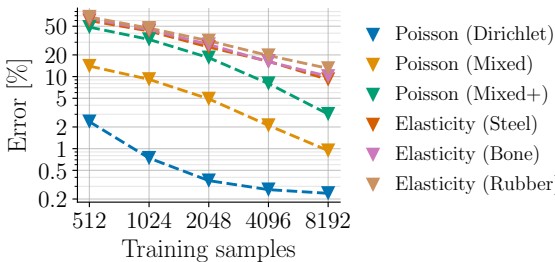

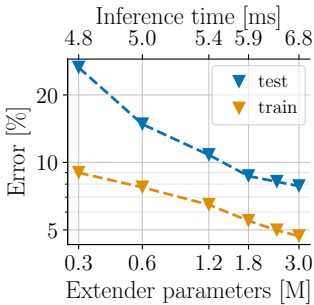

*Figure 3.* Scaling of model accuracy with the size of the training dataset. Median relative $L^2$ test errors on 256 samples are reported. The Poisson and elasticity problems are considered on the Circle and the Circle[H] geometries, respectively. All trainings are done with a model of size 3.6M (1.8M for the extender and 1.8M for the core).

*Figure 4.* Scaling of model accuracy with the size of the extender module. Median relative $L^2$ errors on 256 test samples and 8,192 training samples are reported. We use a core module with 1.8M parameters and train on the elasticity (Steel) dataset on the Circle[H] geometry.

posed framework in this setting, we consider a configuration with jointly varying BCs and source functions. The results shown in Figure 2a (Mixed+) demonstrate that the extended RIGNO and GAOT architectures successfully handle the coexistence of these variations, whereas the BENO baseline fails to resolve the combined effect of varying BCs and domain inputs.

**Learned extensions can accurately resolve BCs for complex problems.** The second set of benchmarks, shown in Figure 2b, considers datasets of elastic and hyperelastic materials subjected to boundary loadings. The physical problems considered here exhibit a more complex dependence on BCs than Poisson's equation. On a single boundary segment, two BCs are applied independently, for example clamped in one direction and free in the other. As a result, the solution is sensitive not only to the magnitude of the imposed BCs but also to their combined composition. In addition, sharp transitions in the boundary functions induce singularities and multi-scale features in the solution fields; visualizations of representative dataset samples are provided in Appendix C. In this setting, learned extensions consistently outperform the other methods by a significant margin. Notably, with a PDE different from the Laplace equation, harmonic extensions no longer provide the same level of accuracy. Given the high computational cost of computing the harmonic extensions, and the poor performance of the BENO baseline, zero and learned extensions emerge as the only viable practical approaches. With RIGNO as the core architecture, learned extensions provide a clear advantage, decreasing the average test error from 30.6% to 11.0%. As illustrated in Appendix F.3, while very large RIGNO models can achieve high accuracy even with zero extensions, incorporating a learned extension enables comparable or better performance with substantially smaller core models. GAOT, in contrast, attains reasonable accuracy at moderate model sizes when using zero extensions, but its performance does not improve with increasing model capacity (see Appendix F.3). When equipped with a learnable extender, however,

GAOT consistently benefits, with an average error reduction of 13% (1.4% in absolute terms) across the nine datasets shown in Figure 2b. Finally, we compare the accuracy and inference time of our proposed ML models with a FEM solver on the hyperelasticity problem. The results in Table 1 indicate that our models achieve accuracies comparable to FEM solutions obtained by using a mesh of 11.8 thousand nodes, while being roughly 20,000 times faster at inference.

*Table 1.* Comparison of accuracy and inference time between FEM (single CPU) and the learned extensions (single GPU) with RIGNO (R) and GAOT (G) as backbone. We study the hyperelasticity problem (Rubber) on the Circle[H] geometry, and report relative $L^2$ errors on the independent components of the strain field. Results of FEM with a fine mesh (44K nodes) are used as reference.

| Method | Size | Error | Time |
|--------|------|-------|------|
| FEM | 6.1K mesh nodes | 16.3% | 52.3 s |
| FEM | 11.8K mesh nodes | 12.2% | 125.0 s |
| LX(R) | 3.6M parameters | 13.8% | 5.9 ms |
| LX(G) | 3.9M parameters | 13.3% | 3.7 ms |

**The accuracy scales with the size of the dataset and the extender.** A suitable deep learning architecture for scientific applications must scale effectively with both the size of the training dataset and the size of the model (Kaplan et al., 2020; Lanthaler et al., 2022; Herde et al., 2024). When the number of training samples is limited, the model tends to overfit the training data, leading to degraded performance on unseen samples. As the number of training samples increases, the data distribution is represented more accurately and the gap between training and test performance gradually decreases. With such scaling behavior, increasingly high accuracy can be achieved by providing more data. To evaluate this property, we train the model using progressively larger training sets and measure its performance on unseen inputs. The results are shown in Figure 3. Across all problem settings, the error consistently decreases as more training data are provided. The rate of this decrease is governed by two factors: the optimization performance, reflected in

the training error, and the generalization gap, defined as the difference between training and test errors. As shown in Appendix F.2, more data generally help closing the generalization gap. With sufficiently large training datasets, model accuracy is no longer limited by data availability but instead by the representational capacity of the model. This transition is clearly visible in Figure 3 for the Poisson problem with Dirichlet BCs. This problem is the least challenging considered here and can be learned with relatively few training samples. With approximately 2,048 training examples, the model approaches its capacity, and additional data no longer lead to significant performance gains. It is therefore equally important that model capacity scales appropriately with model size. We assess this property by increasing the size of the extender and reporting the corresponding results in Figure 4. The training error consistently decreases as the model size grows, indicating that the expressive capacity of the model increases accordingly.

**Learned extensions generalize to unseen geometries, unseen BC types, and unseen physics.** An ML model that truly learns the governing physics of a problem, rather than memorizing a specific problem configuration, should generalize to scenarios that differ substantially from the training setup (Subramanian et al., 2023; McCabe et al., 2024; Herde et al., 2024). In the context of the present work, such differences may arise from changes in domain geometry, BC types, or the underlying physics. To assess whether the knowledge learned from a *base setup* is transferable to a fundamentally different *target setup*, we pre-train the extender on the base setup, fine-tune it using 512 samples from the target setup, and evaluate its performance on unseen samples from the target setup. In the limiting case where the information learned from the base setup is not relevant to the target setup, the final accuracy is expected to be comparable to that obtained from random initialization. At the other extreme, if the learned knowledge is fully transferable, the final accuracy should match or slightly exceed the accuracy achieved on the base setup. Figure 5 presents results for multiple base datasets, with the target dataset corresponding to the Poisson problem on the Circle geometry with the Mixed BC configuration. As shown in the figure, pre-training on the same problem with Dirichlet BCs reduces the test error by a substantial margin of 4.7%. In this case, Neumann and Robin BCs, which account for approximately two thirds of the boundary segments on average, are entirely new to the model and are encountered only during fine-tuning on the target dataset. The extender also demonstrates strong generalization across different geometries. When pre-training is performed on the Boomerang or Square geometries, the improvements are even more pronounced, with error reductions of 7.9% and 10.3%, respectively. Notably, pre-training on the Square geometry yields a final error much closer to the test error of the base

dataset, referred to as the *base error* and indicated by a star in Figure 5. Using the base error as a reference, the relative error reduction is 84.6% for the Square geometry and 69.3% for the Boomerang geometry. This behavior can be intuitively explained by the greater geometric similarity between the Square and the Circle domains compared to the highly non-convex Boomerang geometry. Additional transfer learning experiments are reported in Appendix F.4. There, we show that pre-training on a linear elasticity problem can reduce the error for a (non-linear) hyperelasticity problem from 67% to 23%. Moreover, we demonstrate that suitable pre-training enables accurate performance with significantly smaller target datasets, achieving strong results with up to 64 times fewer samples, using as few as 8 target samples for fine-tuning.

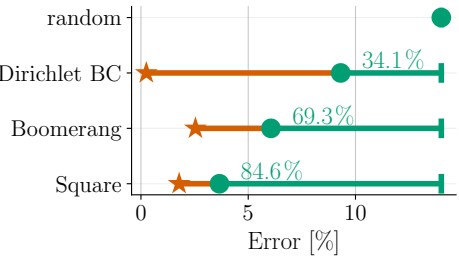

*Figure 5.* Results of transfer learning experiments for the Poisson problem on the Circle domain with the Mixed BCs configuration as the target dataset. The extender shows strong generalization to different geometries and previously unseen BC types. The test error of the pre-trained models on the base dataset is marked with a star. The green dots represent the test error on the target dataset after fine-tuning with 512 samples. The relative accuracy gain compared to the random initialization is shown above each line, using the accuracy of the pre-trained model (star) as reference.

**Trained extenders are robust to input noise.** ML models often lack robustness and can respond poorly to distribution shifts caused by noisy inputs at inference time. In the context of the present work, the primary inputs to the extender are boundary functions that define the BCs. Depending on the application, noise in these inputs may arise from different sources, such as measurement errors in control systems and inverse problems, or upstream approximations in monitoring systems or coupled multi-physics simulations. Since these applications are key motivations for ML-based surrogate models, robustness to input noise is of critical importance. To assess this property, we evaluate the proposed framework by perturbing the boundary inputs with additive Gaussian noise and computing the resulting error relative to clean reference outputs. The results are shown in Figure 6. As illustrated, the framework exhibits strong robustness to noise, with only minor degradation in performance for moderate noise levels. Notably, even with 10% noise in the input boundary functions, the test error remains below 7% across all datasets. Among the different BC types, Dirichlet conditions exhibit the highest sensitivity, owing to their direct influence on the solution values. When Neumann and Robin

conditions are included, the error increases at a slower rate for noise levels up to 1%. In the Mixed+ configuration, the performance degradation is almost negligible for noise levels up to 2%. This behavior can be attributed to the reduced sensitivity to the BCs when noise-free input domain functions are involved.

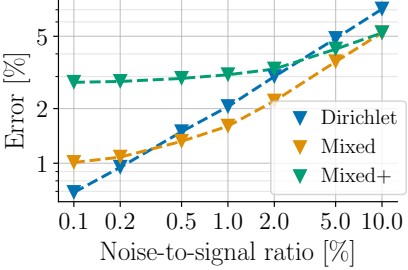

*Figure 6.* Model accuracy on the Poisson problem on the Circle domain against the level of Gaussian noise in the input boundary functions. The y-axis shows relative $L^2$ test errors on 256 samples, and the x-axis shows the ratio between the mean of the squared noise and the mean of the squared signal in percentage. This metric is the inverse of the widely used signal-to-noise ratio.

**Further experiments.** Additional experimental results are provided in Appendix F. Appendix F.5 presents results obtained by repeating the same training procedure with 10 different random seeds, providing additional insight into the variability and uncertainty of the reported results. In Appendix F.6, we report an ablation study in which selected components of the extender architecture are removed to assess their contribution to the overall performance. In Appendix F.7 we show that, by using a masking strategy within the cross-attention blocks, the trained extender remains robust to changes in the resolution of the boundary functions. In particular, accuracy is preserved when the extender is evaluated at higher resolutions, which enables more efficient training using lower resolution boundary data. In Appendix F.8, we take a closer look at the attention scores across the layers of the extender and investigate the mechanisms learned by the model. Finally, Appendix G includes plots of error histograms, visualizations of a few representative samples, and the learned extensions.

## 5. Discussion

In this work, we propose an extension of existing neural operator frameworks that enables the incorporation of complex BCs alongside standard input domain functions such as initial conditions and source terms. The proposed framework (see Figure 1) is fully compatible with the widely used encode–process–decode paradigm (Sanchez-Gonzalez et al., 2020), and augments it with an additional module that ingests raw boundary functions and projects them into latent domain functions suitable for a domain-to-domain processor. To handle varying BC types and heterogeneous boundary

segments, we propose a merging and normalization scheme that provides well-balanced functions defined over the entire boundary, while preserving fidelity to the original BCs. We explore three strategies for the boundary-to-domain extender module: zero extensions, harmonic extensions, and learned pseudo-extensions. For the latter, we propose an architecture based on successive cross-attention blocks that efficiently propagate boundary information throughout the domain. This module can be trained end-to-end with the rest of the network and integrates seamlessly into existing neural operator frameworks with minimal architectural intrusion.

To evaluate the proposed framework, we introduce a suite of 18 datasets featuring extreme per-sample variations in BCs, which pose a significant challenge to the existing approaches as they are either inapplicable or result in consistently high test errors. We assess our framework using both graph-based and transformer-based neural operators, employing identical hyperparameters in both cases. Our results show that while basic extensions can yield reasonable accuracy in some settings, their performance is inconsistent across different problems and backbone architectures. In particular, when combined with a graph-based backbone, basic extensions achieve low test errors on relatively simple problems but struggle on more complex tasks such as elastic bodies subject to boundary forces. In contrast, learned extender modules achieve significantly lower errors on complex problems, while matching the performance of basic extensions on simpler tasks, indicating that they do not introduce prohibitive optimization difficulties. The consistent performance across backbones demonstrates that the proposed framework is general, flexible, and agnostic to the choice of neural operator architecture.

**Limitations.** In the proposed architecture for learnable extenders, we apply cross-attention between boundary nodes and downsampled domain nodes. Since the number of boundary nodes is typically much smaller than the number of domain nodes, this operation remains relatively efficient. However, the architecture could be further optimized by introducing an additional downsampling module to reduce the effective number of boundary nodes participating in the attention mechanism. We leave this optimization to future work. Moreover, the experiments in this work are limited to two-dimensional problems. While the coordinate-based mechanisms used in the proposed learnable extender make the approach readily applicable to three-dimensional (3D) settings, a direct evaluation in 3D is constrained by the lack of publicly available benchmarks that exhibit the level of BC variability considered here. Nevertheless, the datasets studied in this work are comparable to typical 3D problems in terms of complexity and spatial resolution. Another direction for future research is the integration of extenders into PDE foundation models (Hao et al., 2024; Herde et al.,

2024). In light of the strong transfer learning performance observed in our experiments, pre-training extenders on large-scale datasets that span diverse physical systems and domain geometries appears particularly promising.

## Impact Statement

This paper presents work whose goal is to advance the field of machine learning for science and engineering. There are many potential societal consequences of our work, none of which we feel must be specifically highlighted here.

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

# Appendix

## A. Theory

### A.1. The Poisson equation with non-homogeneous BCs

Let us consider the Poisson equation with non-homogeneous Robin BCs,

$$\begin{cases} -\Delta u = f & \forall x \in \Omega, \\ c_D u + c_N \nabla_{\hat{n}} u = \gamma & \forall x \in \partial\Omega, \end{cases} \tag{7}$$

where $u : \overline{\Omega} \to \mathbb{R}$ is the solution function, $f : \Omega \to \mathbb{R}$ is the domain source function, $\gamma : \partial\Omega \to \mathbb{R}$ is the boundary source function, $c_D, c_N \in \mathbb{R}$ are two scalar constants, and $\hat{n}$ is the outward unit vector normal to the boundary. Assume there exists a pseudo-extension $\psi : \overline{\Omega} \to \mathbb{R}$ that satisfies the BCs, that is, $c_D \psi + c_N \nabla_{\hat{n}} \psi = \gamma$ at the boundary $\partial\Omega$. Given the pseudo-extension $\psi$, let us consider $w : \overline{\Omega} \to \mathbb{R}$ as the solution of the following Poisson problem,

$$\begin{cases} -\Delta w = f + \Delta\psi & \forall x \in \Omega, \\ c_D w + c_N \nabla_{\hat{n}} w = 0 & \forall x \in \partial\Omega. \end{cases} \tag{8}$$

We observe that $u = w + \psi$ satisfies Problem (7). Given this context, if a suitable pseudo-extension $\psi$ is known, the solution of Problem (7) can be obtained by solving Problem (8) which has the advantage of having homogeneous BCs.

### A.2. Harmonic extension and the Poisson equation with Dirichlet BCs

Consider the Poisson problem given in (7) with pure Dirichlet BCs, $c_D = 1, c_N = 0$. Assume $\psi$ is the harmonic extension of the boundary source function $\gamma$, which satisfies the BCs of Problem (7) by definition. Therefore, the solution to this problem can be computed as $u = w + \psi$, where $w$ is obtained by solving the problem

$$\begin{cases} -\Delta w = f & \forall x \in \Omega, \\ w = 0 & \forall x \in \partial\Omega. \end{cases} \tag{9}$$

Observe that, for a fixed $f$, the solution $w$ to this problem does not change. In this setting, the task of learning the mapping from $\psi$ and $f$ to the solution $u$ boils down to memorizing a fixed function $w$, and adding it to the given harmonic extension $\psi$.

### A.3. Existence and linearity of the domain-to-domain operator for linear problems

Building on the example given in Appendix A.1, we can consider linear problems in a more general setting. Consider Problem (1) and the settings described thereafter. Here, we consider a subclass of such problems where the differential operator $\mathcal{D}$ can be written in the form $\mathcal{D}(a, u) = \hat{\mathcal{D}}(\hat{a}, u) + \tilde{a}$, with $\hat{\mathcal{D}}$ linear in $u$ and $(\hat{a}, \tilde{a})$ a trivial split of the domain coefficients $a$. Similarly, we assume the boundary differential operator $\mathcal{B}$ takes the form $\mathcal{B}(q, u) = \hat{\mathcal{B}}(u) + q$ with a linear $\hat{\mathcal{B}}$. Additionally, we assume the existence of a pseudo-extension $\psi$ that extends $q$ into the domain such that $\mathcal{B}(q, \psi) = 0$ on the boundary. In this settings, we show that the operator introduced in (5) exists and is affine in its second input, the pseudo-extension $\psi$.

To this end, let us consider $w : \overline{\Omega} \to \mathbb{R}^{d_u}$ as the solution to the problem

$$\begin{cases} \mathcal{D}(a, w) + \hat{\mathcal{D}}(\hat{a}, \psi) = 0 & \forall x \in \Omega, \\ \hat{\mathcal{B}}(w) = 0 & \forall x \in \partial\Omega, \end{cases} \tag{10}$$

with $\mathcal{W}$ as its corresponding solution operator, such that $\mathcal{W}(a, \psi) = w$. Note that this problem has a homogeneous BC and the operator $\mathcal{W}$ is only conditioned on domain functions.

We define the domain-to-domain operator $\Phi^\dagger$ as

$$\Phi^\dagger (a, \psi) := \psi + \mathcal{W}(a, \psi) = \psi + w. \tag{11}$$

By applying the operators $\mathcal{D}$ and $\mathcal{B}$ on the function $\psi + w$, we show that the output of $\Phi^\dagger$ is a solution to Problem (1):

$$\begin{cases} \mathcal{D}(a, \psi + w) = \hat{\mathcal{D}}(\hat{a}, \psi + w) + \tilde{a} = \hat{\mathcal{D}}(\hat{a}, \psi) + \underbrace{\hat{\mathcal{D}}(\hat{a}, w) + \tilde{a}}_{\mathcal{D}(a, w)} = 0 & \forall x \in \Omega, \\ \mathcal{B}(q, \psi + w) = \hat{\mathcal{B}}(\psi + w) + q = \hat{\mathcal{B}}(w) + \underbrace{\hat{\mathcal{B}}(\psi) + q}_{\mathcal{B}(q, \psi)} = 0 & \forall x \in \partial\Omega, \end{cases} \tag{12}$$

where we use (10), the linearity of $\hat{\mathcal{D}}$ in its second variable, the linearity of $\hat{\mathcal{B}}$, and the condition of the pseudo-extension $\psi$ at the boundary.

Furthermore, in this setting, the operator $\mathcal{W}$ is affine in its second variable (the pseudo-extension). In order to show that, let us consider two arbitrary pseudo-extensions $\psi_1, \psi_2$, and their corresponding solutions $w_1 = \mathcal{W}(a, \psi_1)$ and $w_2 = \mathcal{W}(a, \psi_2)$. The operator $\mathcal{W}$ is affine in its second variable iff for any $\lambda \in [0, 1]$ we have

$$\mathcal{W}(a, \lambda\psi_1 + (1 - \lambda)\psi_2) = \lambda\mathcal{W}(a, \psi_1) + (1 - \lambda)\mathcal{W}(a, \psi_2) = \lambda w_1 + (1 - \lambda)w_2, \tag{13}$$

that is,

$$\begin{cases} \mathcal{D}(a, \lambda w_1 + (1 - \lambda)w_2) + \hat{\mathcal{D}}(\hat{a}, \lambda\psi_1 + (1 - \lambda)\psi_2) = 0 & \forall x \in \Omega, \\ \hat{\mathcal{B}}(\lambda w_1 + (1 - \lambda)w_2) = 0 & \forall x \in \partial\Omega. \end{cases} \tag{14}$$

Given the linearity of $\hat{\mathcal{B}}$ and the definitions of $w_1$ and $w_2$, the condition on the boundary is satisfied. The condition in the domain is also satisfied:

$$\begin{aligned} \mathcal{D}&(a, \lambda w_1 + (1 - \lambda)w_2) + \hat{\mathcal{D}}(\hat{a}, \lambda\psi_1 + (1 - \lambda)\psi_2) \\ &= \hat{\mathcal{D}}(\hat{a}, \lambda w_1 + (1 - \lambda)w_2) + \tilde{a} + \hat{\mathcal{D}}(\hat{a}, \lambda\psi_1 + (1 - \lambda)\psi_2) \\ &= \lambda\hat{\mathcal{D}}(\hat{a}, w_1) + \lambda\tilde{a} + \lambda\hat{\mathcal{D}}(\hat{a}, \psi_1) + (1 - \lambda)\hat{\mathcal{D}}(\hat{a}, w_2) + (1 - \lambda)\tilde{a} + (1 - \lambda)\hat{\mathcal{D}}(\hat{a}, \psi_2) \\ &= \lambda\left[\mathcal{D}(a, w_1) + \hat{\mathcal{D}}(\hat{a}, \psi_1)\right] + (1 - \lambda)\left[\mathcal{D}(a, w_2) + \hat{\mathcal{D}}(\hat{a}, \psi_2)\right] = 0, \end{aligned}$$

where we used the linearity of $\hat{\mathcal{D}}$ in its second variable and the definitions of $w_1$ and $w_2$. The affinity of $\Phi^\dagger$ in its second variable $\psi$ directly follows.

## B. Merging and normalization of input boundary functions

Dirichlet, Neumann, and Robin BCs are fundamentally different. In solid mechanics, for example, Dirichlet BCs typically prescribe displacements, whereas Neumann BCs prescribe applied forces. Hence, the unification of all these BCs into the Robin form in (4), despite non-dimensionalization, can lead to severe scale imbalances in the resulting boundary functions, particularly in the source term $\gamma$. This imbalance arises because $\gamma_D$ and $\gamma_N$ may differ substantially in magnitude.

Standard normalization applied to such imbalanced variables preserves these discrepancies and can introduce significant difficulties during gradient-based training. In our experience, careless normalization of raw boundary inputs alone can lead to extremely poor performance. When Robin BCs are absent, independently normalizing $\gamma_D$ and $\gamma_N$ prior to merging is often sufficient. However, since Robin BCs involve the same differential operators as Dirichlet and Neumann BCs, improper normalization can introduce inconsistencies with the original physical formulation. To address these issues, we propose a principled, step-by-step procedure for merging and normalizing BCs that removes physical units, remains consistent with the original BC definitions, and yields well-balanced boundary representations suitable for learning.

Starting from the BCs in (3), the first step is to balance the potentially different scales of $\gamma_D$ and $\gamma_N$ through a shift-and-scale transformation. For simplicity, we describe all steps for the scalar case $u$. For vector-valued solutions with component-wise BCs, the procedure must be applied independently to each component. Let $(\mu_D, \sigma_D)$ and $(\mu_N, \sigma_N)$ denote the mean and standard deviation of $\gamma_D$ and $\gamma_N$, respectively, computed over the entire training dataset. We then define the normalized input functions as

$$\tilde{\gamma}_D = \frac{\gamma_D - \mu_D}{\sigma_D}, \quad \tilde{\gamma}_N = \frac{\gamma_N - \mu_N}{\sigma_N}, \quad \tilde{\alpha}_R = \alpha_R \sigma_D, \quad \tilde{\beta}_R = \sigma_N, \quad \tilde{\gamma}_R = \gamma_R - \alpha_R \mu_D - \mu_N.$$

Next, we further scale the coefficients of the Robin BCs to avoid redundancies,

$$\hat{\alpha}_R = \frac{\tilde{\alpha}_R}{\sqrt{\tilde{\alpha}_R^2 + \tilde{\beta}_R^2}}, \quad \hat{\beta}_R = \frac{\tilde{\beta}_R}{\sqrt{\tilde{\alpha}_R^2 + \tilde{\beta}_R^2}}, \quad \hat{\gamma}_R = \frac{\tilde{\gamma}_R}{\sqrt{\tilde{\alpha}_R^2 + \tilde{\beta}_R^2}}.$$

With these transformations, the BCs in (3) can be rewritten in the unified Robin form given in (4) as

$$\alpha \tilde{\mathcal{B}}_D(u) + \beta \tilde{\mathcal{B}}_N(u) = \gamma \qquad \forall x \in \partial\Omega, \tag{15}$$

where we introduce normalized boundary operators

$$\tilde{\mathcal{B}}_D(u) = \frac{\mathcal{B}_D(u) - \mu_D}{\sigma_D}, \qquad \tilde{\mathcal{B}}_N(u) = \frac{\mathcal{B}_N(u) - \mu_N}{\sigma_N},$$

and define the boundary functions

$$\alpha = \begin{cases} 1 & \partial\Omega_D \\ 0 & \partial\Omega_N \\ \hat{\alpha}_R & \partial\Omega_R \end{cases}, \qquad \beta = \begin{cases} 0 & \partial\Omega_D \\ 1 & \partial\Omega_N \\ \hat{\beta}_R & \partial\Omega_R \end{cases}, \qquad \gamma = \begin{cases} \tilde{\gamma}_D & \partial\Omega_D \\ \tilde{\gamma}_N & \partial\Omega_N \\ \hat{\gamma}_R & \partial\Omega_R \end{cases}.$$

The proposed representation of the BCs is fully consistent with the original representation in (3). To show this, we simply substitute the values of $\alpha$, $\beta$, and $\gamma$ in (15) for each boundary segment. For the Dirichlet ($\partial\Omega_D$) and Neumann ($\partial\Omega_N$) segments, this is straightforward as similar shift-and-scale transformations are done on the normalized boundary operators and the source functions. For the Robin

segment $\partial\Omega_R$, we have

$$\left(\frac{\tilde{\alpha}_R}{\sqrt{\tilde{\alpha}_R^2 + \tilde{\beta}_R^2}}\right)\frac{\mathcal{B}_D(u) - \mu_D}{\sigma_D} + \left(\frac{\tilde{\beta}_R}{\sqrt{\tilde{\alpha}_R^2 + \tilde{\beta}_R^2}}\right)\frac{\mathcal{B}_N(u) - \mu_N}{\sigma_N} = \frac{\tilde{\gamma}_R}{\sqrt{\tilde{\alpha}_R^2 + \tilde{\beta}_R^2}}.$$

Multiplying the above equality by $\sqrt{\tilde{\alpha}_R^2 + \tilde{\beta}_R^2}$, we obtain

$$\frac{\tilde{\alpha}_R}{\sigma_D}\left(\mathcal{B}_D(u) - \mu_D\right) + \frac{\tilde{\beta}_R}{\sigma_N}\left(\mathcal{B}_N(u) - \mu_N\right) = \tilde{\gamma}_R.$$

Substituting the normalized Robin coefficients, we obtain

$$\alpha_R\left(\mathcal{B}_D(u) - \mu_D\right) + \left(\mathcal{B}_N(u) - \mu_N\right) = \gamma_R - \alpha_R\mu_D - \mu_N,$$

from which the Robin BC of (3) can directly be retrieved.

## C. Datasets

Given the lack of suitable and publicly available datasets with high sensitivity to BCs, we generate a collection of 18 new datasets [3] (Mousavi et al., 2026), covering the Poisson equation, the linear elasticity problem, and the non-linear hyperelasticity problem. We solve these problems on convex and non-convex geometries containing holes. In each dataset, we fix the geometry and all PDE parameters (except in the Mixed+ configuration, see Section C.1.3) to maximize the sensitivity of the underlying solution operator to the BCs. Within this collection, we cover the level of flexibility in terms of BCs that is expected in many scientific and engineering applications, while having different levels of complexity across configurations. In the simplest case, we consider a single boundary segment covering the entire boundary with a non-homogeneous random sinusoidal Dirichlet BC. In the most complex BC configuration, we consider component-wise BCs, i.e., independent BC types per solution component, eight boundary segments of random locations and sizes, and a random composition of four different BC types, namely, homogeneous and non-homogeneous Dirichlet and Neumann BCs. Furthermore, the elasticity and hyperelasticity problems admit complex and multi-scale features in the corresponding stress and strain fields, including singularities. In Table C.1, we summarize the main characteristics of each problem and configuration. In the subsequent subsections, the details of each dataset are presented.

*Table C.1.* Problem configurations and their characteristics. Each configuration is applied to three distinct geometries, resulting in a total of 18 datasets.

| Problem | Configuration | Main characteristics |
|---|---|---|
| Poisson | Dirichlet | Non-homogeneous Dirichlet BCs are set on the entire boundary. The BC magnitude is a random sinusoidal function. |
| Poisson | Mixed | Non-homogeneous Dirichlet, Neumann, and Robin BCs are set on four boundary segments with random sizes and locations. The composition of the BC types is chosen randomly for each sample. All BC magnitudes and coefficients are random sinusoidal functions. |
| Poisson | Mixed+ | Same BC configuration as Mixed is applied. The domain source function also varies and is sampled from a random sinusoidal distribution. |
| Linear elasticity | Steel | Component-wise Dirichlet and Neumann BCs are set on eight boundary segments with random sizes and locations. The BCs are randomly chosen for each segment among free surface (Neumann), traction force (Neumann), clamped (Dirichlet), and prescribed displacement (Dirichlet). The magnitudes of the traction force and of the prescribed displacement are random sinusoidal functions. |
| Linear elasticity | Bone | Same BC configuration as Steel is applied. |
| Hyperelasticity | Rubber | Same BC configuration as Steel is applied. Both the material model and the kinematics are non-linear (large deformations), therefore the problem is highly non-linear. |

All configurations of the Poisson problem are considered in a Circle geometry, a Square geometry, and a Boomerang geometry. For the elasticity and hyperelasticity problems, we consider more complex geometries by introducing holes in the aforementioned geometries. We denote the geometries containing holes by Circle[H], Square[H], and Boomerang[H]. Each dataset contains 8,704 realizations (samples), where 256 are reserved for testing. All results are obtained with the FEM using the Python interface of the FEniCS project (Baratta et al., 2023) with meshes generated by the Gmsh library (Geuzaine, Christophe and Remacle, Jean-Francois, 2024). The problems are run in double precision on a single CPU core. All domains are discretized using unstructured triangular meshes with local refinement near the boundaries, with mesh sizes ranging between roughly 11,700 and 17,600 nodes. We use linear shape functions for

---

[3] available at https://doi.org/10.5281/zenodo.18377370.

the Poisson problems and quadratic shape functions for the linear elasticity and hyperelasticity problems. For the hyperelasticity problems, the solutions are obtained using a finer mesh (ranging between 44,000 and 67,000 nodes), and are downsampled before storage. Notably, we only store node coordinates and the corresponding values of each field in the datasets. The spatial discretization of each geometry is illustrated in Figure C.1.

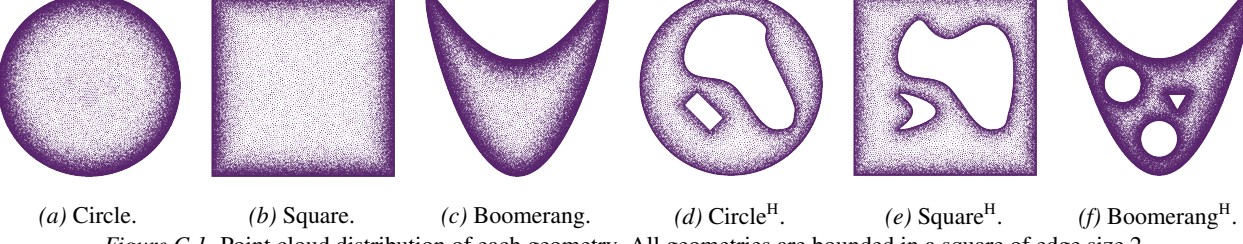

*(a)* Circle.   *(b)* Square.   *(c)* Boomerang.   *(d)* CircleH.   *(e)* SquareH.   *(f)* BoomerangH.

*Figure C.1.* Point cloud distribution of each geometry. All geometries are bounded in a square of edge size 2 centered at the origin, i.e., $\Omega \subset [-1, +1]^2$.

All BC magnitudes are randomly drawn from a parameterized distribution of the form

$$A \sin\left(\frac{\theta}{R} + \phi_0\right) \sum_{k=1}^{K} b_k \sin\left(k\frac{\theta}{R} + \phi_k\right), \quad \theta = \arctan\frac{x_2 - c_2}{x_1 - c_1}, \tag{16}$$

where $C = (c_1, c_2) \in \mathbb{R}^2$ is a constant center coordinate, $R \in \mathbb{R}$ is a constant radius, $A \in \mathbb{R}$ is a constant amplitude, $K$ is the number of modes, $b_k \in \mathbb{R}$ is the random contribution of each mode for $k = \{1, 2, \ldots, K\}$, and $\phi_k$ for $k = \{0, 1, 2, \ldots, K\}$ are phase shifts randomly drawn from a uniform distribution, $\phi_k \sim \mathcal{U}([0, 2\pi))$. The mode contributions are uniformly sampled such that $\sum_{k=1}^{K} b_k = 1$.

### C.1. The Poisson problem

Using the same notation introduced in Section 2 of the main text, the Poisson equation in a domain $\Omega \subset \mathbb{R}^2$ reads

$$\begin{aligned} -\Delta u &= f & x &\in \Omega \\ u &= \gamma_D & x &\in \partial\Omega_D \\ \nabla_{\hat{n}} u &= \gamma_N & x &\in \partial\Omega_N \\ \alpha_R u + \nabla_{\hat{n}} u &= \gamma_R & x &\in \partial\Omega_R \end{aligned}$$

where $u : \overline{\Omega} \to \mathbb{R}$ is the solution function, $f : \Omega \to \mathbb{R}$ is a domain source function, $\alpha_R, \gamma_D, \gamma_N, \gamma_R$ are scalar functions, and $\hat{n}$ is the outward unit vector normal to the boundary. The boundary segments are disjoint and their union covers the entire boundary; that is, $\partial\Omega = \partial\Omega_D \cup \partial\Omega_N \cup \partial\Omega_R$. Expressing the BCs in the form of (3), the boundary operators are defined as

$$\mathcal{B}_D(u) = u, \quad \mathcal{B}_N(u) = \nabla_{\hat{n}} u.$$

The inputs of the Poisson datasets are coordinates, geometric features (signed distance function), domain source function $f$, boundary segments $\partial\Omega_D, \partial\Omega_N, \partial\Omega_R$, and boundary functions $\gamma_D, \gamma_N, \gamma_R$, and $\alpha_D$. The target output is the solution field $u$.

#### C.1.1. CONFIGURATION: DIRICHLET

For each new sample, a Dirichlet BC is set on the entire boundary. The magnitude $\gamma_D$ is randomly drawn from the distribution given in (16). For the Circle and Square geometries, the following settings are used,

$$C = (0, 0), \ R = 1, \ K = 12, \ A \sim \mathcal{U}([2, 10]).$$

For the Boomerang geometry, we use the settings,

$$C = (0, -0.375), \ R = 0.625, \ K = 6, \ A \sim \mathcal{U}\left([2, 10]\right).$$

For all samples, a fixed source function $f(x) = 20 \cos(4\pi\|x\|)$ is used, where we use the 2-norm for the Circle and Boomerang geometries and the infinity norm for the Square geometry. Figure C.2 illustrates three samples of this configuration.

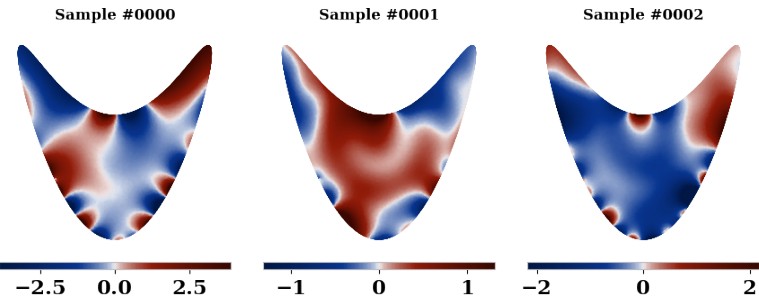

*Figure C.2.* Three samples of the solution of the Poisson problem with the Dirichlet configuration on the Boomerang geometry.

### C.1.2. CONFIGURATION: MIXED

For each new sample, the boundary is randomly partitioned into four segments. Each segment is randomly assigned to $\partial\Omega_D$, $\partial\Omega_N$, or $\partial\Omega_R$, with the constraint that at least one segment belongs to $\partial\Omega_D$ to ensure well-posedness. Next, for each boundary segment, the corresponding functions $\gamma_D$, $\gamma_N$, $\gamma_R$, and $\alpha_R$ are independently drawn from the distribution given in (16). The settings of the random distributions for the Circle and Square geometries are

$$\begin{cases} \gamma_D: & C = (0,0), \ R = 1, \ K = 8, \ A \sim \mathcal{U}\left([1, 4]\right), \\ \gamma_N: & C = (0,0), \ R = 1, \ K = 6, \ A \sim \mathcal{U}\left([2, 10]\right), \\ \gamma_R: & C = (0,0), \ R = 1, \ K = 6, \ A \sim \mathcal{U}\left([2, 10]\right), \\ \alpha_R: & C = (0,0), \ R = 1, \ K = 3, \ A \sim \mathcal{U}\left([0.2, 0.6]\right). \end{cases}$$

For the Boomerang geometry we set

$$\begin{cases} \gamma_D: & C = (0, -0.375), \ R = 0.625, \ K = 6, \ A \sim \mathcal{U}\left([1, 4]\right), \\ \gamma_N: & C = (0, -0.375), \ R = 0.625, \ K = 4, \ A \sim \mathcal{U}\left([2, 10]\right), \\ \gamma_R: & C = (0, -0.375), \ R = 0.625, \ K = 4, \ A \sim \mathcal{U}\left([2, 10]\right), \\ \alpha_R: & C = (0, -0.375), \ R = 0.625, \ K = 3, \ A \sim \mathcal{U}\left([0.2, 0.6]\right). \end{cases}$$

For all samples, a fixed source function $f(x) = 20 \cos(4\pi\|x\|)$ is used, where we use the 2-norm for the Circle and Boomerang geometries and the infinity norm for the Square geometry. Figure C.3 illustrates three samples of this configuration.

### C.1.3. CONFIGURATION: MIXED+

The random BC settings follow exactly the settings described in the Mixed configuration (Appendix C.1.2). In this configuration, the domain source functions are also randomly sampled from a parameterized distribution of the form

$$f(x) = 20 \sum_{k=1}^{2} b_k \sin(2k\pi\|x - C\| + \phi_k),$$

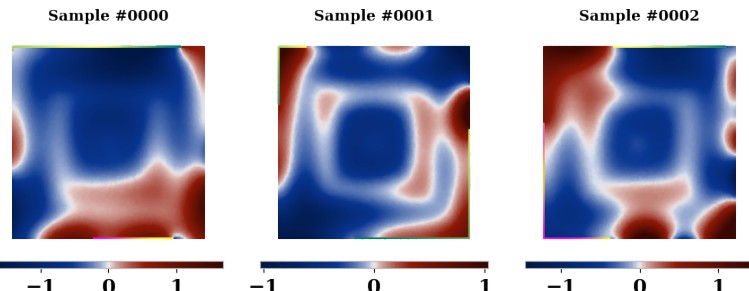

*Figure C.3.* Three samples of the solution of the Poisson problem with the Mixed configuration on the Square geometry. Neumann and Robin BCs are highlighted with pink-yellow and green-yellow color maps, respectively.

where $C \sim \mathcal{U}\left([-1, +1]^2\right)$, $\phi_k \sim \mathcal{U}\left([0, 2\pi)\right)$ for $k \in \{1, 2\}$, and $b_k$ for $k \in \{1, 2\}$ are uniformly sampled such that they sum to one. Once again, we use the 2-norm for the Circle and Boomerang geometries, and the infinity norm for the Square geometry. Figure C.4 illustrates three samples of this configuration.

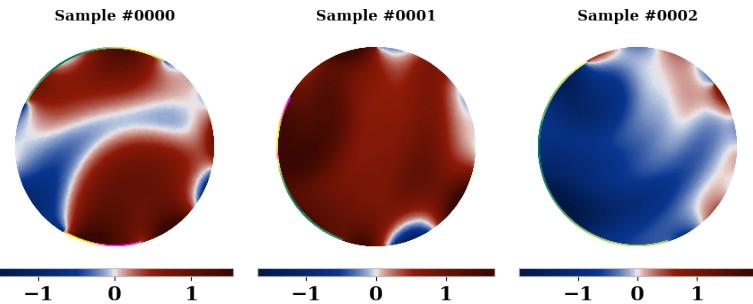

*Figure C.4.* Three samples of the solution of the Poisson problem with the Mixed+ configuration on the Circle geometry. Neumann and Robin BCs are highlighted with pink-yellow and green-yellow color maps, respectively.

### C.1.4. MESH CONVERGENCE STUDY

To verify the correctness and assess the accuracy of the obtained solutions, we have performed a mesh convergence study on all three geometries with the Dirichlet and Mixed+ configurations. The results are presented in Figure C.5. As observed in these results, all obtained solutions used in the datasets (with roughly 16,000 mesh nodes) achieve below 1% relative errors with respect to the reference mesh.

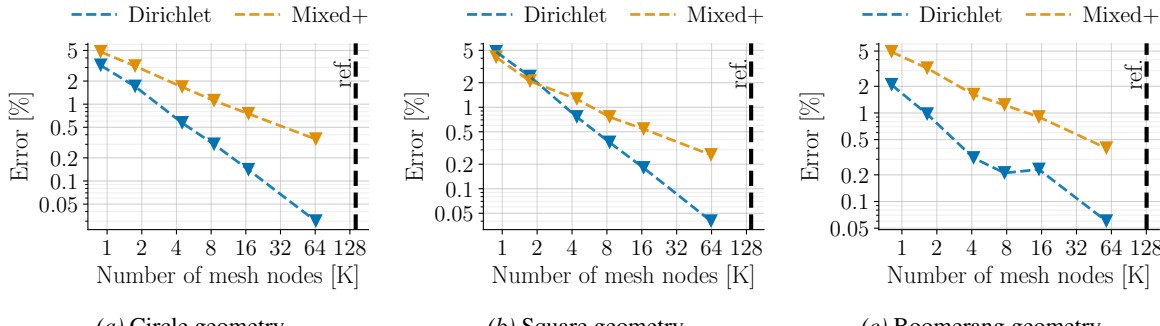

*(a)* Circle geometry.       *(b)* Square geometry.       *(c)* Boomerang geometry.

*Figure C.5.* Mesh convergence study of the Poisson problems. The experiments are repeated with 128 random samples and the median error is reported. The relative $L^2$ error is calculated with respect to the solution obtained from a very fine mesh, indicated with a dark vertical line.

## C.2. Elasticity problems

Following the notation of Section 2 of the main text, we consider the non-dimensionalized balance of linear momentum equation for a solid body $\Omega \subset \mathbb{R}^{d_x}$ as

$$\begin{cases} \nabla \cdot P + f_b = 0 & x \in \Omega, \\ u = \gamma_D & x \in \partial\Omega_D, \\ P\hat{n} = \gamma_N & x \in \partial\Omega_N, \end{cases}$$

where $x$ is the dimensionless reference coordinate, $u : \overline{\Omega} \to \mathbb{R}^{d_x}$ is the dimensionless displacement field, $P : \overline{\Omega} \to \mathbb{R}^{d_x \times d_x}$ is the dimensionless first Piola-Kirchhoff stress tensor, $f_b : \Omega \to \mathbb{R}^{d_x}$ is the dimensionless body force with respect to the reference configuration, $\hat{n}$ is the outward unit vector normal to the reference boundary, $\gamma_D : \partial\Omega_D \to \mathbb{R}^{d_x}$ is the dimensionless prescribed displacement, and $\gamma_N : \partial\Omega_N \to \mathbb{R}^{d_x}$ is the dimensionless traction force. The boundary segments $\partial\Omega_D$ and $\partial\Omega_N$ are disjoint and their union covers the entire boundary. In all datasets, we consider zero body forces, $f_b = 0$. We use a compressible Neo-Hookean material model for which the elastic strain energy density function $\tilde{\psi}$ reads

$$\tilde{\psi} = \frac{\tilde{\mu}}{2}\left[\text{tr}(C) - 3\right] - \tilde{\mu}\ln J + \frac{\tilde{\lambda}}{2}(\ln J)^2,$$

where $C$ is the right Cauchy-Green deformation tensor, $J = \det(F)$ is the determinant of the deformation gradient tensor, and $(\tilde{\lambda}, \tilde{\mu})$ are Lamé parameters. We perform the non-dimensionalization by considering a characteristic length $L$ and a characteristic stress $\Sigma$. The material parameters, the characteristic length and stress are listed for each configuration in Table C.2. The quantities can be retrieved in their original units by

$$\tilde{x} = Lx, \quad \tilde{u} = Lu, \quad \tilde{P} = \Sigma P, \quad \tilde{f}_b = (\Sigma/L)f_b, \quad \tilde{\gamma}_D = L\gamma_D, \quad \tilde{\gamma}_N = \Sigma\gamma_N, \tilde{\psi} = \Sigma\psi.$$

*Table C.2.* Material parameters for the elasticity and hyperelasticity problems.

| Configuration | $\tilde{\lambda}$ [GPa] | $\tilde{\mu}$ [GPa] | $\Sigma$ [MPa] | $L$ [m] |
|---|---|---|---|---|
| Steel | 115 | 76.9 | 100 | 1 |
| Bone | 5.77 | 3.85 | 10 | 1 |
| Rubber | 1 | 0.01 | 50 | 1 |

In the formulation above, for a given $x \in \partial\Omega$, either Dirichlet or Neumann BCs are applied to both components of $u$. A more general formulation decouples the BCs of the two solution components, resulting in potentially different BC types for each component of $u$ for a given $x$ at the boundary. In this case, the BCs are expressed as

$$\begin{cases} u_i = \gamma_D^{(i)} & x \in \partial\Omega_D^{(i)}, \\ (P\hat{n})_i = \gamma_N^{(i)} & x \in \partial\Omega_N^{(i)}, \end{cases}$$

for $i \in \{1, \ldots, d_x\}$, where $\gamma_D^{(i)} : \partial\Omega_D^{(i)} \to \mathbb{R}$ and $\gamma_N^{(i)} : \partial\Omega_N^{(i)} \to \mathbb{R}$ are scalar functions. Here, the boundary segments are only disjoint per component and $\partial\Omega = \partial\Omega_D^{(i)} \cup \partial\Omega_N^{(i)}$. These BCs can be expressed in the form of (3) by defining $\mathcal{B}_D^{(i)}(u) = u_i$ and $\mathcal{B}_N^{(i)}(u) = (P\hat{n})_i$ for each $i$. Note that the first Piola-Kirchhoff stress tensor itself is a function of the gradient of $u$.

The inputs of the elasticity and hyperelasticity datasets are coordinates, geometric features (signed distance function), boundary segments $\partial\Omega_D^{(i)}$, $\partial\Omega_N^{(i)}$, and boundary functions $\gamma_D^{(i)}$, $\gamma_N^{(i)}$. The target outputs are the displacement field $u$, independent components of the Green-Lagrange strain field, and independent components of the Cauchy stress field. A few samples of these datasets are illustrated in Figures C.6-C.8.

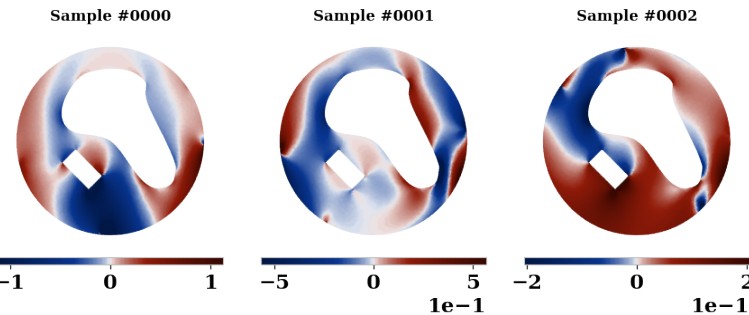

*Figure C.6.* Three samples of the dimensionless vertical normal stress of the elasticity problem (Steel) on the Circle^H geometry.

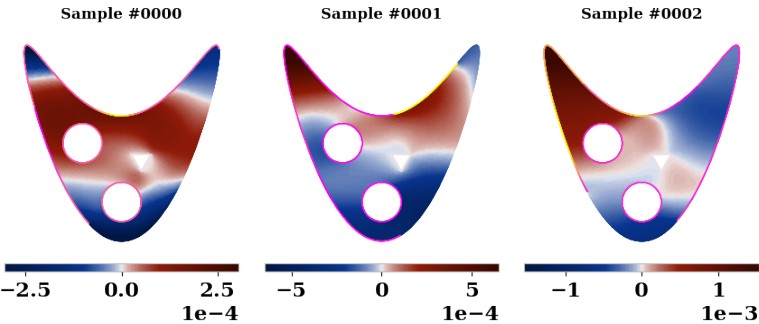

*Figure C.7.* Three samples of the dimensionless horizontal displacement of the elasticity problem (Bone) on the Boomerang^H geometry. Neumann BCs are highlighted with pink-yellow color maps.

### C.2.1. BC CONFIGURATION

A zero-displacement Dirichlet BC (clamped) is applied on the boundary of the smallest hole. The boundary of the other hole(s) is (are) set as free surface(s) (homogeneous Neumann BC). For each new sample, the exterior boundary is randomly partitioned into four segments. One segment is randomly chosen as free surface (homogeneous Neumann BC), a random prescribed displacement (Dirichlet BC) is applied to another random segment, and a random traction force (Neumann BC) is applied to the remaining two segments. This procedure (including the random partitioning) is repeated independently for each component. Hence, the resulting segments can overlap with each other and have different per-component BC types. Finally, random displacements and random traction forces are drawn from the distribution presented in (16) with $C = (0, 0)$, $K = 2$, $R = 1$, and amplitude $A$ in a given range. For the prescribed displacements, an amplitude of $\mathcal{U}\left([0.1, 0.5]\right)$, $\mathcal{U}\left([0.4, 2.0]\right)$, and $\mathcal{U}\left([100, 400]\right)$ (all in mm) is considered for the Steel, Bone, and Rubber configurations, respectively. The traction force

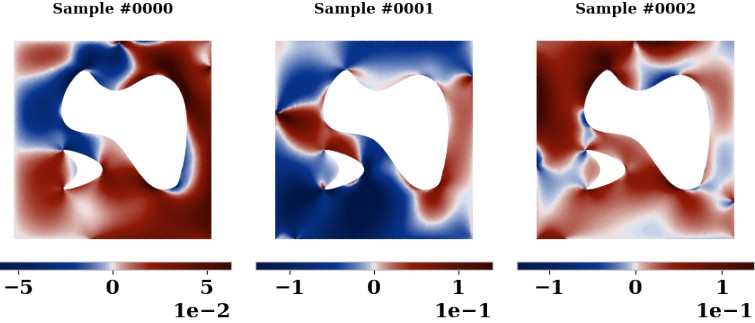

*Figure C.8.* Three samples of the Green-Lagrange shear strain of the hyperelasticity problem (Rubber) on the Square^H geometry.

amplitudes are $\mathcal{U}\left([4, 40]\right)$, $\mathcal{U}\left([0.4, 4.0]\right)$, and $\mathcal{U}\left([0.5, 3]\right)$ (all in MPa), respectively. Note that both traction forces and displacements are non-dimensionalized before being applied.

### C.2.2. SINGULARITIES

Given the discontinuities in the BCs and the existing sharp corners in some of the considered geometries, singular points appear in almost all samples of these datasets. In particular, singular stresses appear on the two ends of each boundary segment as well as at the sharp boundary corners. Clearly, the values obtained for these singular stresses in the discrete setting increase with decreasing mesh size. For this reason, each sample includes a few extremely high stress values and potentially NaNs (due to overflow). Given that these theoretically infinite values are not realistic, we recommend excluding the NaNs and the extremely high values of each sample both in the training and in the tests. However, the locations of the extreme values can be evaluated. See Appendices D and E for more details.

### C.2.3. MESH CONVERGENCE STUDY

As for the previous datasets, a mesh convergence study is performed for the Steel and Rubber configurations on the Circle[H] geometry. The results are presented in Figure C.9. For the Steel configuration, the mesh used for the datasets has 11,786 nodes. At this mesh resolution, a relative error of 1.1%, 3.3%, and 3.3% are observed for displacement, strain, and stress fields, respectively. For the Rubber configuration, a mesh with 44,067 nodes is used, which yields 0.3%, 1.2%, and 1.7% relative errors, respectively.

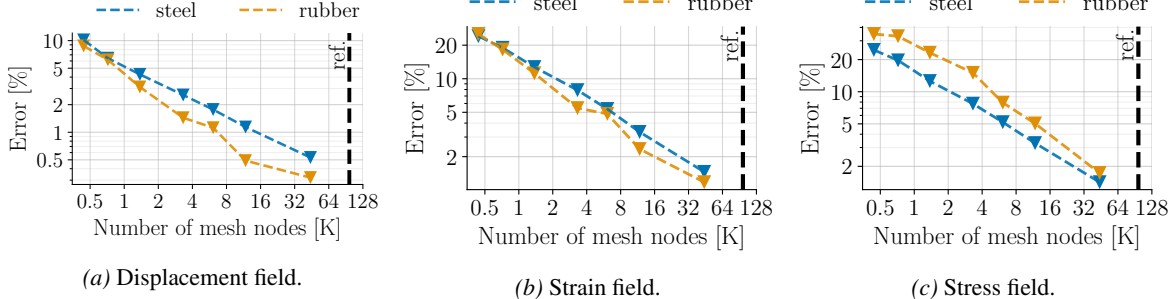

*(a)* Displacement field.

*(b)* Strain field.

*(c)* Stress field.

*Figure C.9.* Mesh convergence study of the elasticity problem on the Circle[H] geometry. The experiments are repeated with 128 (Steel) and 8 (Rubber) random samples and the median error is reported. The relative $L^2$ error is calculated with respect to the solution obtained from a very fine mesh, indicated with a dark vertical line. A margin of 0.05 from the boundaries is excluded in the error calculation to exclude singularities at the boundaries. For each sample, the reported errors are average absolute values over all components of the respective field.

# D. Metrics

## D.1. Relative $L^2$ error

The relative $L^2$ error for a given scalar function $\hat{u} : \overline{\Omega} \to \mathbb{R}$ with respect to a ground truth $u : \overline{\Omega} \to \mathbb{R}$ is given as

$$\frac{\|\hat{u} - u\|_{L^2(\Omega)}}{\|u\|_{L^2(\Omega)}} \approx \frac{\sqrt{\sum_i (\hat{\boldsymbol{u}}_i - \boldsymbol{u}_i)^2}}{\sqrt{\sum_i \boldsymbol{u}_i^2}},$$

where $\hat{\boldsymbol{u}}_i$ and $\boldsymbol{u}_i$ are the discretized values of the model estimate and ground truth at spatial coordinate $i$. Now, consider a set of singularities in the subdomain $\check{\Omega} \subset \overline{\Omega}$ and their corresponding coordinate indices $\check{I}$. Excluding these singularities, we calculate the relative $L^2$ error as

$$\frac{\|\hat{u} - u\|_{L^2(\Omega \setminus \check{\Omega})}}{\|u\|_{L^2(\Omega \setminus \check{\Omega})}} \approx \frac{\sqrt{\sum_{i \notin \check{I}} (\hat{\boldsymbol{u}}_i - \boldsymbol{u}_i)^2}}{\sqrt{\sum_{i \notin \check{I}} \boldsymbol{u}_i^2}}.$$

Note that the location of the singularities varies across samples, therefore, the exclusion is applied in a per-sample fashion. In practice, we identify for each sample the singularities by finding the 0.2% values of the ground truth field that have the largest magnitude. For a point cloud of size 16,000, this corresponds to 32 singular points. Although the singularities are primarily reflected in the stress fields, for consistency we use this metric for all reported relative $L^2$ errors. After calculating the relative error for each component, the average of all components is calculated to have a single error for each sample. The final reported value is the median across all test samples.

## D.2. Chamfer distance and recall score

Although the values of the stress and strain fields at the singular points are unrealistic and should not be evaluated, their location is meaningful and carries valuable information. Thus, we identify the location of the ground-truth singularities $\mathbf{x}_s \subset \mathbf{x}$ and of those of the fields estimated by the model $\hat{\mathbf{x}}_s \subset \mathbf{x}$, and evaluate their proximity using suitable metrics. Note that both are discrete sets containing coordinates in $\mathbb{R}^{d_x}$.

The Chamfer distance estimates the average distance of two segments. Denoting the Euclidean distance between two points by $d(.,.)$, we calculate the relative Chamfer distance as

$$\text{Relative Chamfer distance} = \frac{\text{mean}_{x_s \in \mathbf{x}_s} \min_{\hat{x}_s \in \hat{\mathbf{x}}_s} d(x_s, \hat{x}_s)}{D},$$

where $D = 2\sqrt{2}$ is the diameter of the bounding box of the domain. The smaller this distance, the closer the two sets.

As a second metric, we use the recall score with a spatial tolerance. To calculate this, we consider a radius of $0.02D = 0.04\sqrt{2}$ around each coordinate in $\mathbf{x}_s$, and count the number of coordinates in $\hat{\mathbf{x}}_s$ that are near at least one of these ground-truth singularities. The recall score is then calculated by dividing this number by the total number of singular points.

# E. Details of the experiments

In this appendix, architectural details as well as settings used in most of the experiments are described. We use the same settings in all experiments unless otherwise specified. All model input and output features are normalized to zero mean and unit variance. Importantly, in the calculation of the initial means and variances, we exclude the 0.4% entries of each sample with the highest magnitudes. In presence of singularities (see Appendix C.2.2), this step is essential for getting well-balanced features. For similar reasons, the top 0.2% (per sample) entries of each field are excluded in the calculation of the loss function. Unless otherwise stated, we use 8,192 training samples and evaluate the models on 256 test samples. Each model is trained for 2,048 epochs. A validation set (256 samples) is used to assess the model's accuracy, and a checkpoint is made if the validation error improves. We use batch size 8, the mean relative $L^2$ error (see Appendix D) as loss function, the AdamW (Loshchilov & Hutter, 2019) optimizer with weight decay $10^{-4}$, adaptive gradient clipping (Brock et al., 2021) with ratio 0.5, and cosine learning rate decay with linear warm-up period. The learning rate is warmed up in the first 5% training epochs from $10^{-5}$ to $2 \times 10^{-4}$, and decays with cosine schedule to $10^{-5}$ in the next 95% of the epochs. All trainings are conducted using 2 NVIDIA GeForce RTX 3090 GPUs (24GB), except for the BENO benchmarks where we have used a single GPU for training.

Although hyperparameter tuning is often central to training ML models, a robust OL framework should not require extensive tuning when applied to new problems. For this reason, we identify a single setting for each module that performs reliably across all datasets and use it consistently in all experiments without further adjustment. Additional details on each module are provided in the subsections below.

## E.1. Learned extender

The learned extender follows a simple design as illustrated in Figure1. Each FF block consists of a multi-layer perceptron (MLP) with one hidden layer and the Swish (Ramachandran et al., 2017) activation function. We use a latent dimension of 128 for all intermediate outputs and all hidden layers. Each FF or CA block is followed by a LayerNorm (Ba et al., 2016) normalization, except for the final FF block. We use 6 CA blocks in the extender, and use 4 attention heads in each one. Each attention head uses a latent size of 128. The final layer outputs pseudo-extensions with dimension 16.

In the experiments of Appendix F.7, a masking mechanism is used in the CA blocks. The masking procedure starts with assigning a random binary mask $\boldsymbol{m}_j$ to each source feature (boundary nodes). Let us consider $\mathcal{J}$ as the set of all source (boundary) features, and $\mathcal{J}_m$ as the subset of source features that are not masked in a particular CA block. As for regular CA, key and value vectors are computed from the source (boundary) features using a single-layer MLP, and query vectors are computed from the target (domain) features. For each pair of target $i$ and source $j$ features, a similarity score $\boldsymbol{s}_{ij}$ is defined as the inner product of their corresponding query and key values. The involvement of the masking is in applying a softmax normalization. The masked source features must have no effect on the outputs. Therefore, they are excluded in the denominator. We calculate a weight for each unmasked target-source pair as

$$\boldsymbol{w}_{ij} = \boldsymbol{m}_j \operatorname*{softmax}_{j \in \mathcal{J}_m}(\boldsymbol{s}_{ij}) = \boldsymbol{m}_j \frac{\exp(\boldsymbol{s}_{ij})}{\sum_{j \in \mathcal{J}_m} \exp(\boldsymbol{s}_{ij})} = \boldsymbol{m}_j \frac{\exp(\boldsymbol{m}_j \boldsymbol{s}_{ij})}{\sum_j \exp(\boldsymbol{m}_j \boldsymbol{s}_{ij})} = \boldsymbol{m}_j \operatorname*{softmax}_{j \in \mathcal{J}}(\boldsymbol{m}_j \boldsymbol{s}_{ij}).$$

The output features are then computed as a weighted sum of the value vectors, using $\boldsymbol{w}_{ij}$ as weights. Note that with $\boldsymbol{m}_j = 1$ for all $j$ we obtain the regular softmax normalization.

## E.2. RIGNO

We follow the settings of RIGNO-12 listed in Appendix E of (Mousavi et al., 2025), and make the following adjustments to make the architecture more efficient for elliptic problems. The graph-building algorithm is improved for non-convex geometries by avoiding spurious edges that lie outside of the domain. We use a subsampling factor of 16 to obtain the coordinates of the regional nodes from the

physical nodes. In the hierarchical downsampling of the regional nodes, we use a subsampling factor of 1.2 and continue downsampling until the nodes are saturated. This ensures more multi-scale edges and helps propagate information in fewer passes. Lastly, we use an overlap factor of 1.5 in the decoder graph. This setting results in a RIGNO with roughly 1.8M parameters.

### E.3. GAOT

A simple GAOT (Wen et al., 2025) model is implemented in JAX with some modifications. The graph building of the encoder and decoder follows the same settings as RIGNO, using adaptive radii, in contrast to the multiple fixed radii used in the original implementation. We use no geometry embedding, since the geometry is fixed in every dataset considered in this work. The cosine similarity in the attention mechanism is replaced with dot product. Lastly, we use a slightly different integral operator in the encoder and decoder, where the inputs of both the attention-based weights and the integration kernel are latent node embeddings. We use a grid resolution of $32^2$, patch size of 2, four processor blocks, four attention heads, one hidden layer in each MLP, a latent size of 128, and a hidden size of 128. This setting results in a GAOT model with roughly 2.1M parameters.

### E.4. BENO

Although the BENO (Wang et al., 2024) framework, in theory, does not require any structure on the given coordinates, the implementation assumes gridded data. Therefore, the original implementation is slightly altered to be applicable to our unstructured datasets. In the original implementation, gradients of the input domain functions, calculated via finite differences, and a smoothened version of them, obtained by applying a blur kernel, are concatenated to the inputs. Both procedures assume a grid structure and are therefore removed in the benchmarked version. Given that most of our datasets are not sensitive to input domain functions, these changes should not have any significant impact on the obtained results. The distance of the coordinates from the closest boundary is given as input to all models to be consistent with the rest of our trained models. Furthermore, we apply the proposed merging and normalization procedure (see Appendix B) and use the resulting functions $\alpha$, $\beta$, and $\gamma$ as boundary inputs to BENO.

The original implementation of BENO does not support batch training or training with multiple GPUs. In order to train the model more efficiently, we extend the original implementation to support batch training and use a batch size of 2. Given that training on multiple GPUs is not supported and the spatial resolution of our datasets is roughly 16 times larger than the ones considered in (Wang et al., 2024), we adapt the model hyperparameters to obtain the highest accuracy while fitting in the memory of a single NVIDIA RTX 3090 GPU (24GiB). To this end, we use a single hidden layer in the MLPs and a latent size of 64 for node features, edge features, and boundary embedding. In our experience, these hyperparameters have minor effect on the achieved accuracy on our datasets. On the other hand, we use double the recommended transformer layers (6) to give more capacity to the model for handling complex BCs. Despite our efforts, the trainings remain extremely slow. We train each model up to 120 hours, which allows it to complete roughly 200 epochs.

We use the AdamW (Loshchilov & Hutter, 2019) optimizer with weight decay $5 \times 10^{-4}$ and a cosine annealing learning rate scheduler with warm restarts using the recommended settings. We experienced a much faster convergence and significantly lower test errors with the starting learning rate $2 \times 10^{-4}$ and therefore have used this tuned learning rate. We experienced unstable trainings with abrupt jumps, and therefore always report results that correspond to the best-performing checkpoint in terms of validation error. The normalization scheme and the training loss follow exactly those of the other trainings: we exclude the top 0.4% (in magnitude) values when calculating the statistics, and use the mean relative $L^2$ norm as the loss function, excluding the top 0.2% values in each sample.

# F. Further results

In this appendix, we present additional experimental results that either complement the discussions in the main text or evaluate other aspects of the proposed framework.

## F.1. Detailed errors for elasticity and hyperelasticity datasets

In Figure 2b of the main text we present the average relative error of all target fields. However, not unexpectedly, the accuracy is not the same in different fields. In Table F.1 we report the errors separately for the displacement, strain, and stress fields. The displacement field is generally smoother (see Figure C.7), hence the model is more accurate in this field. As described in Appendix D, the relative $L^2$ errors exclude the singularities. It is therefore important to also evaluate their location. To this end, the recall score and the relative Chamfer distance of the location of the singularities are also reported. Although the singularities are completely excluded from the training, their location is detected by the models. In the worst case, the model achieves a recall score of 88.2% and a Chamfer distance of 3.56% in the stress field of the hyperelasticity problem. In Figure F.1, the location of the singularities and their corresponding recall score are illustrated for the stress field of two samples.

*Table F.1.* Detailed test errors of the LX(R) model from Figure 2b. We report the relative $L^2$ error excluding the top 0.2% (L2), the recall score with 2.0% tolerance (RT), and the relative Chamfer distance (CD) with respect to the diameter of the domain. See Appendix D for the details of each metric. All reported values are the median over 256 samples of the test set.

| | | Median relative test errors [%] | | | | | | | | |
| | | Displacement | | | Strain | | | Stress | | |
| **Material** | **Geometry** | L2 | RT | CD | L2 | RT | CD | L2 | RT | CD |
| Steel | Circle[H] | 7.56 | 100 | 0.39 | 9.49 | 97.2 | 1.12 | 9.63 | 97.2 | 1.19 |
| Steel | Square[H] | 8.10 | 100 | 0.27 | 10.6 | 96.8 | 1.16 | 10.6 | 96.8 | 1.18 |
| Steel | Boomerang[H] | 6.38 | 100 | 0.06 | 9.87 | 96.3 | 0.77 | 9.81 | 96.3 | 1.03 |
| Bone | Circle[H] | 8.26 | 93.8 | 0.60 | 10.3 | 95.8 | 1.13 | 10.6 | 97.2 | 1.02 |
| Bone | Square[H] | 7.84 | 100 | 0.29 | 10.9 | 95.7 | 1.26 | 10.8 | 96.8 | 1.07 |
| Bone | Boomerang[H] | 6.78 | 100 | 0.08 | 10.6 | 97.2 | 0.78 | 10.7 | 96.3 | 0.91 |
| Rubber | Circle[H] | 9.29 | 97.9 | 0.41 | 13.8 | 93.1 | 1.76 | 14.1 | 88.9 | 2.91 |
| Rubber | Square[H] | 10.9 | 93.6 | 0.53 | 15.5 | 90.3 | 2.84 | 15.8 | 88.2 | 3.56 |
| Rubber | Boomerang[H] | 7.78 | 100 | 0.21 | 13.8 | 94.4 | 1.68 | 14.0 | 89.8 | 2.27 |

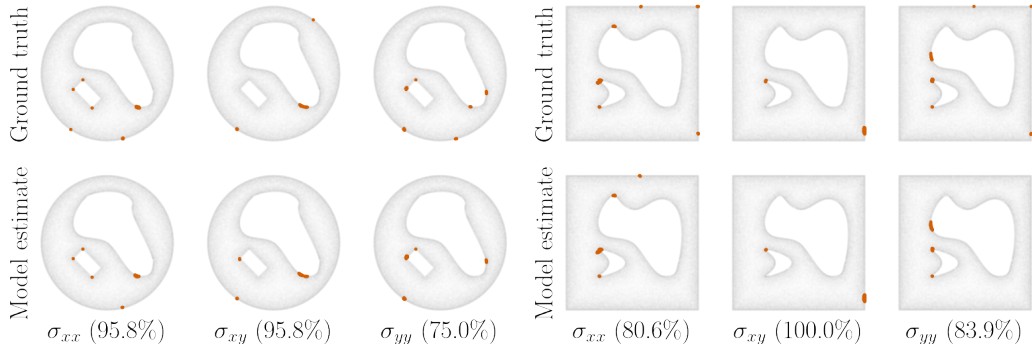

*Figure F.1.* Location of the 0.2% largest stress values (singularities) for two test samples of the hyperelastic problems. The recall score (with 2% tolerance) is reported for each stress component.

## F.2. Scaling with dataset size

Figure F.2 reports the results shown in Figure 3 of the main text, together with the corresponding training errors. For all datasets except the Poisson problem with Dirichlet BCs, the observed decrease in test error can be primarily attributed to a reduction in the generalization gap. In contrast, for Dirichlet BCs the generalization gap is already very small with as few as 1,024 training samples, indicating that this task is comparatively easier than the others. As additional samples are introduced, the model gradually saturates its representational capacity, with the error plateauing at approximately 0.2%. Achieving lower errors would therefore require increasing the model capacity.

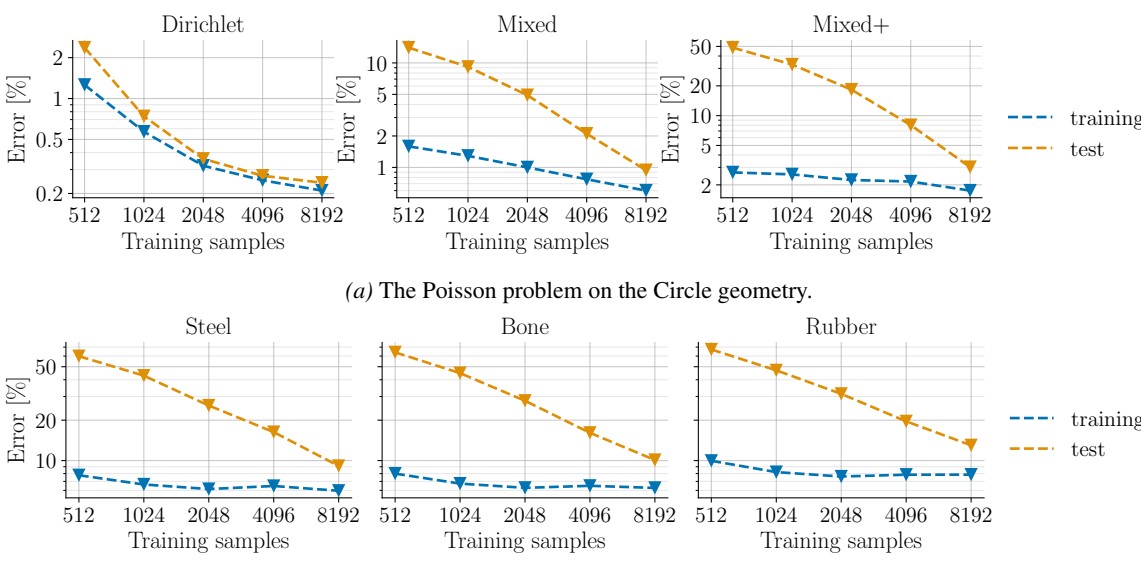

*(a)* The Poisson problem on the Circle geometry.

*(b)* The elasticity and hyperelasticity problems on the Circle$^{\mathrm{H}}$ geometry.

*Figure F.2.* Scaling of model accuracy with the size of the training dataset. Median relative $L^2$ training and test errors are reported. These plots are complementary to Figure 3 of the main text.

## F.3. Scaling with the model size

Adding a learnable extender module increases the total number of trainable parameters and therefore the overall model size. For a fair comparison, the model capacity must be matched across configurations. To this end, we scale up the core architecture in the absence of extenders and compare its performance against that of the corresponding extended models with a matched number of parameters. The results of this comparison are shown in Figure F.3.

Without an extender, the RIGNO model can reach test errors below 10% when using 24 or more processor blocks. However, increasing the number of processor blocks is computationally expensive, particularly for large-scale problems with fine discretizations. With only 12 processor blocks, the test error remains close to 40%, indicating that RIGNO's message-passing mechanism is inefficient at capturing BC dependencies when relying on zero extensions. In contrast, when a learned extender is introduced, RIGNO achieves test errors below 10% using only two processor blocks, while matching the total number of trainable parameters of the deeper baseline model. Applying the same scaling procedure to GAOT reveals a markedly different behavior. Owing to its transformer-based processor, GAOT attains a test error of 9.7% with only four processor blocks, which constitutes a moderate model size. Increasing the number of processor blocks beyond this point does not yield further improvements, and the performance saturates at this accuracy. In contrast, augmenting GAOT with a learned extender leads to a further reduction in test error, reaching 7.7%.

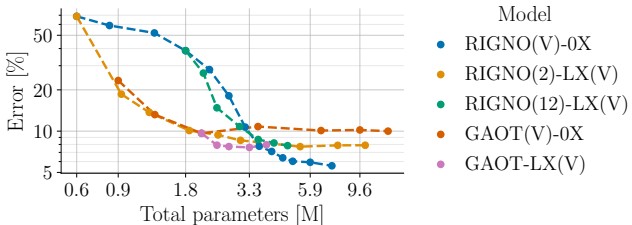

*Figure F.3.* Relative $L^2$ test error with models of different sizes. We scale the core by increasing the number of its processor blocks, and the learned extender is scaled by using more CA blocks. Only one of the modules is extended in each line. This module is indicated by a (V) in the legends. All results are based on the elasticity problem (Steel) on the Circle$^H$ geometry.

## F.4. Transfer learning

In Figure 5 of the main text, we presented transfer learning results for the Poisson problem with Mixed BCs on the Circle geometry, demonstrating that knowledge can be transferred both across geometries and to previously unseen BC types. Here, we extend this analysis to three additional target datasets. In each case, we use random initialization as a baseline and fine-tune models pre-trained on 8,192 samples from a different problem setup. To highlight the gains in data efficiency, fine-tuning is performed using varying numbers of training samples from the target setup. The corresponding results are shown in Figure F.4.

As shown in Figure F.4a, a model pre-trained on the Mixed BC configuration requires only eight training samples from the Dirichlet configuration to achieve test errors below 2%. Compared to a randomly initialized model, this corresponds to an improvement of more than 64x in data efficiency. This result is particularly encouraging, as it closely aligns with the foundation-model paradigm, where pre-training is performed on a broader and more diverse setting than the downstream task.

In Figure F.4b, we consider the elasticity problem with the Steel material as the target setup. As in previous experiments, the model benefits from pre-training on a different geometry. More notably, it benefits even more from pre-training on a different material (Rubber), despite the fact that Rubber is fundamentally different from Steel and exhibits large-deformation behavior. With only 512 training samples, the pre-trained model achieves a test error of 17.7%, whereas the randomly initialized model reaches only 59.9% test error and requires 4,096 samples to achieve an error below 17.7%. We also perform the reverse experiment, pre-training on Steel and fine-tuning on Rubber, as shown in Figure F.4c. In this case as well, pre-training yields substantial gains: with 512 training samples, the test error is reduced from 67.2% to 23.4%.

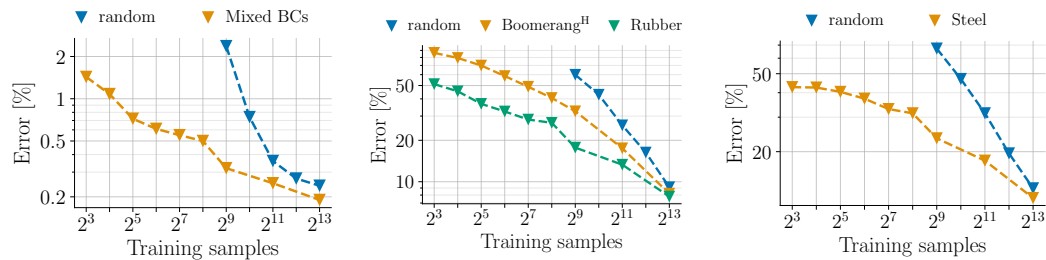

*(a)* The Poisson problem (Dirichlet) on the Circle geometry.

*(b)* The elasticity (Steel) problem on the Circle$^H$ geometry.

*(c)* The hyperelasticity (Rubber) problem on the Circle$^H$ geometry.

*Figure F.4.* Results of transfer learning experiments on multiple target datasets. Median relative $L^2$ errors on 256 test samples of the target setup are reported. The target datasets are shown in the caption of each sub-figure, and the base datasets are indicated in the legends.

## F.5. Training randomness

Training ML models is sensitive to random processes such as masking mechanisms or random initializations. Repeating the same training with a different random seed often results in fluctuations in the final accuracy. To show the level of uncertainty in the presented results, we pick four representative datasets and train each of them 10 times from scratch, each time using a different random seed. The minimum, maximum, mean, and standard deviation of the observed test errors are reported in Table F.2. Although fluctuations exist, the standard deviations are relatively low, indicating that the observed trends are statistically significant.

*Table F.2.* Median relative $L^2$ test error [%] on 256 test samples of a few datasets. All trainings are done using 8,192 training and 256 validation samples. Each training is repeated 10 times with different random seeds. The reported statistics are across the 10 repetitions.

| | | Relative $L^2$ test error [%] | | | |
|---|---|---|---|---|---|
| **Problem** | **Geometry** | **Min.** | **Mean** | **Std.** | **Max.** |
| Poisson (Dirichlet) | Circle | 0.16 | 0.24 | 0.04 | 0.32 |
| Poisson (Mixed) | Circle | 0.84 | 0.94 | 0.04 | 1.01 |
| Elasticity (Steel) | Circle[H] | 8.64 | 9.10 | 0.18 | 9.38 |
| Elasticity (Rubber) | Circle[H] | 12.7 | 13.1 | 0.31 | 13.9 |

## F.6. Ablations

To justify the design choices in the learned extender architecture, we perform ablation studies by training variants of the model from scratch. Each ablation is evaluated on four datasets, and the results are summarized in Table F.3. Replacing multi-head attention with single-head attention leads to a significant performance drop for the elasticity and hyperelasticity problems, particularly for the Rubber configuration, while the Poisson datasets remain largely unaffected. This is not surprising, as the Poisson problem is simpler and does not require complex attention mechanisms. Adjusting the number of extensions, using either a single extension or many, slightly degrades performance on two datasets, suggesting that a moderate size of 16 is a reasonable choice. The initial FF layers are critical, as their removal substantially harms performance. Intermediate and final FF layers generally provide gains, though they are less crucial than the initial layers. Finally, removing the residual connections degrades performance across all datasets, with the largest impact observed for the most challenging configuration, the hyperelasticity problem.

*Table F.3.* Median relative $L^2$ test error [%] on 256 test samples of a few datasets. All trainings are done using 8,192 training and 256 validation. The ablations are carried out on Poisson datasets on the Circle geometry with Dirichlet (D) and Mixed (M) setups, and Elasticity datasets on the Circle[H] geometry with the Steel (S) and Rubber (R) materials. Using the last row as reference, blue entries indicate improvements to the test error and red entries indicate degradation.

| | Relative $L^2$ test error [%] | | | |
|---|---|---|---|---|
| | **Poisson** | | **Elasticity** | |
| **Description** | **(D)** | **(M)** | **(S)** | **(R)** |
| Single-head attentions | 0.24 | 0.97 | 14.9 | 21.3 |
| One extension | 0.23 | 1.48 | 9.27 | 13.2 |
| Many extensions | 0.23 | 0.91 | 9.22 | 13.2 |
| W/o initial FFs | 0.72 | 3.95 | 35.7 | 67.4 |
| W/o intermediate FFs | 0.23 | 0.87 | 11.6 | 15.6 |
| W/o final FF | 0.22 | 1.06 | 9.13 | 13.5 |
| W/o residual connections | 0.27 | 1.22 | 13.5 | 20.5 |
| No ablation | 0.25 | 0.91 | 9.07 | 13.2 |

### F.7. Generalization to different resolutions

When trained with a single spatial resolution, neural operators often struggle to maintain the same level of accuracy with different resolutions at inference. This is in contrast with the nature of an operator, which should accept input functions with any valid discretization as long as it carries all significant frequencies in the function (Kovachki et al., 2023; Bartolucci et al., 2023). Importantly, zero-shot super-resolution generalization allows for cheaper trainings with lower resolutions. In this context, we assessed our framework with respect to the resolution of the input boundary functions. As domain functions are not inputs to the extender, we keep the resolution of the input domain functions fixed. We trained extenders with multiple resolutions on the Poisson problem with Dirichlet BCs for the Circle geometry, and tested them with lower and higher resolutions. We use 2,048 samples for training, and 256 for testing. In these experiments, we have exceptionally used a smaller RIGNO as the core neural operator with latent size 64 and 8 processor blocks. Motivated by the proposed masking strategies in (Mousavi et al., 2025) for obtaining resolution invariance, we repeat each training with masked CA blocks with a masking ratio of 0.3. The masking procedure in described in detail in Appendix E.

The results are presented in Figure F.5. The models trained with regular CA exhibit strong dependence on the training resolution, with a slightly milder drop in the performance for higher resolutions as compared to lower resolutions. With a training resolution of 669, the error jumps from $0.68\%$ to $3.15\%$ with a resolution of 1004 and $8.37\%$ with a resolution of 286. However, with the implemented masking strategy in the CA blocks, the accuracies are much more consistent with all resolutions. With the same training resolution, we obtain $1.44\%$ and $3.11\%$ test errors with 1004 and 286 nodes at inference, respectively. The super-resolution generalization is particularly satisfactory. When trained with the low resolution of 401 boundary nodes, the test error increases from $0.92\%$ to $1.57\%$ with 2.5x more nodes at inference.

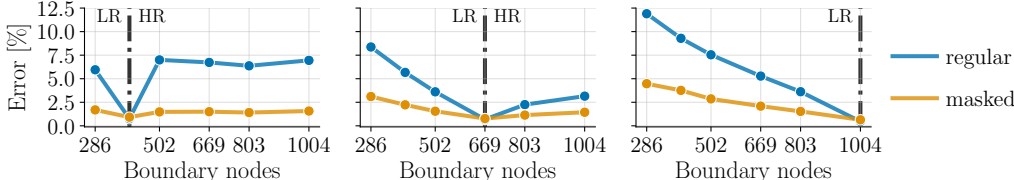

*Figure F.5.* Inference results with lower and higher resolutions than the training resolution. Median relative $L^2$ test errors on 256 test samples (Poisson problem, Dirichlet BCs, Circle) are reported. The gray vertical line indicates the training resolution. LR and HR stand for low and high resolution, respectively.

### F.8. Attention scores

In Figure F.6, we show the attention scores (averaged over multiple heads) of every domain node to every boundary node for the Poisson problem on the Circle geometry. The nodes are sorted based on the angle of their polar coordinates. This figure shows that while the very first cross-attention block is extremely focused on the direct projection of the domain node to the boundary, the receptive field gradually widens in the subsequent CA blocks. Interestingly, while the local effects persist until the last CA block for the Mixed configuration, with the Mixed+ configuration the domain nodes receive information from all boundary nodes in the last two CA blocks.

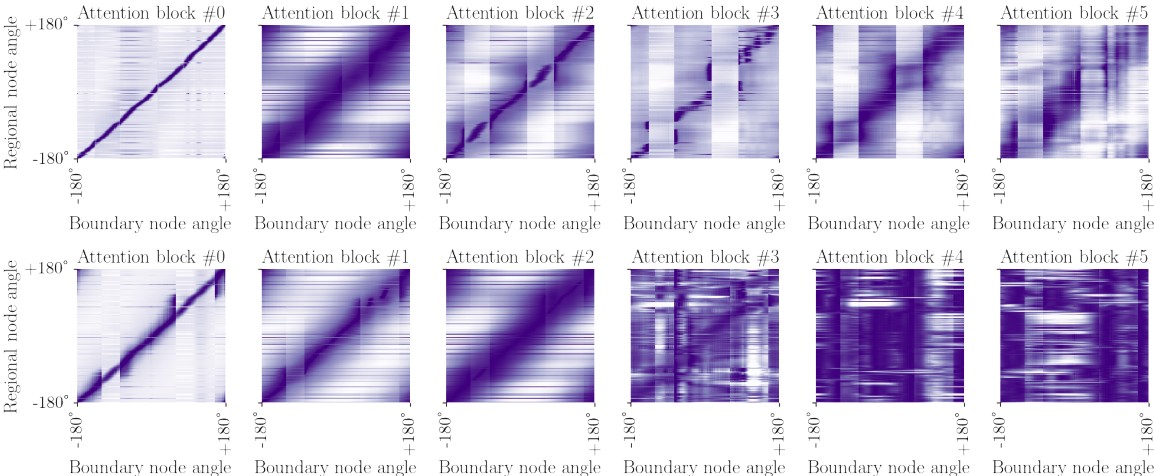

*Figure F.6.* Average attention scores in each CA block for the Poisson problem on the Circle geometry with the Mixed (top) and Mixed+ (bottom) configurations. Domain and boundary nodes are sorted based on their polar angle.

### F.9. FNO and learned extensions

To test the effectiveness of this approach with FNOs (Li et al., 2021), we have conducted a few small-scale experiments. Since FNO requires inputs and outputs on a uniform grid, we interpolated the Poisson dataset on the Square geometry dataset onto a uniform grid and used the interpolated fields for both training and tests, ignoring the introduced interpolation errors. The results, summarized in Table F.4, indicate that FNO substantially benefits from learned extenders in problems with high sensitivity to BCs. In particular, an FNO with 0.8M parameters (depth 2, width 20, 16 modes) struggles to accurately capture BC influence when zero-padding is used. Zero-padding introduces high-frequency Fourier modes, which are difficult to represent under the truncated spectral modes of FNO. By contrast, a lightweight extender with only 0.2M parameters produces smooth BC representations (see Appendix G), making them more compatible with FNO's spectral truncation. This enables FNO to better capture the dependency on BCs and achieve significantly lower errors.

*Table F.4.* Median relative $L^2$ test error [%] on 256 test samples of the Poisson problem on the Square geometry. All trainings are done using 4,096 training and 256 validation samples.

| | BC Configuration | | |
|---|---|---|---|
| Model | Dirichlet | Mixed | Mixed+ |
| FNO (0X) | 2.51 | 20.0 | 30.9 |
| FNO (LX) | 1.33 | 3.86 | 7.89 |

# G. Visualizations

In this Appendix, we present visualizations of the outputs of our trained models. All visualizations correspond to the LX(R) columns of Figure 2.

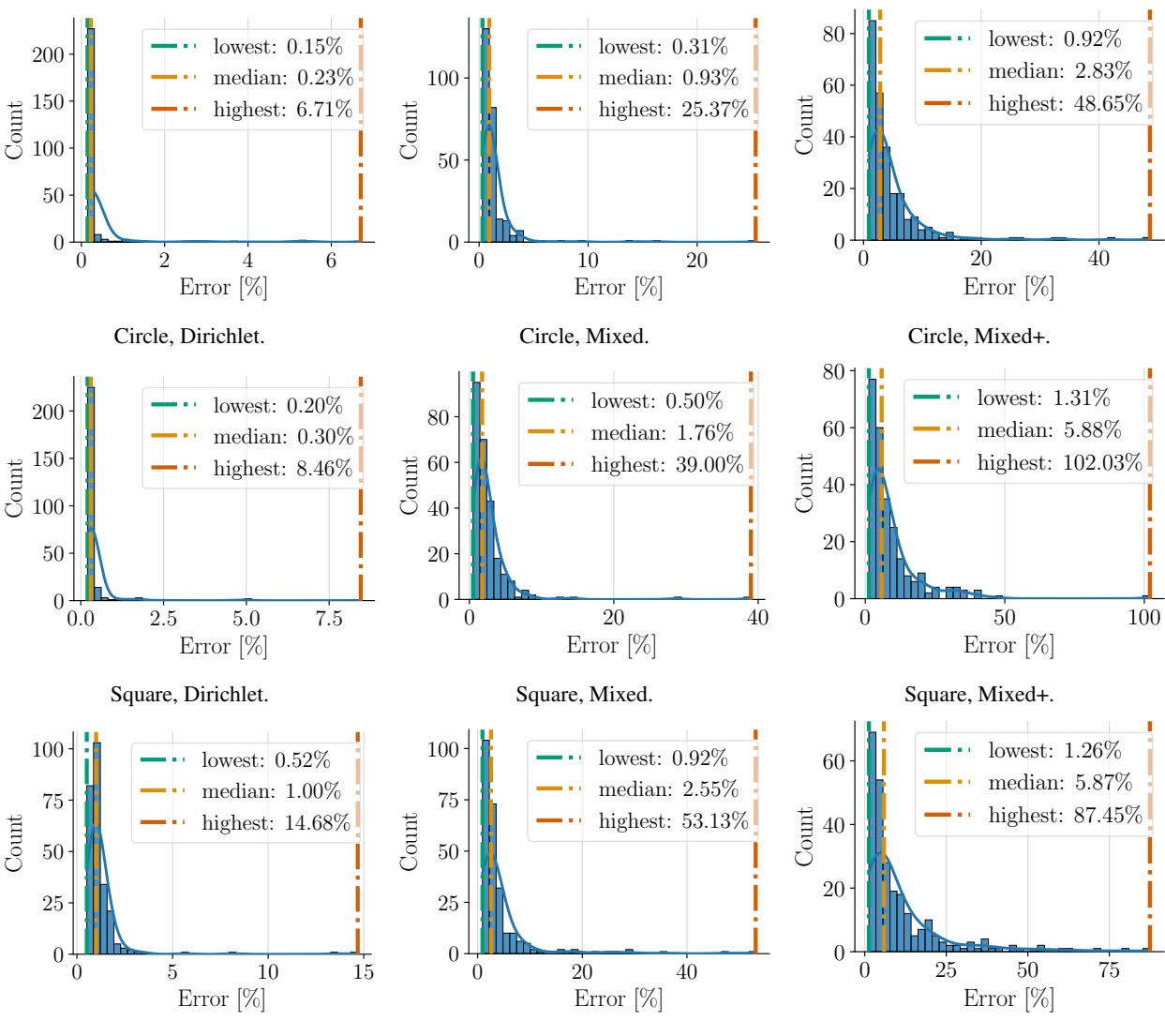

Circle, Dirichlet.

Circle, Mixed.

Circle, Mixed+.

Square, Dirichlet.

Square, Mixed.

Square, Mixed+.

Boomerang, Dirichlet.

Boomerang, Mixed.

Boomerang, Mixed+.

*Figure G.1.* Error distribution of the trained LX(R) model on the test datasets of the Poisson problem.

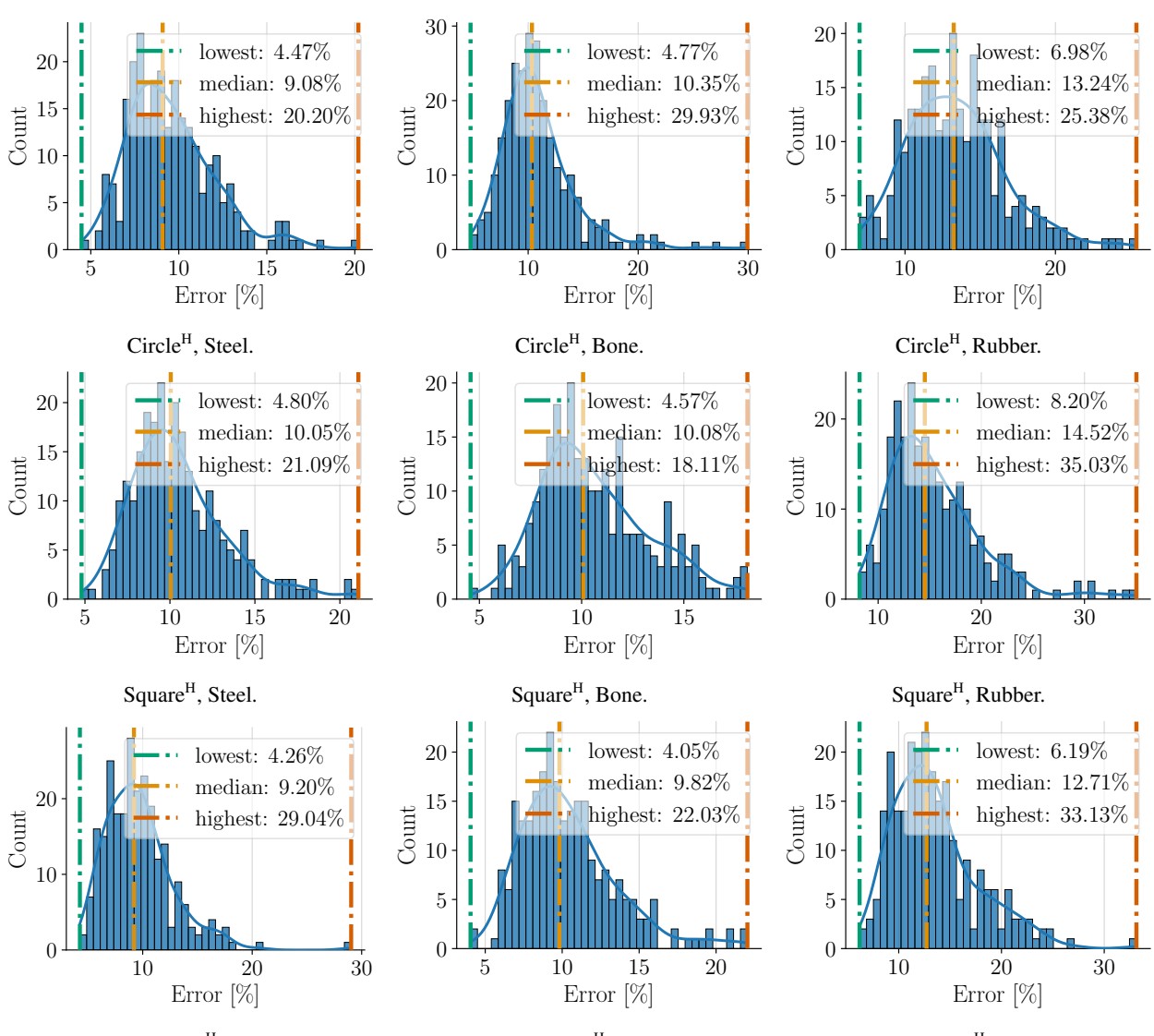

*Figure G.2.* Error distribution of the trained LX(R) model on the test datasets of the elasticity and hyperelasticity problems.

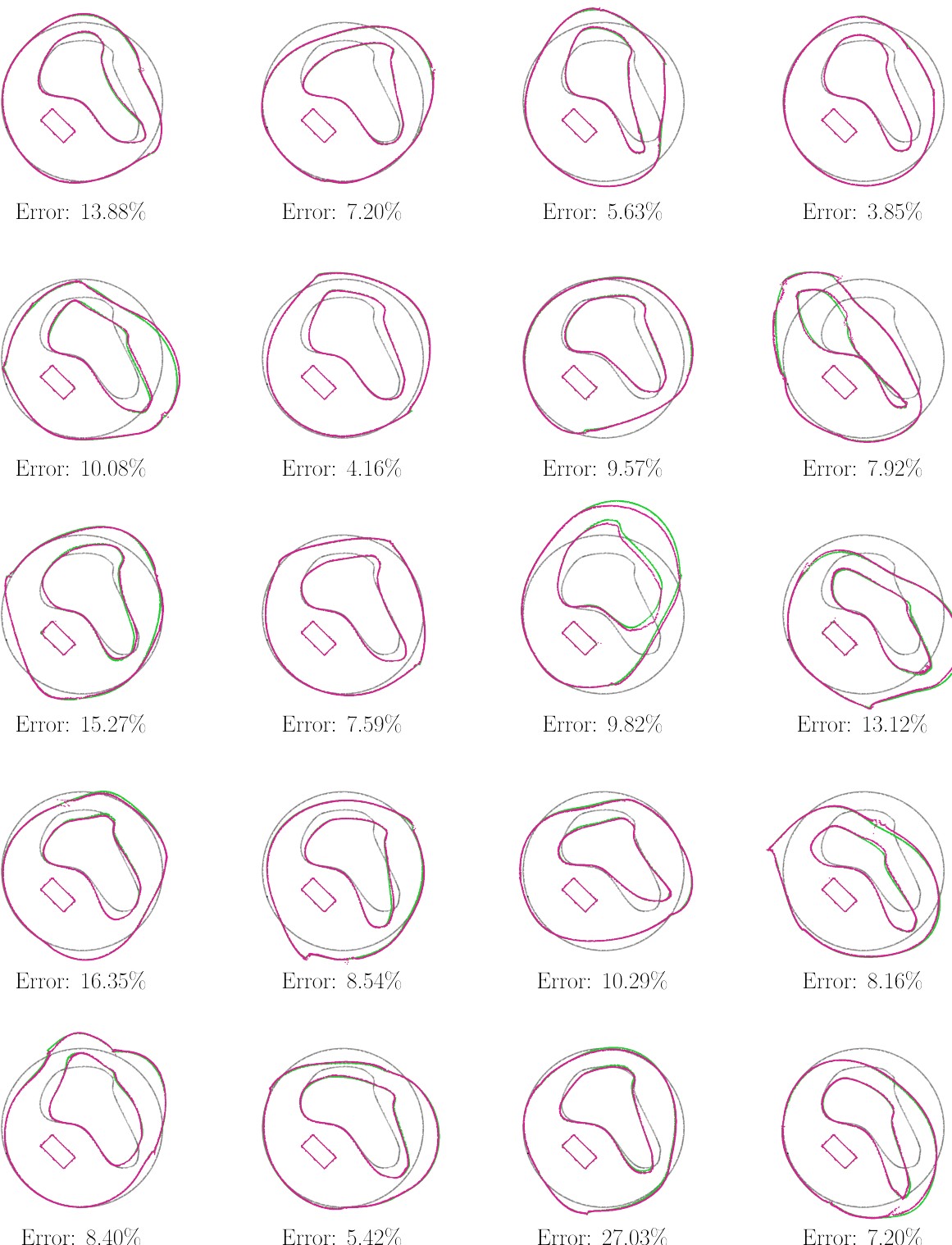

*Figure G.3.* Ground-truth (green) deformed domain and estimates of the LX(R) model (purple) for 20 test samples of the hyperelasticity problem (Rubber) on the Circle$^{\mathrm{H}}$ dataset.

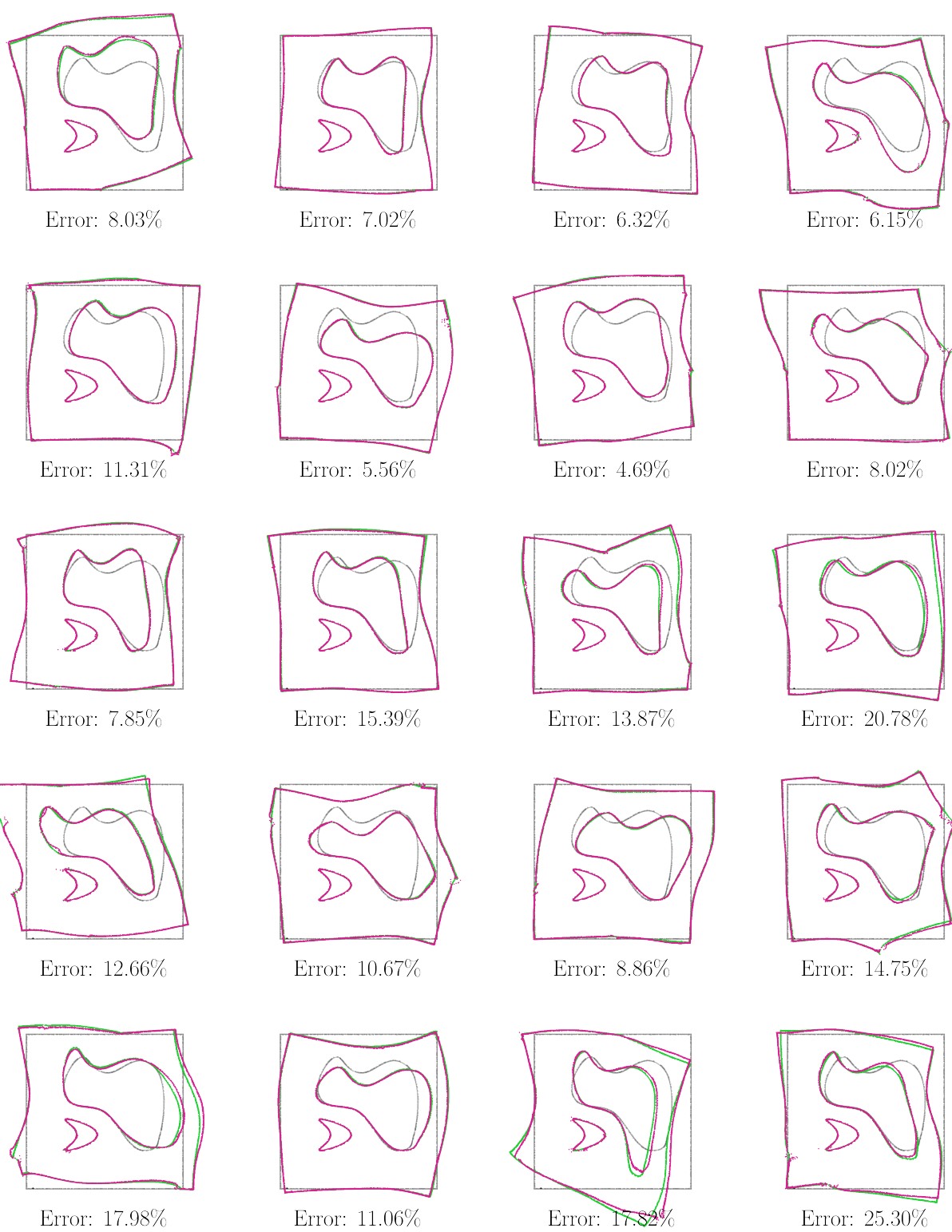

*Figure G.4.* Ground-truth (green) deformed domain and estimates of the LX(R) model (purple) for 20 test samples of the hyperelasticity problem (Rubber) on the Square[H] dataset.

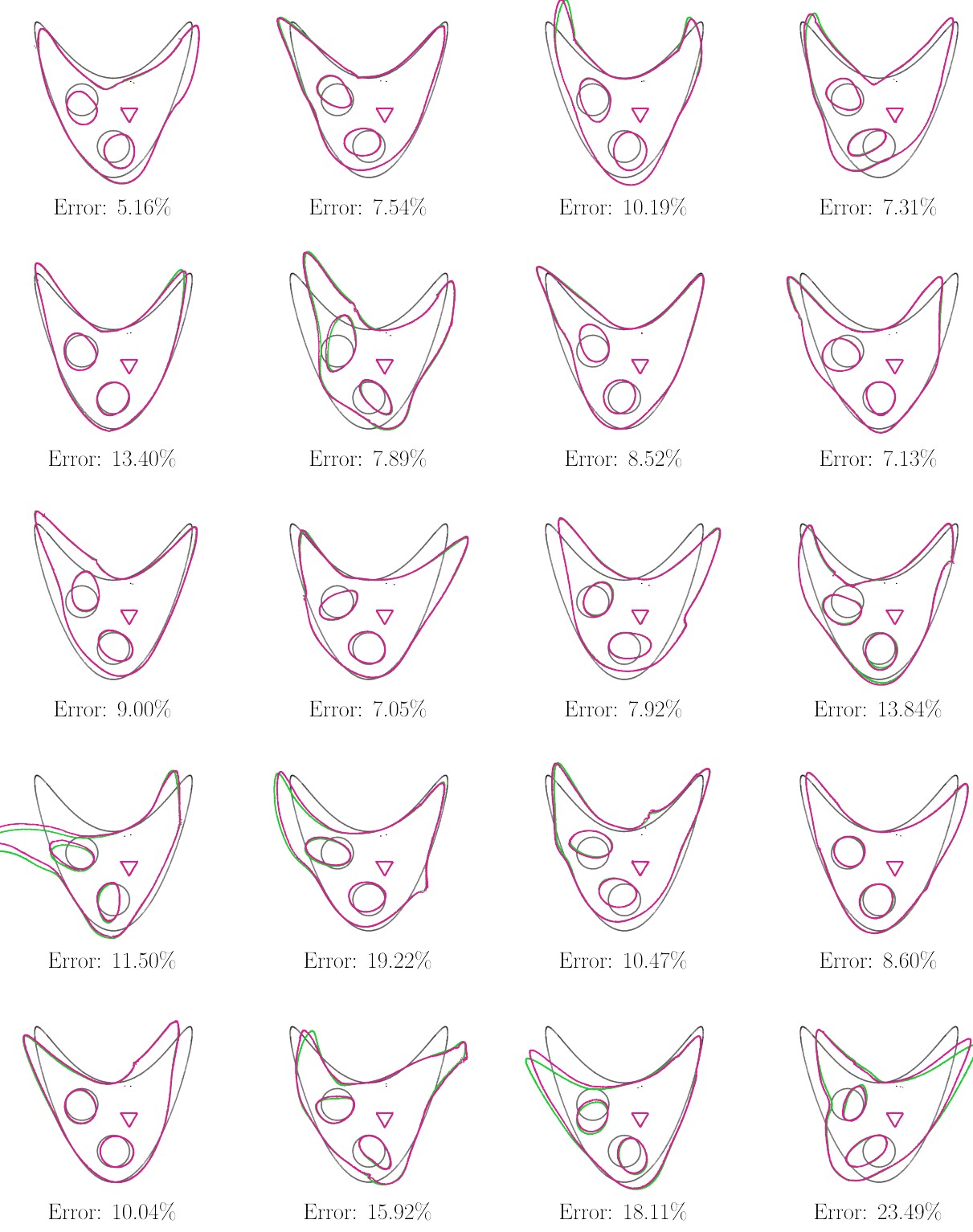

*Figure G.5.* Ground-truth (green) deformed domain and estimates of the LX(R) model (purple) for 20 test samples of the hyperelasticity problem (Rubber) on the Boomerang[H] dataset.

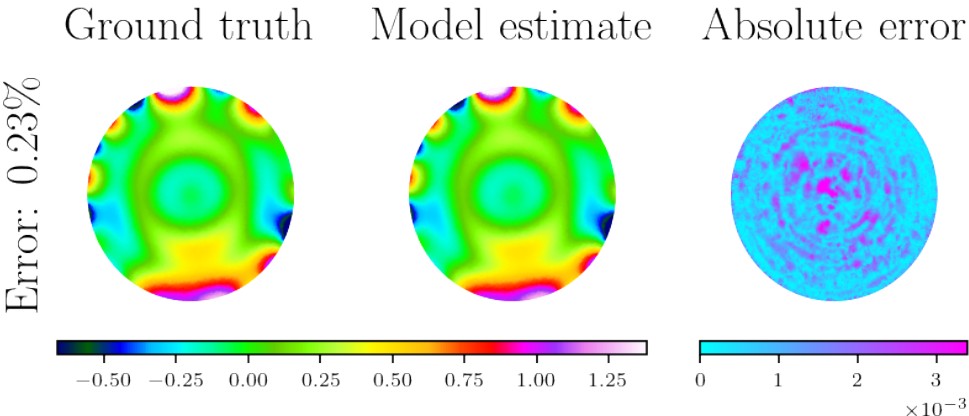

*Figure G.6.* Estimate of a trained LX(R) model for a test sample of the Poisson (Dirichlet) problem.

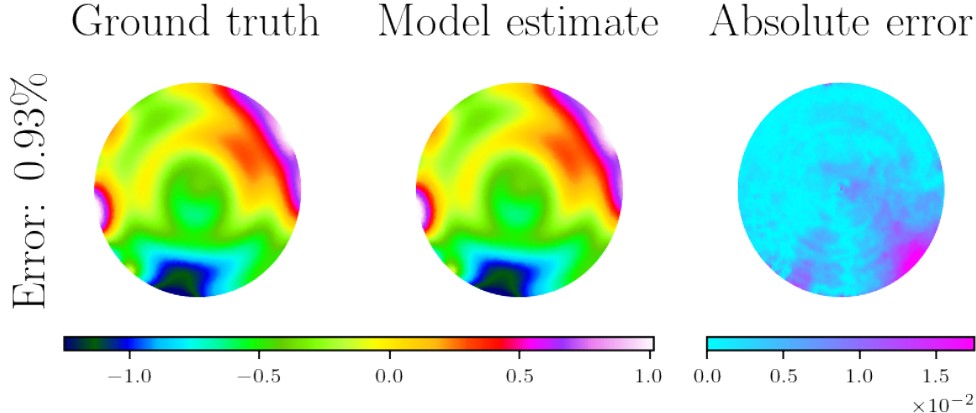

*Figure G.7.* Estimate of a trained LX(R) model for a test sample of the Poisson (Mixed) problem.

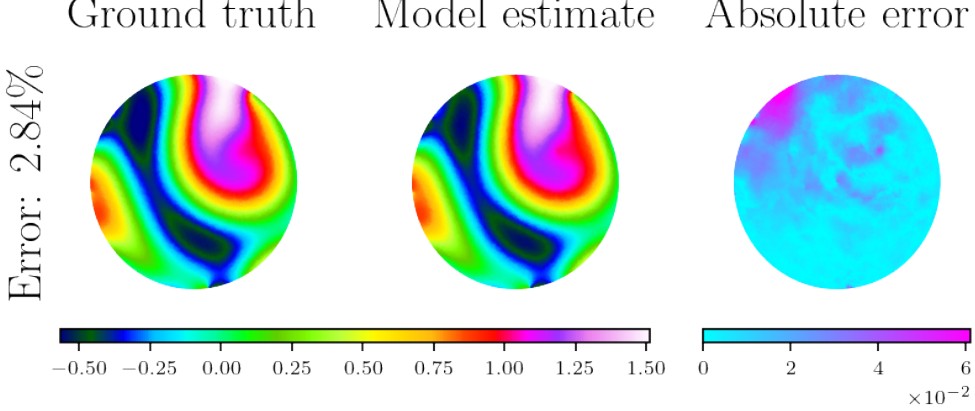

*Figure G.8.* Estimate of a trained LX(R) model for a test sample of the Poisson (Mixed+) problem..

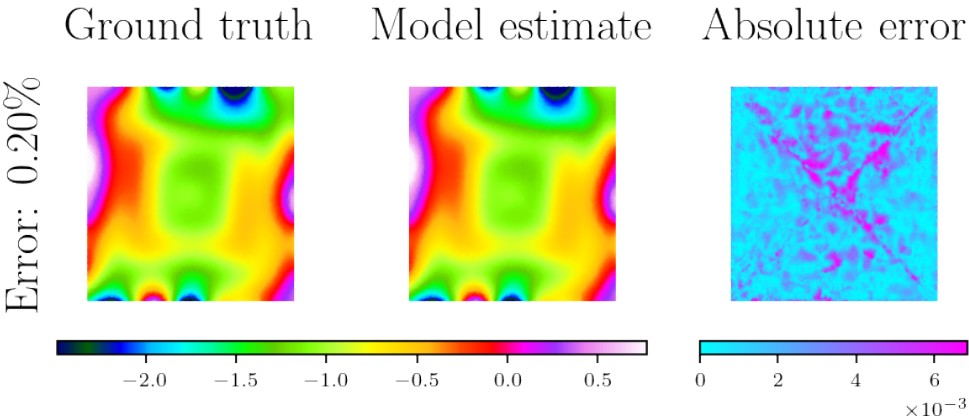

*Figure G.9.* Estimate of a trained LX(R) model for a test sample of the Poisson (Dirichlet) problem.

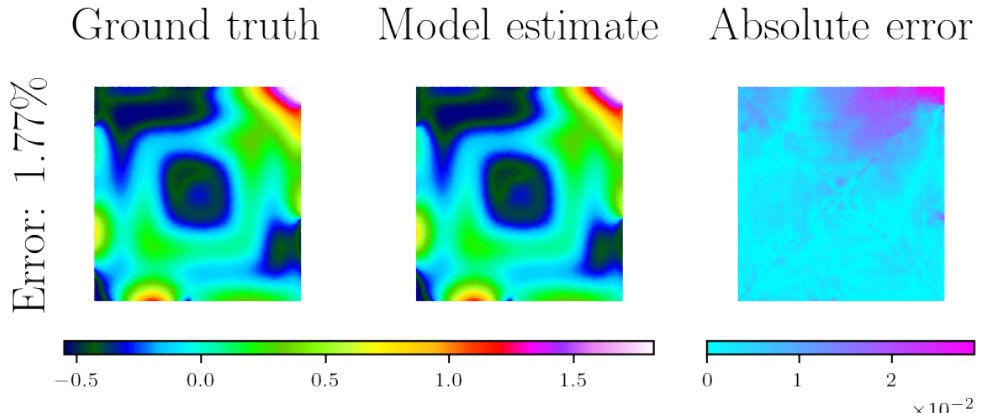

*Figure G.10.* Estimate of a trained LX(R) model for a test sample of the Poisson (Mixed) problem.

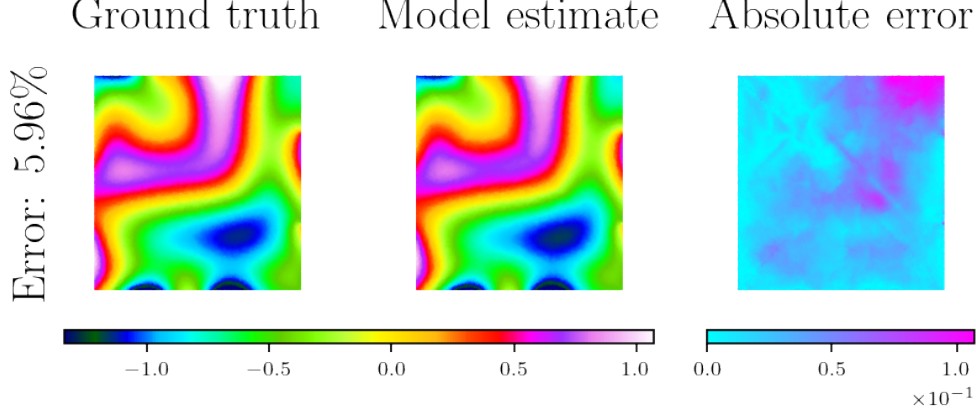

*Figure G.11.* Estimate of a trained LX(R) model for a test sample of the Poisson (Mixed+) problem.

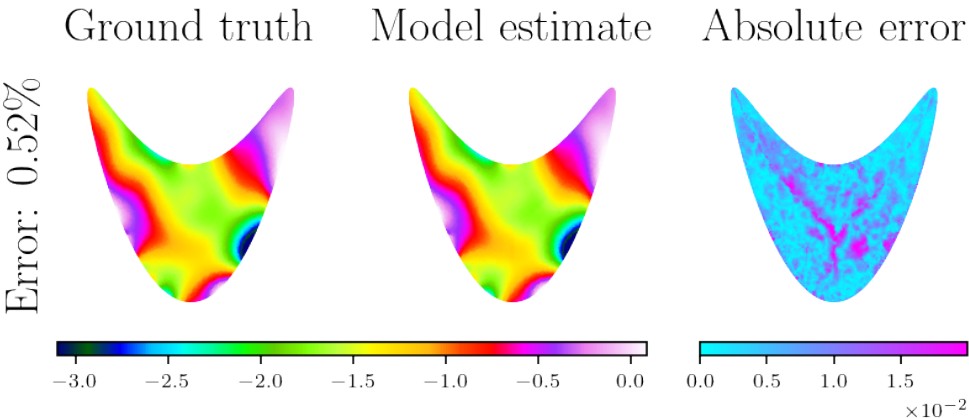

*Figure G.12.* Estimate of a trained LX(R) model for a test sample of the Poisson (Dirichlet) problem.

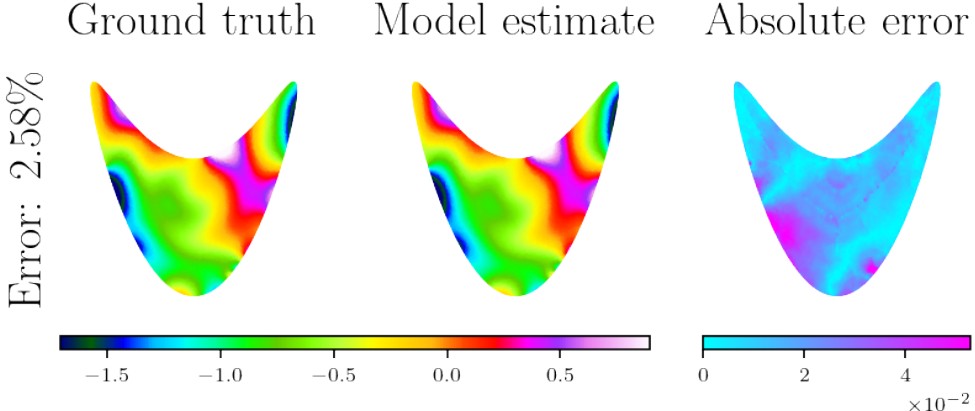

*Figure G.13.* Estimate of a trained LX(R) model for a test sample of the Poisson (Mixed) problem.

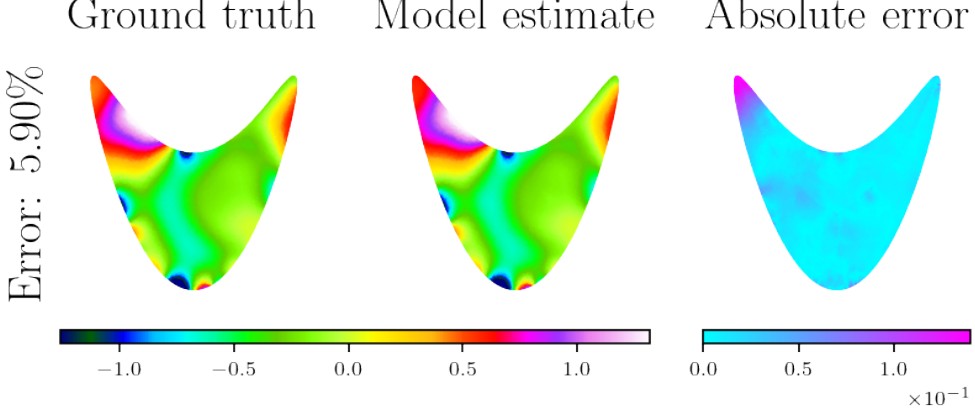

*Figure G.14.* Estimate of a trained LX(R) model for a test sample of the Poisson (Mixed+) problem.

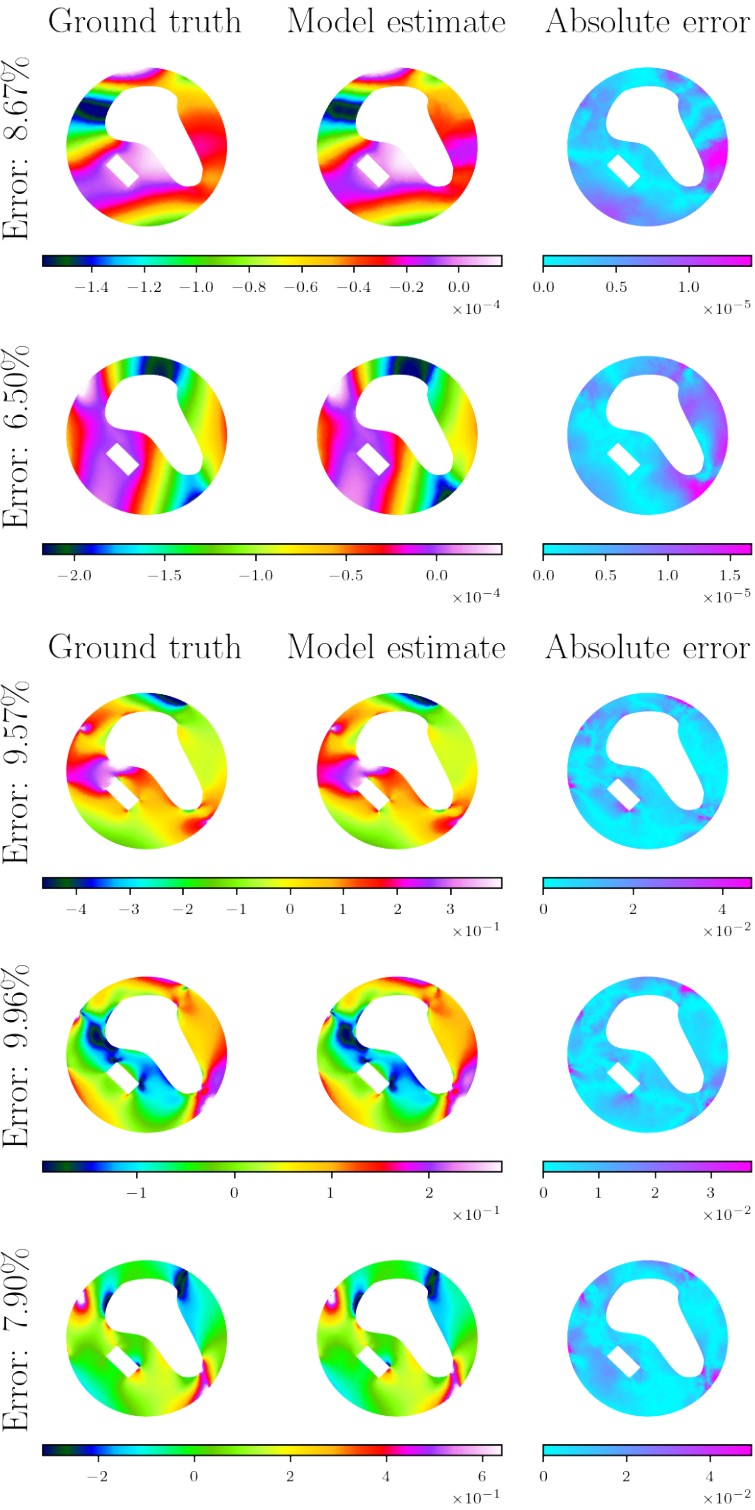

*Figure G.15.* Estimate of a trained LX(R) model for a test sample of the elasticity (Steel) problem. The first two rows correspond to displacement components and the following three rows to the stress components.

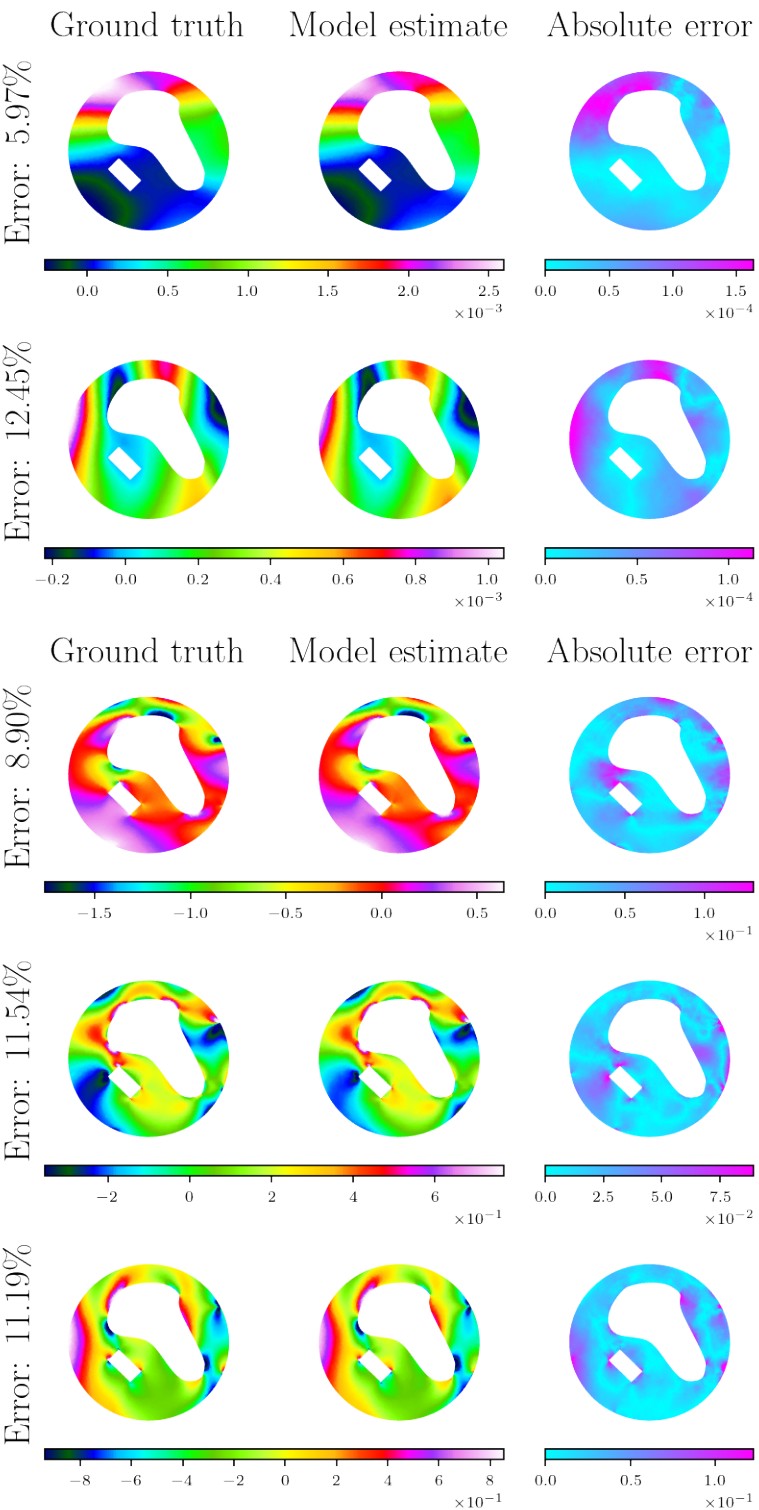

*Figure G.16.* Estimate of a trained LX(R) model for a test sample of the elasticity (Bone) problem. The first two rows correspond to displacement components and the following three rows to the stress components.

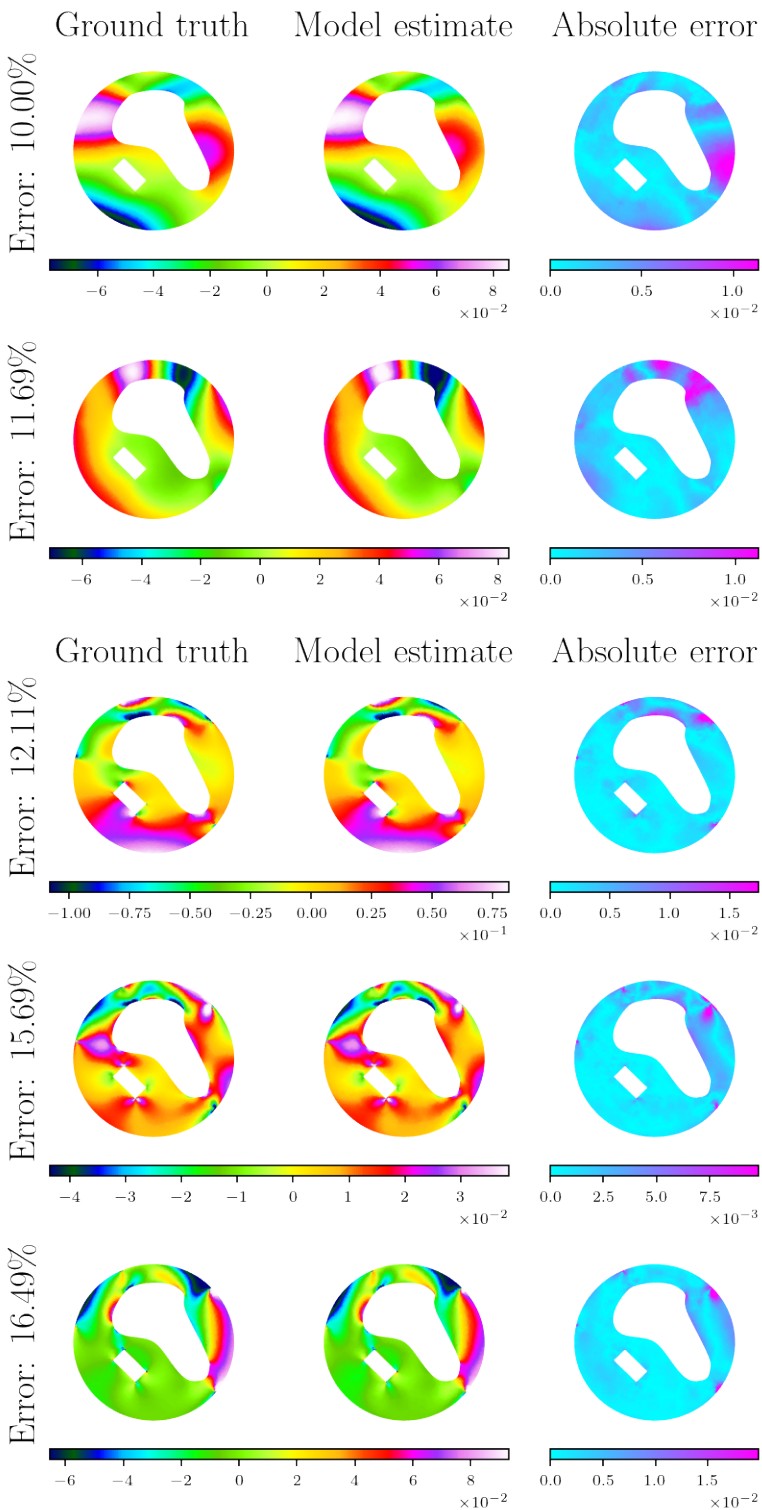

*Figure G.17.* Estimate of a trained LX(R) model for a test sample of the hyperelasticity (Rubber) problem. The first two rows correspond to displacement components and the following three rows to the stress components.

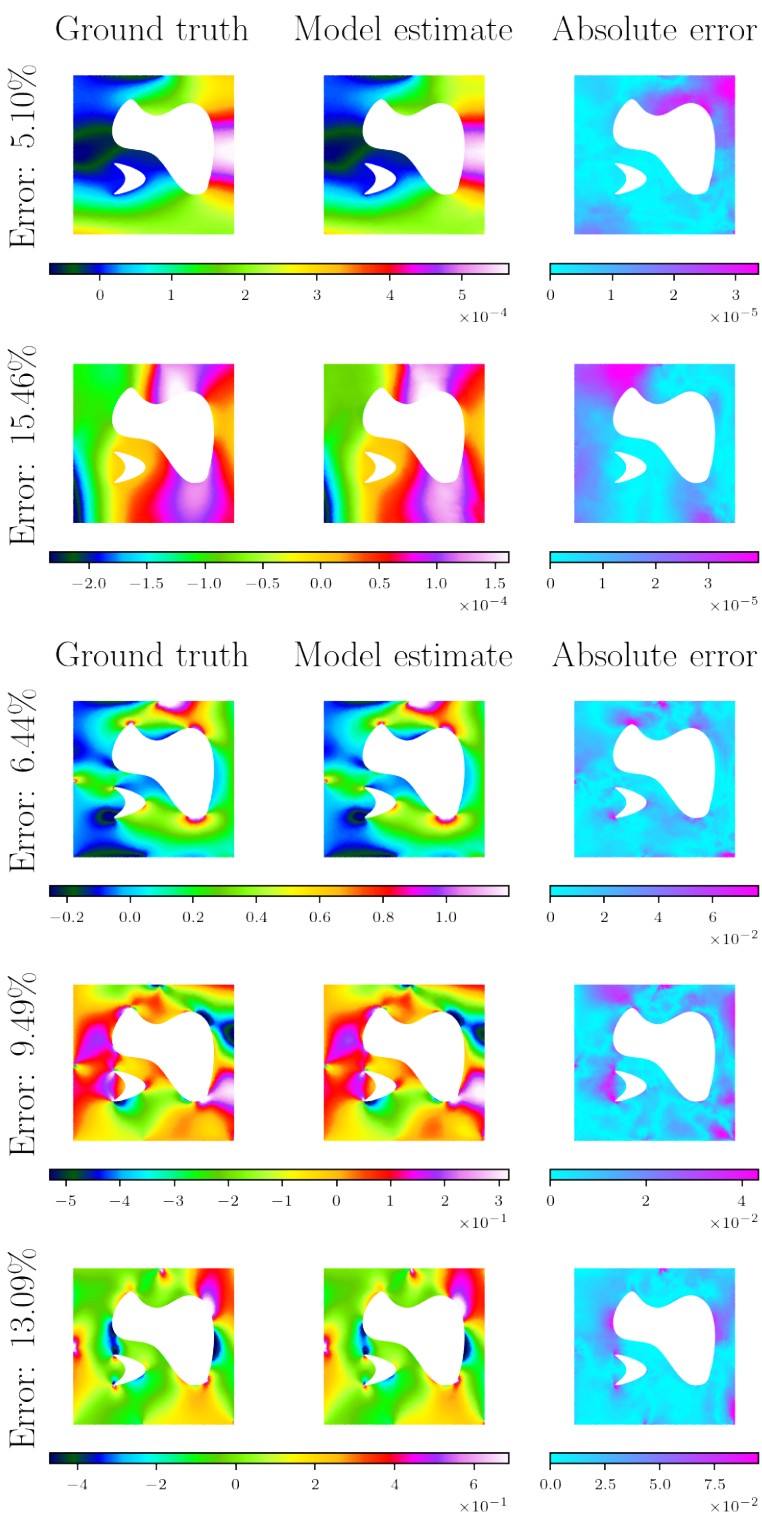

*Figure G.18.* Estimate of a trained LX(R) model for a test sample of the elasticity (Steel) problem. The first two rows correspond to displacement components and the following three rows to the stress components.

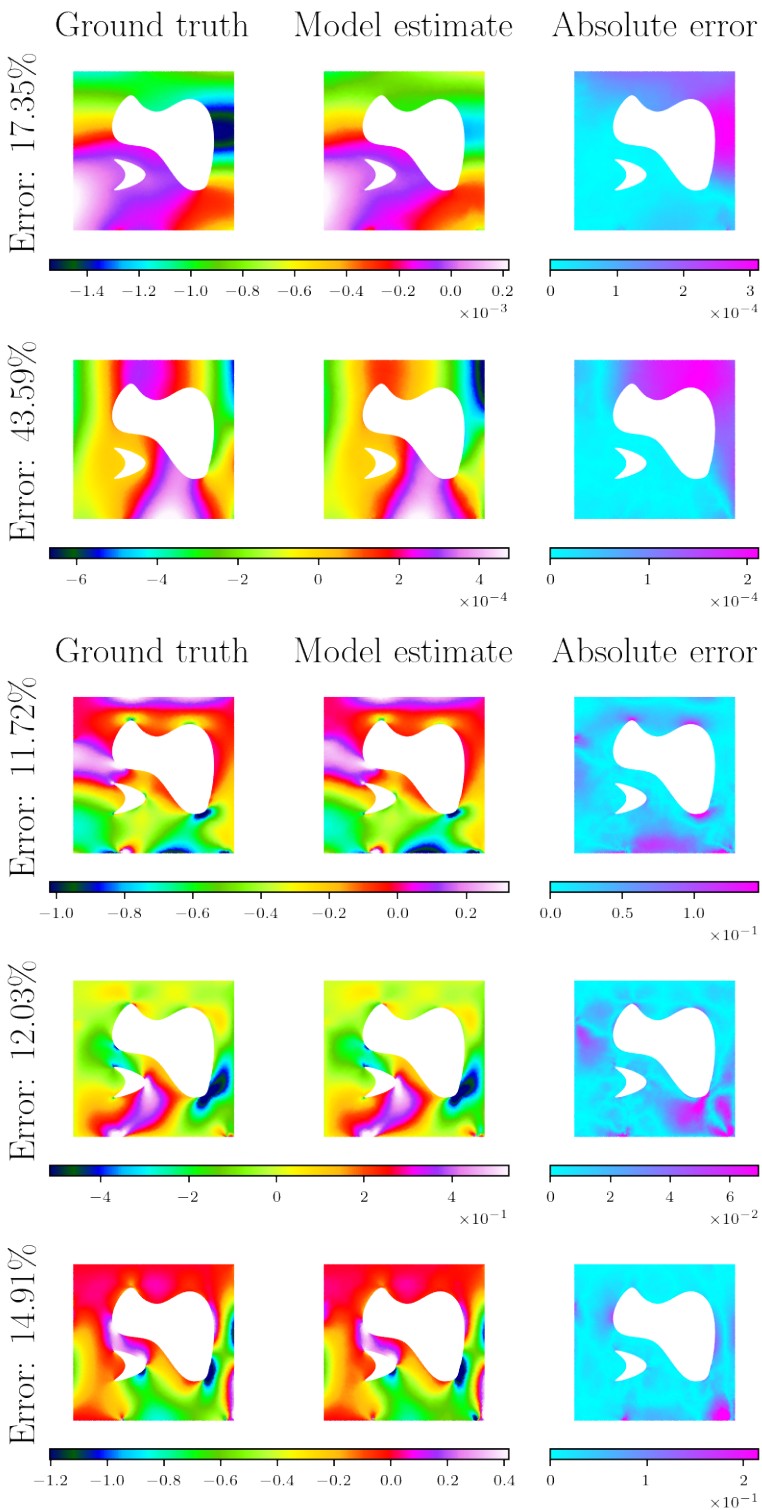

*Figure G.19.* Estimate of a trained LX(R) model for a test sample of the elasticity (Bone) problem. The first two rows correspond to displacement components and the following three rows to the stress components.

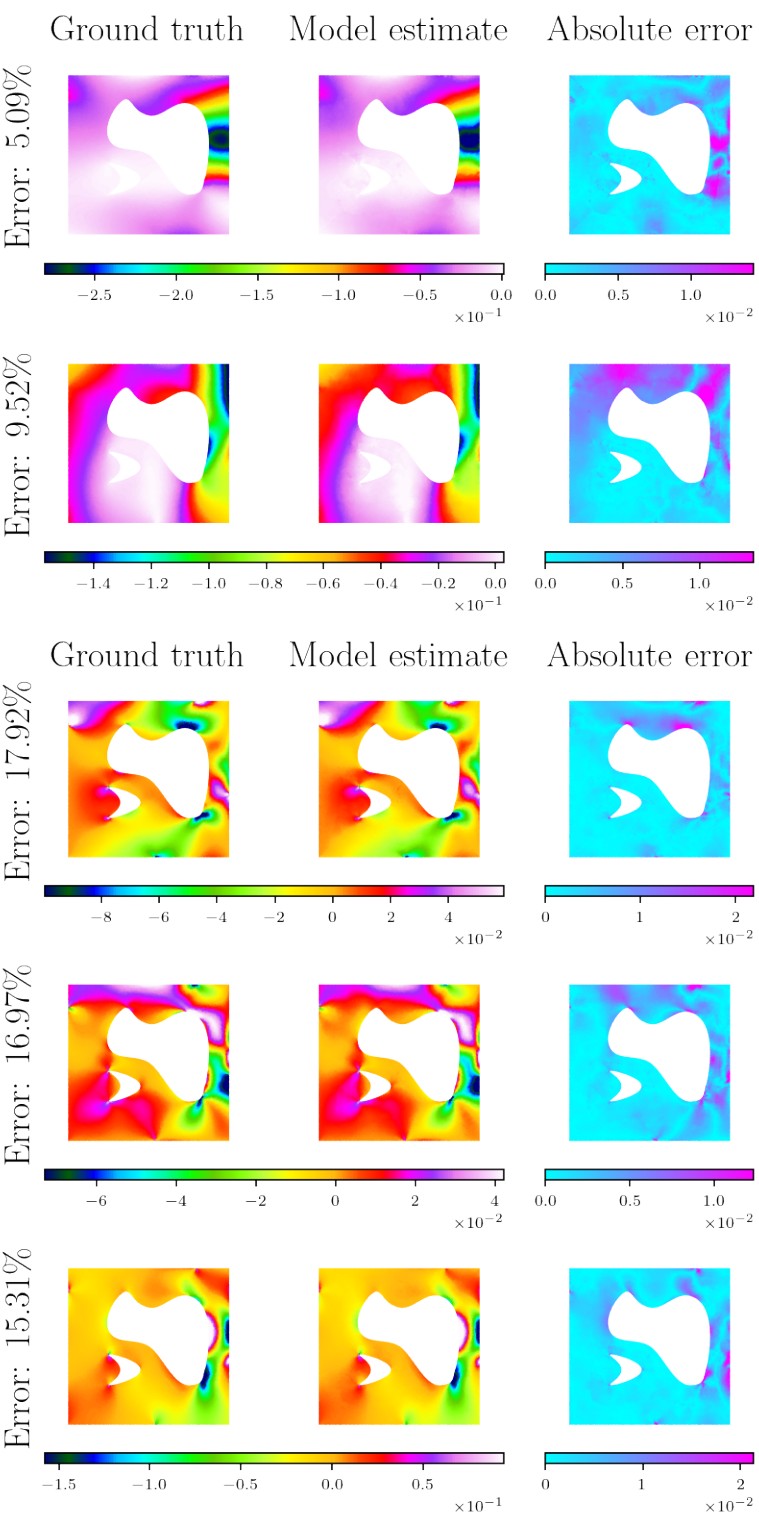

*Figure G.20.* Estimate of a trained LX(R) model for a test sample of the hyperelasticity (Rubber) problem. The first two rows correspond to displacement components and the following three rows to the stress components.

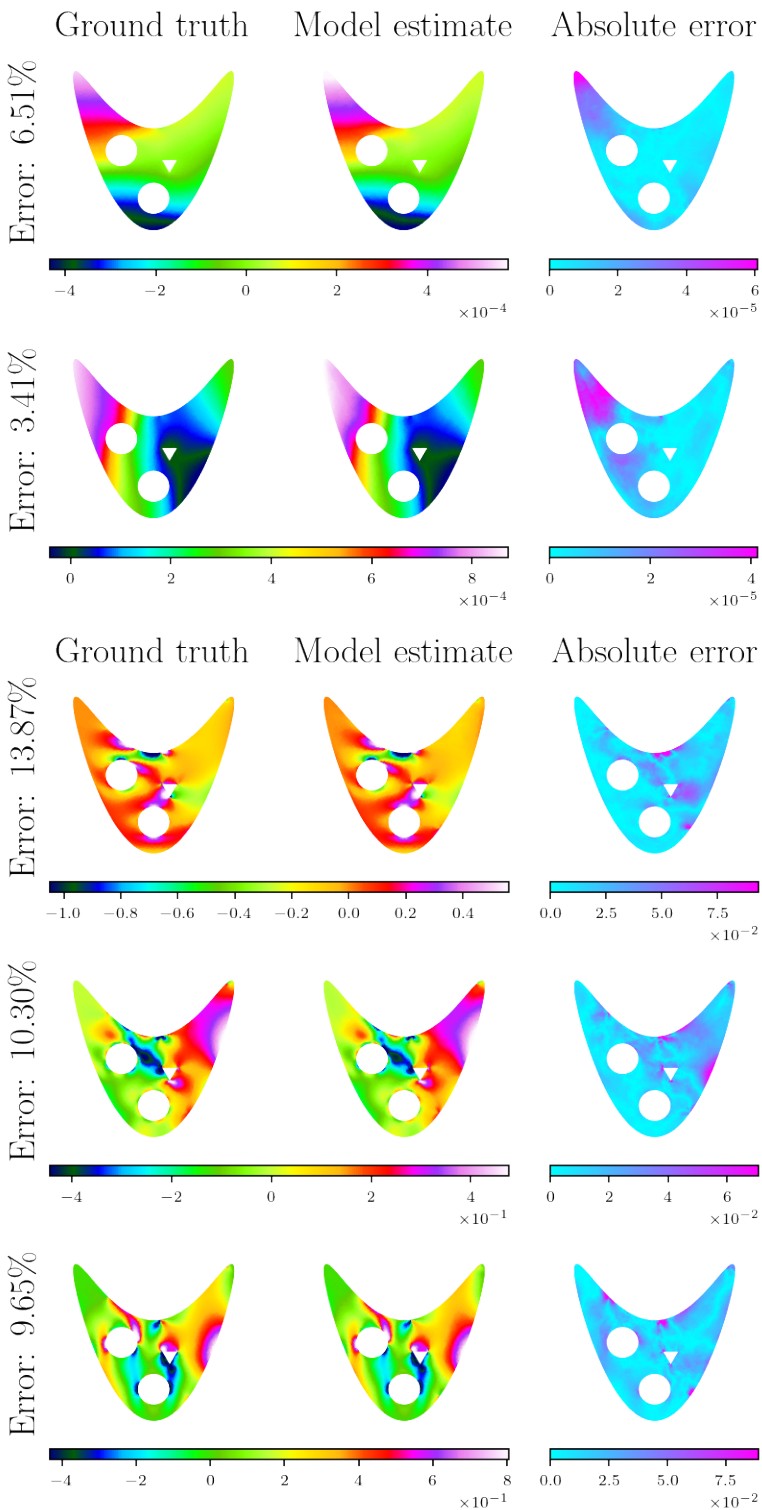

*Figure G.21.* Estimate of a trained LX(R) model for a test sample of the elasticity (Steel) problem. The first two rows correspond to displacement components and the following three rows to the stress components.

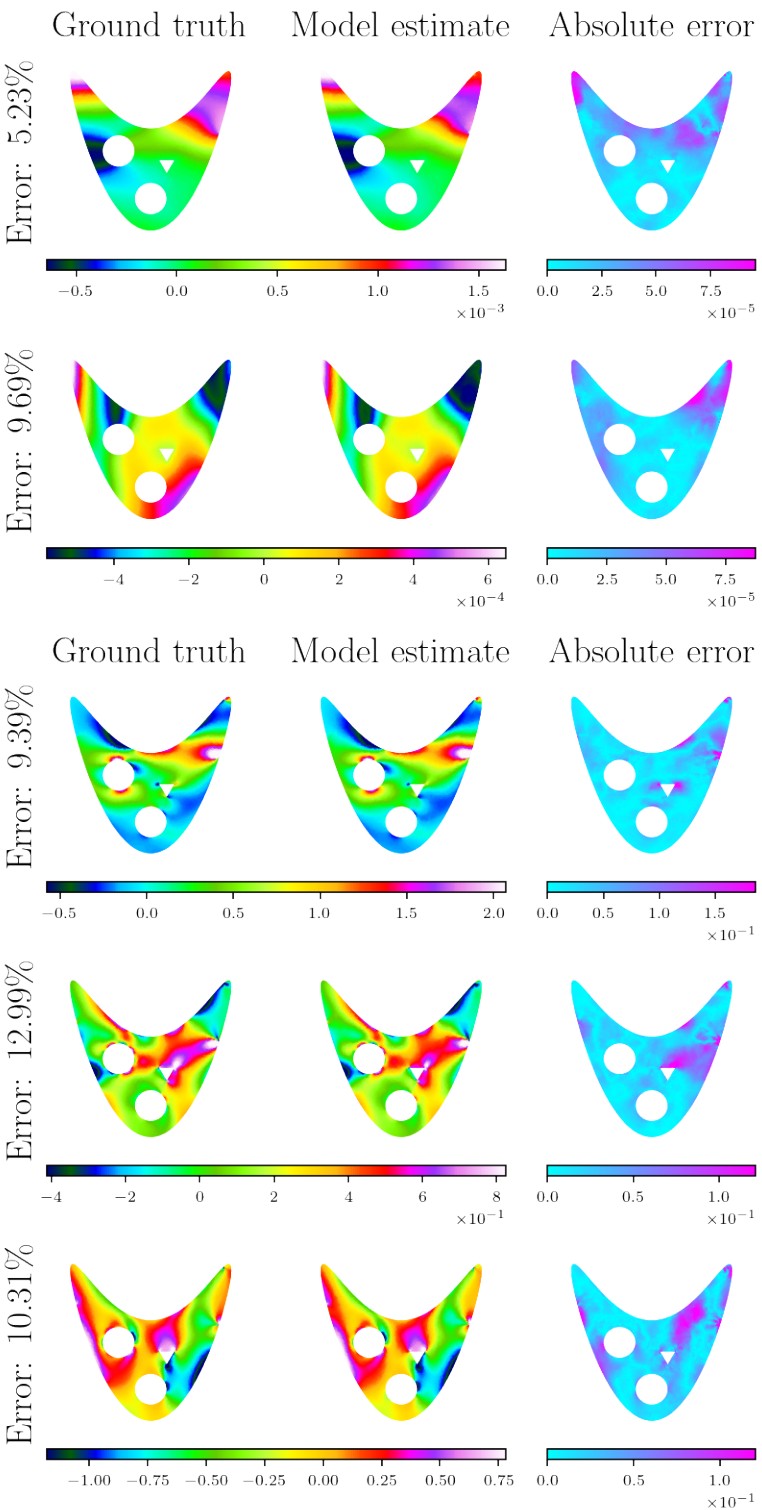

*Figure G.22.* Estimate of a trained LX(R) model for a test sample of the elasticity (Bone) problem. The first two rows correspond to displacement components and the following three rows to the stress components.

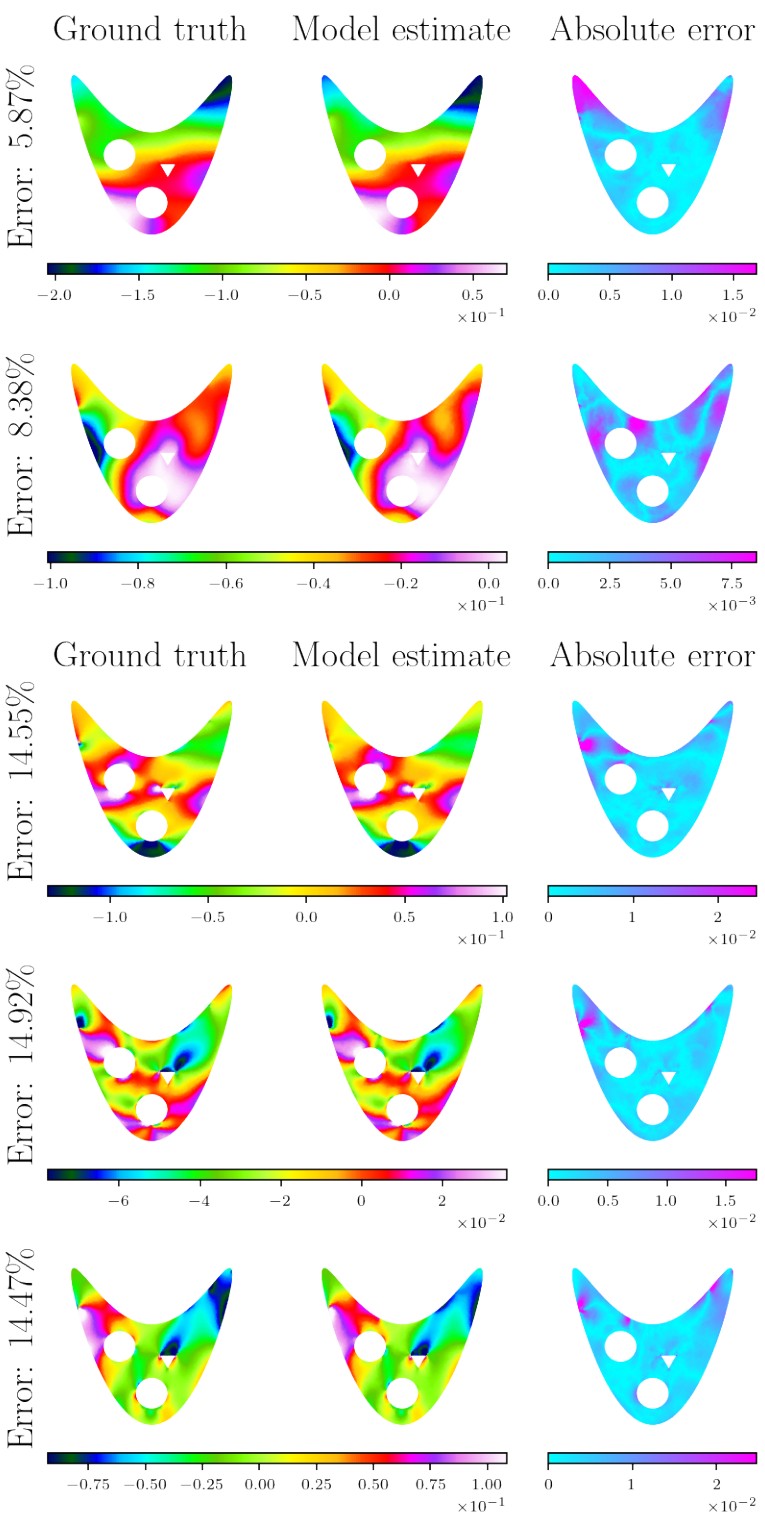

*Figure G.23.* Estimate of a trained LX(R) model for a test sample of the hyperelasticity (Rubber) problem. The first two rows correspond to displacement components and the following three rows to the stress components.

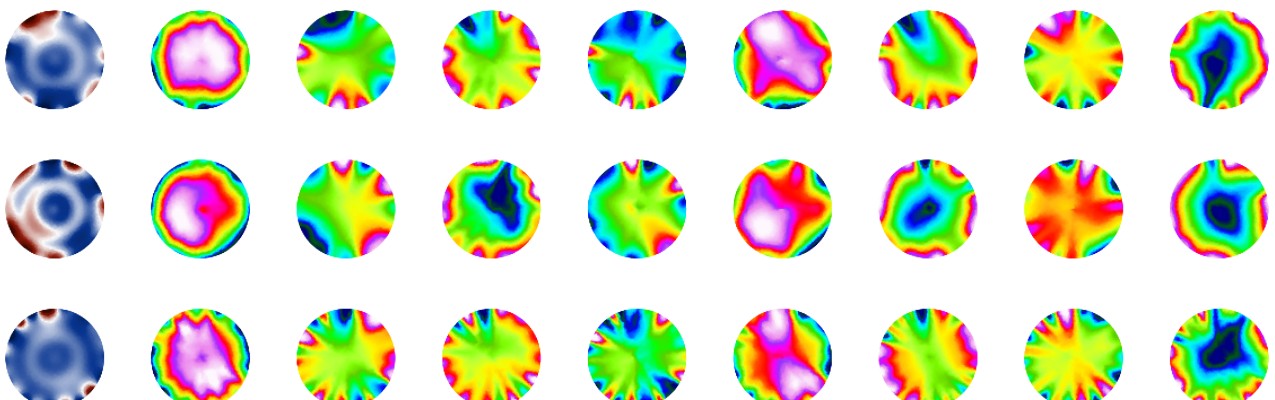

*Figure G.24.* Solution (first column) and the first 8 learned extensions for three test samples (rows) of the Poisson (Dirichlet) problem.

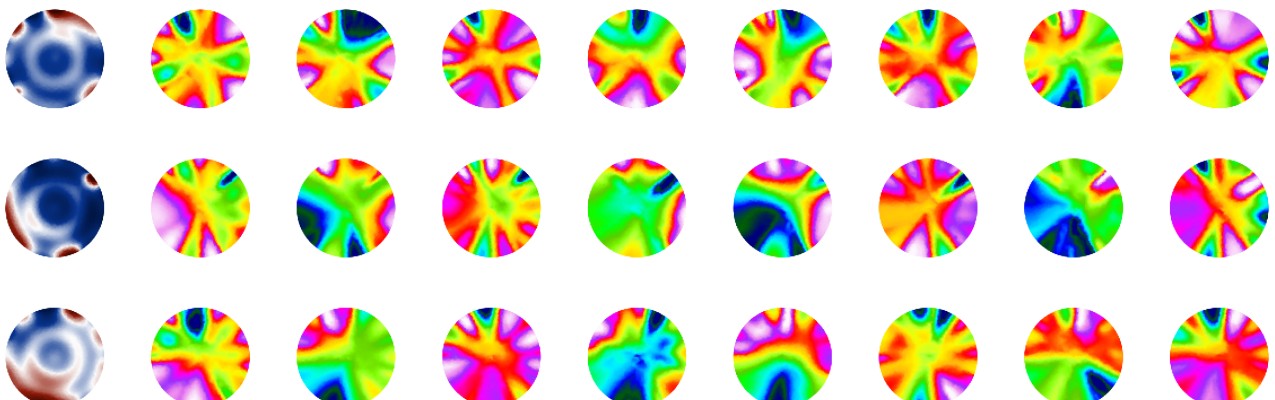

*Figure G.25.* Solution (first column) and the first 8 learned extensions for three test samples (rows) of the Poisson (Mixed) problem.

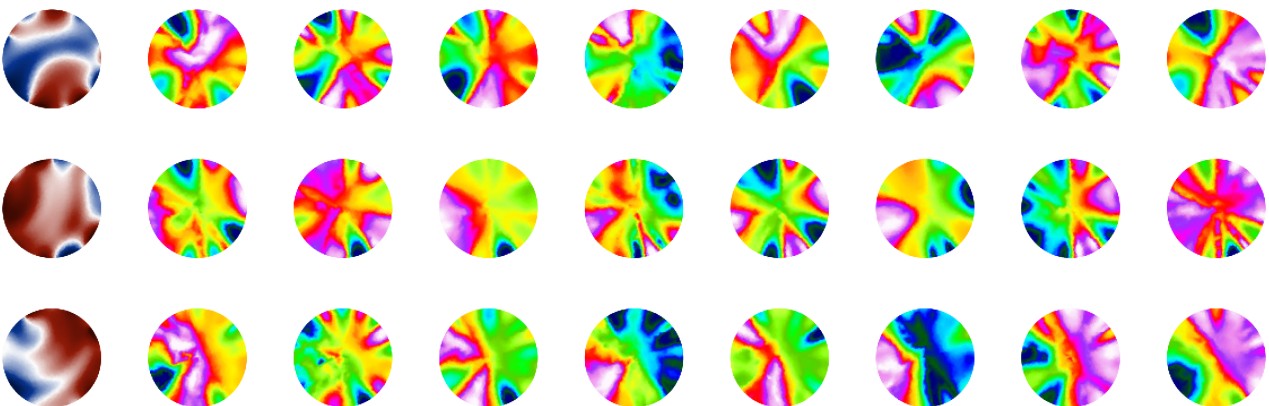

*Figure G.26.* Solution (first column) and the first 8 learned extensions for three test samples (rows) of the Poisson (Mixed+) problem.

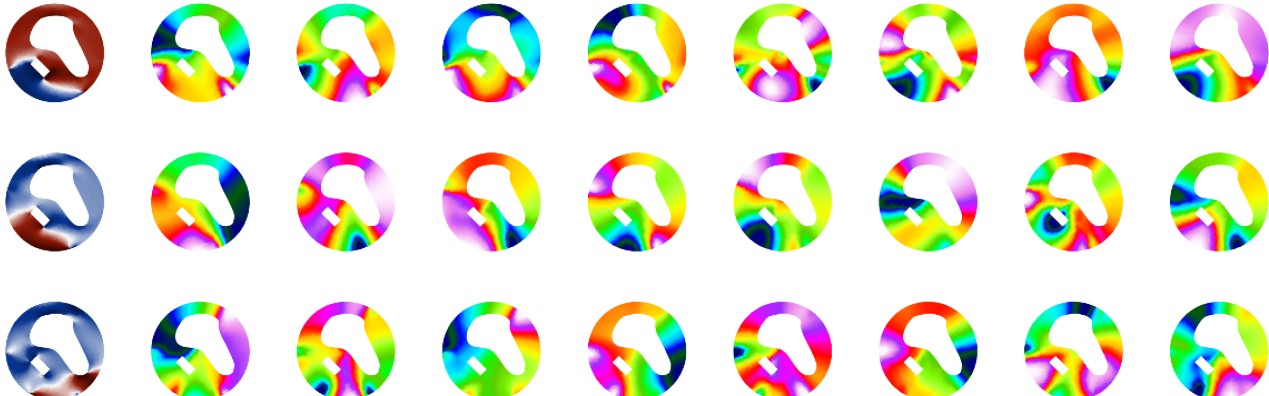

*Figure G.27.* Horizontal normal stress (first column) and the first 8 learned extensions for three test samples (rows) of the elasticity (Bone) problem.

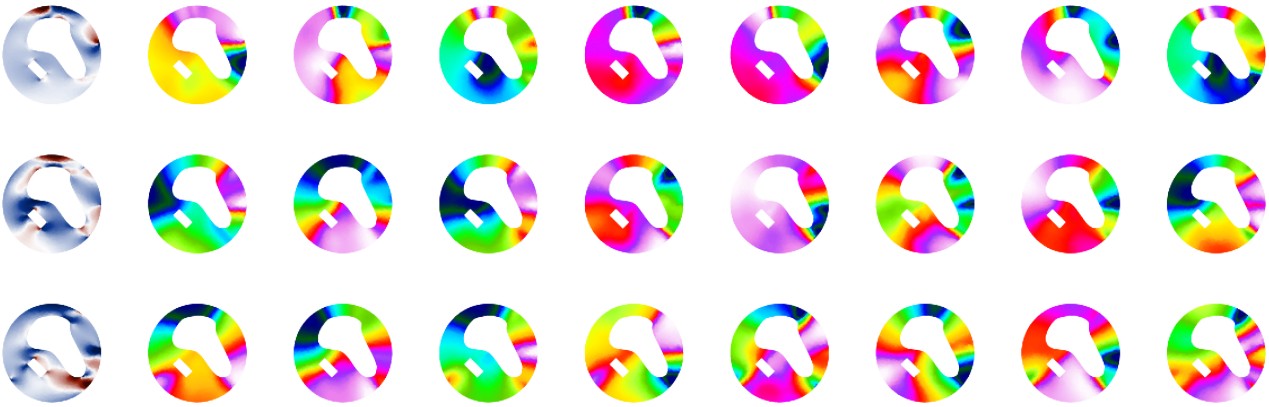

*Figure G.28.* Horizontal normal stress (first column) and the first 8 learned extensions for three test samples (rows) of the hyperelasticity (Rubber) problem.

