# OpenReview forum: "Imposing Boundary Conditions on Neural Operators via Learned Function Extensions"
_ICML.cc/2026/Conference — ICML 2026 regular_

### Official Review · Reviewer_ZYmy · 2026-03-10

**Soundness:** 2
**Presentation:** 3
**Significance:** 3
**Originality:** 2
**Overall Recommendation:** 4
**Confidence:** 4

**Summary:**

This paper introduces a general framework for conditioning neural operators on complex and highly variable boundary conditions (BCs) by utilizing **learned function extensions**.
The authors address a significant limitation in current operator learning (OL) frameworks: their inability to handle problems where the solution is extremely sensitive to varying boundary forcings, such as in elliptic partial differential equations (PDEs). Most existing models are restricted to constant or weakly varying BCs and often struggle with complex, non-Cartesian geometries.

The core of the proposed solution is to map boundary data onto a **latent pseudo-extension** defined over the entire spatial domain. This allows any standard domain-to-domain neural operator architecture to "consume" boundary information as if it were a standard domain input. The researchers developed an attention-based **extender module** that can be trained end-to-end with the core neural operator. This module uses cross-attention layers to inform domain features with boundary data, creating a rich representation of the BCs across the entire space.

### Key Contributions

**Generalized OL Framework:** A flexible framework that enables existing neural operator architectures to explicitly incorporate complex BCs through function extensions.

**Unified BC Representation:** A principled procedure that unifies Dirichlet, Neumann, and Robin BCs into a single, physically consistent, and normalized representation, avoiding training instabilities caused by scale imbalances.

**Learned Pseudo-Extensions:** The introduction of a learnable extender module that avoids the computational expense of traditional methods (like solving a Laplace equation for harmonic extensions) while maintaining high expressive flexibility.

**Comprehensive Benchmarking:** The creation of 18 challenging new datasets featuring Poisson, linear elasticity, and hyperelasticity problems. These datasets include diverse geometries with holes and complex BC configurations, such as eight boundary segments with randomized sizes, locations, and types.

**State-of-the-Art Performance:** Demonstration through extensive experiments that the proposed approach significantly outperforms existing baselines, achieving superior accuracy without requiring per-dataset hyperparameter tuning.

**Compliance With Llm Reviewing Policy:**

Affirmed.

**Key Questions For Authors:**

N.A.

**Limitations:**

Yes.

**Strengths And Weaknesses:**

### **1. Soundness**

The paper presents a technically robust framework motivated by mathematical techniques and backed by empirical evidence in 2D setting. Motivated by linear PDEs, where harmonic extension can help constructing solutions to Poisson equation with non-homogeneous BCs, the authors generalize this methodology to non-linear problems with learnable extensions.

**Strengths:**

* **Methodological Rigor:** The authors provide a principled procedure for unifying Dirichlet, Neumann, and Robin BCs into a physically consistent representation, which prevents training instabilities caused by scale imbalances. Specifically, extension is constructed by cross-attention between domain and boundary features.

* **Theoretical Grounding:** The use of "function extensions" is motivated by explicit mathematical constructions for linear problems. The authors demonstrate that for such problems, a domain-to-domain operator using these extensions exists and is affine.

* **Extensive Benchmarking:** The claims are supported by 18 new datasets covering complex physics (Poisson, linear elasticity, hyperelasticity) and geometries (non-convex, with holes).

* **Honest Comparison:** The paper evaluates multiple extender variants—Zero Extensions (0X), Harmonic Extensions (HX), and the proposed Learned Pseudo-extensions (LX)—and compares them against existing baselines like BENO.

**Weaknesses:**

* **Focus on Elliptic PDEs:** The authors focus on elliptic PDEs as a "worst-case scenario." While they claim the methods apply to initial-boundary value problems, the current empirical results are primarily static/time-independent. As the authors honestly admitted, the current experiment is limited to 2D problems. The effectiveness of the method lacks validation in 3D setting.

### **2. Presentation**

The submission is well-structured and follows the standard "encode-process-decode" paradigm used in operator learning.

* **Clarity of Narrative:** The problem formulation (Section 2) and methods (Section 3) clearly define the transition from boundary-to-domain mapping to a more manageable domain-to-domain mapping.

* **Effective Visuals:** Figure 1 provides a clear overview of how the extender module integrates into existing neural operator frameworks, making the conceptual "latent pseudo-extension" easy to visualize .

* **Reproducibility:** The authors include detailed appendices (e.g., Appendix B for normalization) and provide a DOI link to their 18 datasets, which significantly aids reproducibility.

### **3. Significance**
* **Addressing a Critical Limitation:** Most existing neural operators are limited to constant or weakly varying BCs. This work enables them to handle cases where the solution is extremely sensitive to boundary forcings—a requirement for many real-world engineering tasks .

* **Broad Utility:** The proposed extender module is architecture-agnostic; it can be "coupled with an arbitrary domain-to-domain neural operator," potentially improving models like FNO, CNO, or GNO.

### **4. Originality**

The work is original in its conceptualization of the boundary conditioning problem as a "learned extension" problem.

---

> ### Author Rebuttal · Authors · 2026-03-29
>
> We begin by thanking the reviewer for their detailed review and their recognition of the value of our work. We address their concerns in detail below.
>
> 1. **Lack of available BC-dominated datasets:** The reviewer has raised concerns about validating the framework on time-dependent problems. In this regard, we would like to clarify that, to the best of our knowledge, no prior work has addressed time-dependent BC-dominated settings in PDE operator learning. As a result, our evaluation is limited to our curated collection of 18 datasets. In this collection, **we prioritized diversity in geometry, boundary condition (BC) complexity, and PDE difficulty** over including time-dependent problems. This choice is supported in Lines 59–74 (left column), where we note that parabolic and hyperbolic PDEs exhibit more localized and time-delayed boundary effects, and are therefore generally less sensitive to BCs than elliptic problems. Furthermore, most prior datasets involve simple BCs that are either fixed or vary minimally over the dataset, allowing neural operators to learn solutions without explicitly incorporating BC inputs, vastly reducing the need for additional mechanisms like ours. Nonetheless, in response to the reviewer's suggestion, we evaluated our method on three time-dependent fluid flow datasets from Saad et al. (2023).
>
> 2. **Results on a Navier-Stokes problem:** Saad et al. (2023) provide datasets consisting of solution trajectories for the 2D lid-driven cavity problem with three Reynolds numbers. No-slip BCs are imposed on the walls, while a constant horizontal velocity is applied at the lid (top boundary), varying across samples in the range [1, 1.5]. Notably, the BCs in this problem can be fully parameterized by a **single scalar**: the lid velocity. In contrast, the BCs in our datasets are substantially more complex, cannot be reduced to low-dimensional parameterizations, and must instead be provided to the model in a discretized form. We kindly invite the reviewer to read Point 4 in our response to Reviewer `5rS1 ` in this regard. For our experiments, we represent the lid BC as a discretized function defined along the top boundary. The results, summarized below, show that **even for such simple BCs, incorporating lightweight extenders leads to consistent improvements in accuracy compared to standalone models.** Notably, RIGNO-LX achieves comparable accuracy to models that are 16x larger, and FNO-LX matches the accuracy of the BC-imposing BOON with a smaller model. These findings demonstrate that **learned extensions are also effective for time-dependent problems**, enabling high accuracy with significantly smaller models and consistently improving upon zero-padding baselines.
>
> ---
> > **Table: Relative $L^2$ test error [%] on the lid-driven cavity benchmark, averaged across all time steps. PINO and BOON results correspond to Saaed et al. (2023).**
> > ||Parameters [M] |Re=10|Re=100|Re=1000|
> > |:-:|:-:|:-:|:-:|:-:|
> > |**RIGNO-0X**|0.06|7.33|5.42|1.87|
> > |**RIGNO-LX**|0.1|0.61|0.40|0.67|
> > |**FNO-0X**|0.8|0.25|0.19|0.41|
> > |**FNO-LX**|1.1|0.20|0.14|0.22|
> > |**PINO**|1.6|0.72|0.53|0.69|
> > |**BOON**|1.6|0.21|0.14|0.18|
> ---
>
> 3. **Extension to 3D:** As the reviewer rightly points out, the current work does not include empirical results on 3D problems. However, we would like to emphasize that all components of the proposed framework are naturally extendable to 3D settings, and we do not anticipate significant technical or computational barriers in doing so. In particular, the learned extender operates on downsampled point clouds and leverages cross-attention mechanisms, making it **directly applicable to 3D point cloud representations**. Additionally, in the datasets shown in Figure 2b, we consider component-wise BCs applied independently across spatial dimensions. The framework successfully handles these **multi-dimensional BCs**, providing further evidence of its flexibility in multiple dimensions. Finally, we note that the 2D datasets used in our study are already large-scale, each consisting of 8K samples defined over as many as 16K spatial points. This scale exceeds that of many commonly used 2D benchmarks and is comparable to certain 3D datasets in the literature, which often contain fewer than 1K samples at spatial resolutions between 10K and 40K points.
>
> 4. **New benchmarks:** We would like to highlight new results using FNO as the backbone of our framework, which show that, due to their truncated spectral representation, FNOs struggle with zero-padded BCs. These experiments demonstrate that learned extensions are highly beneficial for FNOs, providing a smooth representation of the BCs. We kindly invite the reviewer to refer to Points 1 and 2 of our response to Reviewer `kYYg` for further details.
>
> We hope that our responses have addressed all of the reviewer’s concerns and kindly ask the reviewer to consider updating their assessment in light of these clarifications and the new experimental results.

---

> > ### Author Rebuttal · Reviewer_ZYmy · 2026-04-03
> >
> > I understand that 3D experiments would cause extra effort and work, but lacking it still affects the value of the paper for those who intend to follow the extension method for boundary condition. Therefore, my score remains.

---

> > > ### Author Response · Authors · 2026-04-05
> > >
> > > We thank the reviewer for their feedback and agree that including experiments on 3D problems would be a valuable extension. While our current study focuses on 2D settings, the considered problems are already comparable to 3D cases in both scale and complexity (see Point 4 in our response to Reviewer `5rS1`). Moreover, all components of our framework, including the unification and normalization of BC types, the general extension approach, and the cross-attention-based learned extender, are directly applicable to 3D problems without requiring modifications or introducing additional challenges. With the introduction of 18 datasets featuring highly complex boundary conditions, we make a significant step forward in problem difficulty, moving beyond simple scalar BC representations to rich, fully discretized boundary functions. Given this level of complexity and the inclusion of multi-dimensional effects in 2D, we expect similar trends and effectiveness to carry over to 3D settings.
> > >
> > > We are pleased to have addressed the reviewer’s other concerns through our rebuttal. In particular, the inclusion of a time-dependent problem further strengthens the evaluation and helps demonstrate the effectiveness of the proposed method more comprehensively.

---

### Official Review · Reviewer_k6gE · 2026-03-11

**Soundness:** 3
**Presentation:** 2
**Significance:** 3
**Originality:** 3
**Overall Recommendation:** 4
**Confidence:** 3

**Summary:**

This paper focuses on neural operators in handling complex boundary conditions and proposes some approaches that map boundary data to latent pseudo-extensions over the entire spatial domain to enhance the performance. It unifies the three types of boundary conditions (Dirichlet, Neumann, and Robin) and achieves physically consistent normalization. Experimental results demonstrate that the framework achieves better accuracy across various geometries and boundary configurations, compared to the baseline model.

**Compliance With Llm Reviewing Policy:**

Affirmed.

**Final Justification:**

My main concern is that this paper compared many models proposed by themselves rather than some SOTA baselines. Based on the experiments, it is unclear in which situation which model should be used.

**Key Questions For Authors:**

Questions:
1. How about the performance of BENO + proposed approaches?
2. Since there is no requirement on $\psi$, could one use the given formulation to generate $\psi$, for example, using $\nabla \cdot P =0 $for  your elasticity problems?
3. Could you provide more details about the training and inference time? Since a transformer module is introduced and the boundary-condition inputs are extended, will the training time increase significantly?
4. In which cases would you recommend using the Zero Extension (0X)?

Suggestions:
1. Figure 2 is quite complex. I strongly suggest presenting the results in tables with highlighted minimum values, which would improve readability.
2. Add the results of the original RIGNO and GAOT methods for comparison.

**Limitations:**

yes

**Strengths And Weaknesses:**

Strengths:
1. This paper proposes an approach that unifies three types of boundary conditions, and the approach has good generalization.
2. The proposed learnable extensions can be trained simultaneously and are more efficient than harmonic extensions.
3. The paper provides comprehensive experiments, especially those showing that accuracy scales with the size of the dataset and that learned extensions generalize to unseen geometries, unseen boundary-condition types, and unseen physics, which are important and valuable aspects.

Weaknesses:
1. Figure 1 appears misleading. The solved problems are 1D and 2D, but the domain function is illustrated as a 3D problem.
2. Since the loss function remains unchanged, introducing the learned extensions seems more like adding an additional module to exploit features from the boundary input.
3. It is unclear what is the effect of Zero Extensions (0X).

---

> ### Author Rebuttal · Authors · 2026-03-29
>
> We begin by thanking the reviewer for their thoughtful comments, constructive suggestions, and their recognition of the value of our work. We address their concerns in detail below.
>
> 1. **Effect on Speed**: The proposed framework allows flexible extender sizes, letting users match the module to the expected complexity of boundary conditions (BCs). Adding an extender increases training and inference times proportionally. For a fair comparison, standalone models should be scaled to match the size of the extended models. In Appendix F.3, we report such experiments and compare test errors across different model sizes. For instance, a standalone RIGNO achieves 38.6% test error with 4.5ms inference latency, whereas a smaller RIGNO with a lightweight extender **reduces the error to 10.1%, improves latency to 4.0ms, and maintains comparable total parameters (1.8M) and training throughput (192 samples/sec)**. Although the extender increases model size, it delivers significant gains in accuracy and speed for complex BCs, outperforming similarly sized standalone models. We thank the reviewer for raising this important point and, if accepted, we will expand this discussion in the camera-ready version (CRV).
>
> 2. **Standalone models and 0X**: The datasets in this work are highly sensitive to BCs, making it infeasible for standalone models without BC inputs to accurately predict solutions. For example, in the Poisson problem with Dirichlet BCs, the source and domain are fixed across samples, yet **the solutions vary widely, influenced solely by the BCs** (see Appendix Figure C.2). Other datasets are similarly designed to isolate BC effects. Without BC inputs, models would converge to an average field, if they converged at all. Zero padding (0X) provides a simple way to incorporate BCs by trivially extending boundary information into the domain interior, without modifying the architecture, which is why standalone models are excluded, and 0X is used as a fair baseline. While 0X performs adequately for simple PDEs or transformer-based models, learned extensions further improve accuracy at the cost of additional implementation effort. We will clarify this reasoning more explicitly in the CRV if the paper is accepted.
>
> 3. **Modular design**: The reviewer is correct that the proposed learned extender serves as an additional module leveraging boundary features, without requiring modifications to the loss function. We consider this simplicity and flexibility a key strength of our framework rather than a limitation. The extender is transferable across different backbones and can substantially improve performance in the presence of complex BCs. In this regard, we kindly invite the reviewer to read Point 5 in our response to Reviewer `kYYg`. In this context, BENO already incorporates a mechanism to absorb BCs and serves as a benchmark for comparison. Since BENO couples this mechanism with a graph neural network backbone, replacing it with our learned extensions would correspond to a version of the LX(R) model in Figure 2.
>
> 4. **Using the PDE in the extender**: In the context of Problem (1), satisfying $\mathcal{D}(a, u) = 0$ requires access to $u$. However, $u$ is the output of the model and depends strongly on the extension. As a result, it is not straightforward to directly incorporate the PDE prior to computing the extension. That said, physics-informed terms can still be incorporated into the loss function. In such a setup, one can combine the extension $\psi$ with the solution $u$ to minimize $\mathcal{B}(\psi, u)$. This approach aligns more closely with the motivational example in Appendix A.1, but it may introduce additional computational overhead during training, particularly if gradient computations are required within $\mathcal{B}$.
>
> 5. **New benchmarks:** We would like to highlight new results using **FNO as the backbone of our framework** and **applying our method to a time-dependent problem**. We kindly invite the reviewer to read Points 1-2 in our response to Reviewer `kYYg` and Points 1-2 in our response to Reviewer `ZYmy`.
>
> 6. **Editorial comments:** We thank the reviewer for their suggestions and will consider using a simpler representation of Figure 2 in the CRV, if accepted. Regarding Figure 1, all images correspond to 2D domains from our hyperelasticity dataset (Rubber) with the Boomerang geometry. Two aspects may give the impression of a third dimension: 1. multiple layers at each stage represent multiple channels of 2D functions, such as stress components; and 2. irregular deformations of the output field can make it appear 3D, even though it depicts a single stress component of a deformed 2D body. If accepted, we will clarify these points in the CRV to prevent any potential misunderstandings.
>
> We hope that our responses have satisfactorily addressed all of the reviewer’s concerns, and in light of these clarifications and the new results, we kindly ask the reviewer to consider updating their assessment.

---

> > ### Author Rebuttal · Reviewer_k6gE · 2026-04-01
> >
> > My concerns have been adequately addressed. Considering the 0X is useless (I do not think it is a suitable baseline). I will keep my score

---

> > > ### Author Response · Authors · 2026-04-01
> > >
> > > We thank the reviewer for their acknowledgement and are glad to have addressed most of their concerns adequately.
> > >
> > > Considering the 0X baseline, we would like to highlight the following points:
> > >
> > > 1. **The considered task requires a very high level of flexibility, making most prior work inapplicable.** Across the 18 provided datasets, we study extremely complex BCs on irregular geometries with holes, including **mixed-type** and **component-wise** BCs defined by **multi-modal** sinusoidal functions over up to **8 random segments**. Prior to this work, the most complicated BC setup studied could be parameterized with a **single scalar**. In contrast, our BCs must be discretized over the full boundary ($\approx$1K points). This significant advancement creates a clear gap in the existing literature, limiting the availability of comprehensive and directly applicable baselines.
> > >
> > > 2. **Prior BC-admitting frameworks face limitations.** While some prior works have addressed incorporating boundary conditions in operator learning, BENO is the only BC-aware neural operator that can be applied to our very general learning tasks with minimal modifications. Other methods face important limitations: PENN (Horie & Mitsume, 2022) cannot handle Robin BCs and requires mesh-based inputs; BOON (Saad et al., 2023) is limited to Cartesian boundaries and uniform grids; PhyMPGN (Zeng et al., 2025) does not support non-homogeneous Neumann/Robin BCs or general geometries; SPFNO (Liu et al., 2023) is limited to Cartesian domains and spatially constant BCs; and Kashi et al. (2024) is also limited to Cartesian domains. Overall, our work targets a largely overlooked yet essential problem in operator learning, where strong and flexible baselines are scarce.
> > >
> > > 3. Other reviewers have requested presenting the results of standalone GAOT and RIGNO in the benchmarks. While we explain that standalone GAOT and RIGNO do not support BC inputs (Point 2), **we consider 0X as the most natural use of standalone architectures.** It leverages the capacity of the core architectures without introducing significant computational overhead or implementation complexity.
> > >
> > > 4. **0X remains the most practical baseline.** Given the limited applicability of prior methods, the poor performance of BENO, and the high cost of HX, 0X is the only broadly viable baseline. In particular, 0X consistently matches or outperforms BENO on all datasets. Moreover, our experiments (including FNO, Point 2 in our response to Reviewer `kYYg`) show that architectures respond differently to 0X: while FNO and RIGNO struggle with zero-padded BCs, GAOT achieves competitive performance, confirming that 0X is a reasonable and meaningful baseline.
> > >
> > > To conclude, given the lack of flexible BC-aware alternatives, the strong performance of 0X across multiple architectures, and its minimal assumptions and overhead, we believe that 0X constitutes a fair and appropriate baseline. Importantly, our results consistently demonstrate that learned extensions (LX) provide clear and systematic improvements over this baseline, particularly in complex BC settings, thereby validating the effectiveness and generality of the proposed framework. We view this work as pioneering in regard to absorbing BCs in operator learning, and hope it will serve as a strong baseline for future works.
> > >
> > > We hope that these additional clarifications have addressed the reviewer’s remaining concerns. If so, we kindly request the reviewer to reconsider and update their assessment accordingly, while remaining available for any further inquiries and questions.

---

### Official Review · Reviewer_5rS1 · 2026-03-13

**Soundness:** 3
**Presentation:** 3
**Significance:** 3
**Originality:** 3
**Overall Recommendation:** 5
**Confidence:** 2

**Summary:**

This paper proposes a neural operator framework for solving boundary value problems (BVPs) via function extension. By normalizing boundary conditions, the method unifies different boundary types and incorporates domain and boundary functions as separate inputs. Experiments demonstrate strong performance under challenging boundary conditions and showcase several extension strategies.

**Compliance With Llm Reviewing Policy:**

Affirmed.

**Final Justification:**

The paper is well-written and addresses a meaningful problem. My concerns have been satisfactorily addressed in the rebuttal. I maintain my recommendation of Accept.

**Key Questions For Authors:**

- I am curious about the results when RIGNO and GOAT are used as the main models without applying the extension.
- Other graph-based or domain-aware models also seem applicable in this setting. Is there a particular reason why only BENO is included? It would be interesting to see comparisons with additional models.
- While the extension idea is understandable at the equation level, it is unclear whether the downsampled (encoded) extension remains meaningful. In the actual implementation, the extension is not even added but simply concatenated with the encoded domain representation. Could the authors provide further justification or explanation for why this design is still meaningful?

**Limitations:**

yes

**Strengths And Weaknesses:**

## Soundness

#### Strength

The idea of extending boundary conditions to the entire domain is interesting and appears to be a natural approach for solving BVPs.

#### Weakness

- It is unclear whether the proposed approach would still work well for more complex PDEs, such as those arising in fluid dynamics.
- While the extension idea seems theoretically reasonable in the problem formulation and methodological description, the actual implementation in the model appears to apply the extension differently. In practice, the encoded domain representation and the extension are concatenated in a downsampled (embedding) space. In this case, it seems that the method might be applied in a way that differs significantly from the original extension concept, potentially weakening its mathematical interpretation.
- The comparison between models is somewhat limited. It would be beneficial to include more baseline models, especially graph-based models that allow flexible inputs. In addition, for RIGNO and GAOT, it is unclear why the results without using the extension are not reported. Ultimately, the main comparison model appears to be only BENO.


## Presentation
#### Strength

Although the concept of extension could be quite complex, the explanations related to normalization and mapping are generally clear and well presented.

#### Weakness

The paper seems to emphasize the advantages of the learned extension (as also suggested by the title). However, in Figure 2a, the LX results sometimes appear worse than in other cases. While the results in Figure 2b are discussed in detail, there is little to no discussion about the results in Figure 2a. Even if the performance is worse, it would be helpful to provide possible explanations or discuss the limitations of the approach.

## Significance
#### Strength
Beyond the methodology itself, the problem setting is meaningful in that it focuses on BVPs rather than IVPs. Many experiments in the neural operator literature primarily consider IVPs, so addressing BVPs and proposing a method tailored to this setting is valuable.

#### Weakness
Given that many neural operator models today can already handle irregular grids effectively, it would strengthen the paper if the method were demonstrated on more challenging PDEs or more complicated boundary conditions. Alternatively, including more extensive model comparisons could also help clarify the advantages of the proposed approach.

## Originality
#### Strength
The idea of using an extension framework and allowing the model to automatically learn meaningful information through this process is interesting and creative.

#### Weakness
In PINN training, there already exist approaches where boundary conditions are incorporated into the governing equation (for example, by multiplying boundary-related functions), although not necessarily in the BVP setting considered here. From a broader perspective, applying boundary conditions across the whole domain may therefore not be entirely unique.

---

> ### Author Rebuttal · Authors · 2026-03-29
>
> We begin by thanking the reviewer for their thoughtful comments and constructive suggestions. We address their concerns in detail below.
> 1. **Extending in the downsampled space:**  The reviewer has made a subtle and correct observation that the implemented architecture differs in some respects from the motivational concept described in Appendix A. We clarify these differences in two parts:
>   **a.** Extensions are concatenated with the encoder output rather than added directly to the predicted solution. This reflects the **data-driven nature** of the model and reinterprets pseudo-extensions as *encodings* of the input boundary conditions (BCs). By fusing these encodings upstream, the model can leverage the expressive capacity of its processor to learn sophisticated interactions with other inputs. We note that addition remains possible within this framework and could be learned during training.
>   **b.** Extensions are computed directly in the downsampled space to improve efficiency. Since the extender’s cross-attention scales as $O(NM)$, with the number of domain nodes $N$ much larger than the number of boundary nodes $M$, operating in this compressed space **reduces memory and computational costs** in proportion to the encoder’s compression ratio, without affecting the model’s expressivity. This design choice also preserves interpretability. As illustrated in Figures 24–28 of Appendix G, the learned extensions, although computed in the downsampled space, closely resemble the BCs and can be viewed as fields over the full domain, with each channel capturing distinct yet consistent features across samples.
> 2. **Fluid dynamics dataset:** We have conducted additional experiments that demonstrate the effectiveness of our approach for a Navier-Stokes problem. Our results show that a very small extender can help achieve accuracies that are on par with baselines that are 16 times larger in size. Furthermore, we show that learned extensions consistently yield improvements over the zero-padding baseline and match the performance of BOON while paired with a smaller FNO. We kindly refer the reviewer to Points 1 and 2 in our response to Reviewer `ZYmy` for more details.
> 3. **Comparison between models:** In the benchmarks of Figure 2, learned extensions (LX) are compared to zero extensions (0X) and harmonic extensions (HX) as baselines. Given that BCs are the main driving forces in the considered datasets, and regular neural operators only accept domain functions as inputs, standalone RIGNO and GAOT are not applicable. Please see Point 2 in our response to Reviewer `k6gE` in this regard. BENO serves as the only BC-admitting neural operator that is applicable to our datasets. We elaborate on this in Point 1 of our response to Reviewer `kYYg`. Additionally, we would like to highlight new results using an FNO backbone, detailed in Point 2 of our response to Reviewer `kYYg`.
> 4. **Complex BCs:** We would like to emphasize that in the 18 provided datasets, comprising 156K samples, we have considered extremely complex BCs on irregular geometries with holes. We consider **mixed-type, component-wise BCs defined by multi-modal sinusoidal functions** on up to **8 random segments**. Prior to this work, the most complicated BC setup studied could be parameterized with a single scalar. In contrast, our BCs must be discretized over the entire boundary on 1K points.
> 5. **Results of Figure 2a:** The reviewer correctly notes the limited advantage of LX in Figure 2a. These results are particularly discussed in Lines 262 (left) to 268 (right). There, we explain that the advantages of LX for simple PDEs are not as prominent as for complex problems. Even so, for the Poisson problem, LX improves the average accuracy of 0X by 12% and outperforms it in 6 of 9 datasets. In the Mixed configuration, where BC complexity and solution sensitivity are highest, 0X achieves an average of 2.61% error, while LX reduces it to 1.71%. The very low errors seen with HX result from their close similarity to the true solution; they act as *perfect extensions* for Poisson problems but incur extra computational cost and are therefore not fair baselines. We further develop this discussion in Appendix A.2, where we provide an explanation for this observation.
> 6. **BCs in PINNs:** PINNs use physics-informed loss functions and can **accept BCs symbolically**; for example, as additional integral terms in a PDE’s variational formulation. These methods rely on classical numerical techniques and are not directly applicable to purely data-driven operator learning. In contrast, our framework allows models to **absorb BC data specifically within operator learning**, addressing a problem that has largely been overlooked or only studied in very simplified settings.
>
> We hope that our responses have adequately addressed the reviewer’s comments and questions, and we kindly request the reviewer to reconsider their assessment in light of these clarifications and the new results.

---

> > ### Author Rebuttal · Reviewer_5rS1 · 2026-04-02
> >
> > My questions and concerns have been satisfactorily resolved. I will keep my current score. Thank you.

---

> > > ### Author Response · Authors · 2026-04-05
> > >
> > > We sincerely thank the reviewer for reading our rebuttal and are glad to have satisfactorily addressed their questions and concerns. We also appreciate their constructive suggestions, which will help us further improve the clarity of the paper.

---

### Official Review · Reviewer_kYYg · 2026-03-13

**Soundness:** 3
**Presentation:** 4
**Significance:** 3
**Originality:** 3
**Overall Recommendation:** 5
**Confidence:** 4

**Summary:**

The authors provide a novel framework for imposing boundary conditions on existing neural operator approaches. This framework consists of using function extensions of Dirichlet, Neumann, and Robin-type boundary conditions to extend boundary data into the interior of the domain. Three variations are proposed: 0-padded-extensions, harmonic extensions and a cross-attention based learned pseudo-extension NN module.

Further, a benchmark dataset of several PDEs with varying boundary conditions and geometries (including non-convex domains with holes) is introduced as part of this submission. The authors use this dataset to demonstrate superior performance of their learned extension module when used in conjunction with respectively one existing transformer -- and graph-based NO approach.

**Compliance With Llm Reviewing Policy:**

Affirmed.

**Final Justification:**

The authors addressed my concerns well, especially the incorporation of an additional neural operator.

**Key Questions For Authors:**

None

**Limitations:**

yes

**Strengths And Weaknesses:**

# Strengths

The authors present a novel, learnable BC extender module framework for NOs as well as a novel dataset to benchmark. The paper is well-written and the experiments, including ablation studies are extensive. Further, the presentation of a novel benchmark as part of this paper complements the contribution of this submission to the advancement of boundary condition treatment in neural operators. Through careful investigations, the authors also demonstrate strong performance of their method.
Reproducibility is also enabled through the provision of the code.


# Major Issues

## 1. Only 2+1 neural operators were investigated

The methodological comparison could profit from having more models being included in the comparison. The authors compare one existing BC-information-admitting NO to just two NOs they augment with their extensions. Since the extension framework is introduced as general, one can expect to include more existing neural operators such as FNO, DeepONets, UPT and even foundation models such as Poseidon, and Walrus.

## 2. No Time-dependent PDEs are considered

Although the authors claim direct applicability to initial-boundary value problems, this is not demonstrated. Time-dependent problems are important application cases for neural operators. An additional time-dependent benchmark experiment could help demonstrate the generality of the proposed approach, especially since different problem classes exhibit different behaviour.

# Minor Issues

1. **Design novel neural operator based on the framework:** The motivation of why the framework introduced in this work as a mere extension of existing neural operators could be more clearly stated. Instead of applying the extender module to existing neural operators, why not devise a novel NO that directly incorporates the learned extender module itself? Even if one would like to use the framework as a model-agnostic improvement, a dedicated neural operator could still constitute an additional point of comparison.


# Editorial Comments

* The methods section could benefit from a small introduction to provide an overview to the reader what is to be expected.
* Line 92: The solution space is introduced but without specifying what m stands for.

---

> ### Author Rebuttal · Authors · 2026-03-29
>
> We begin by thanking the reviewer for their thoughtful comments, constructive suggestions, and their recognition of the value of our work. We address their concerns in detail below.
>
> 1. **BC-admitting baselines:** While some prior works have addressed incorporating boundary conditions (BCs) in operator learning, BENO is the only BC-aware neural operator that can be applied to our (very general) learning tasks with minimal modifications. Other methods face important limitations: PENN (Horie & Mitsume, 2022) cannot handle Robin BCs and requires mesh-based inputs; BOON (Saad et al., 2023) is limited to Cartesian boundaries and uniform grids; PhyMPGN (Zeng et al., 2025) does not support non-homogeneous Neumann/Robin BCs or general geometries; SPFNO (Liu et al., 2023) is limited to Cartesian domains and spatially constant BCs; and Kashi et al. (2024) is also limited to Cartesian domains. Overall, our work targets a largely overlooked yet essential problem in operator learning, where strong and flexible baselines are scarce. For these reasons, we adopt zero-padded extensions of available neural operators as a fair baseline to our proposed learned extensions.
>
> 2. **FNO as backbone:** Following the reviewer’s suggestion, we conducted additional experiments using the widely used FNO as the backbone of our framework. Since FNO requires inputs and outputs on a uniform grid, we interpolated the Poisson Square dataset onto a $128^2$ grid. The results (see the table below) indicate that **FNO substantially benefits from learned extenders** in problems with high sensitivity to BCs. In particular, an FNO with 0.8M parameters (depth 2, width 20, 16 modes) struggles to accurately capture BC influence when zero-padding is used. Zero-padding introduces high-frequency Fourier modes, which are difficult to represent under the truncated spectral modes of FNO. By contrast, a lightweight extender with only 0.2M parameters produces **smooth BC representations** (see Figures 24–28 of Appendix G), making them more **compatible with FNO’s spectral truncation**. This enables FNO to better capture the dependency on BCs and achieve significantly lower errors, although still much larger than GAOT and RIGNO backbones (see Figure 3). Should the paper be accepted, we would be happy to include these additional results in the camera-ready version (CRV).
> ---
> > **Table: Relative $L^2$ test error [%] of the Poisson problem (Square) with 4096 training samples.**
> > |Model|Dirichlet|Mixed|Mixed+|
> > |:-:|:-:|:-:|:-:|
> > |**FNO-0X**|2.51|20.0|30.9|
> > |**FNO-LX**|1.33|3.86|7.89|
> ---
>
> 3. **Other backbones:** The proposed framework consistently improves performance using both graph-based and transformer-based neural operators, as well as FNOs. For other transformer-based models like UPT, Walrus, and Poseidon, we expect similar trends as for GAOT. Our transfer learning experiments (page 7, left column) demonstrate that learned extenders have shown promise in the context of foundation models. However, we leave this investigation to future work, as it requires careful evaluation of different training and adaptation scenarios, as current foundation models have not been trained on BC-dominated problems.
>
> 4. **Time-dependent PDEs:** We have conducted additional experiments that demonstrate the **effectiveness of our approach for a Navier-Stokes problem**. We kindly ask the reviewer to read Points 1 and 2 in our response to Reviewer `ZYmy`, where we show that learned extensions consistently yield improvements over the 0X baseline and match the performance of BOON while paired with a smaller FNO.
>
> 5. **Novel neural operator:** Numerous domain-to-domain neural operators exist in the literature, each with specific strengths and limitations. By introducing a flexible framework, we allow the choice of the core backbone to be tailored to the problem at hand, while enabling it to incorporate BCs. For comparison, BENO admits BCs but is tied to specific design choices. Although it achieves acceptable accuracy on some problems, its performance is not consistent (see Figure 2). This variability is largely due to architectural design choices specific to BENO rather than the underlying mechanism for incorporating BCs. Flexible frameworks such as ours are easily **adaptable to different problem requirements** and capable of **leveraging future architectural innovations**.
>
> 6. **Editorial comments:** The Sobolev order $m$ in $\mathcal{W}^{m,p}$ corresponds to the highest derivative appearing in the PDE. In a strong formulation of a second-order PDE, $m=2$ is typically required, whereas a weak formulation reduces this requirement to $m=1$. If the paper is accepted, the extra page allowance in the CRV will enable us to clarify this point and include a brief introductory paragraph in Section 3.
>
> We hope that our responses have fully addressed the reviewer’s comments and concerns, and we kindly ask the reviewer to consider updating their assessment in light of these clarifications.

---

> > ### Author Rebuttal · Reviewer_kYYg · 2026-04-02
> >
> > I thank the authors for their detailed rebuttal and appreciate the incorporation of FNO as a backbone. I maintain my score.

---

> > > ### Author Response · Authors · 2026-04-05
> > >
> > > We sincerely thank the reviewer for their positive feedback on our rebuttal and are glad that their concerns have been addressed. We also appreciate their thoughtful suggestions, particularly regarding experiments with additional backbones and time-dependent problems, which have helped us further improve the quality of the paper and and better demonstrate the effectiveness of our approach.

---

### Decision · Program_Chairs · 2026-04-30

**Decision:**

Accept (regular)

**Comment:**

This paper addresses an important and underexplored problem in neural operator learning: handling complex, highly variable boundary conditions across diverse geometries and PDEs. Reviewers agreed that the proposed framework is interesting, practically relevant, and well executed. The core idea of learning boundary-to-domain extensions was viewed as a flexible, architecture-agnostic mechanism, and the new benchmark datasets were also seen as a valuable contribution.

The empirical evaluation was a clear strength. Reviewers highlighted the breadth of experiments across 18 challenging datasets, multiple PDE classes, diverse boundary-condition settings, and several backbones, together with strong ablations and good reproducibility support. Overall, the results were considered convincing and showed clear gains in boundary-condition-dominated regimes.

Some concerns were raised about scope, particularly the limited initial coverage of backbones, time-dependent problems, and 3D settings. The rebuttal addressed several of these points constructively by clarifying baseline choices and adding further results with FNO and on a time-dependent Navier-Stokes problem. While the lack of 3D experiments remains a limitation, it does not outweigh the overall strength of the paper.

Overall, this is a technically solid paper with strong empirical support, useful released resources, and clear practical significance. I recommend acceptance.